# Volcanism and basalt weathering drove Ordovician climatic cooling

He Zhao [1], Lei Zhang [2] ✉, Thomas J. Algeo [2,3,4,5], Zhengyi Lyu[2], Xiangdong Wang[2] & Fang Hao[6]

Causal connections among major Ordovician environmental and biological events (i.e., long-term climatic cooling, Hirnantian Glaciation, Great Ordovician Biodiversification Event, and Late Ordovician Mass Extinction) remain in debate, and the hypothesis of volcanism-driven long-term cooling is untested. Here, we present both newly generated and compiled literature data for global volcanic activity (Hg geochemistry), sea-surface temperatures (conodont oxygen isotopes), and chemical weathering intensity (conodont strontium isotopes). This dataset documents a ~25-Myr-long interval of climatic cooling (~470-445 Ma), commencing around the onset of the Middle Ordovician, intensifying during the Late Ordovician, and ultimately culminating in the Hirnantian Glaciation. Cooling was associated with long-term intensified volcanic rock (basalt) weathering and atmospheric $pCO_2$ drawdown, as well as episodic marine photic-zone euxinia, during and after major volcanic episodes. These relationships favor volcanism (i.e., continental large igneous provinces) as the primary driver of contemporaneous environmental and climatic changes, thus revealing complex modulation of life-environment coevolution during the Ordovician Period.

The Ordovician Period (485.4–443.8 Ma) was bookended by major biotic events in the marine realm, i.e., the early stages of the Great Ordovician Biodiversification Event (GOBE) at its onset and the Late Ordovician Mass Extinction (LOME), the first of the "Big Five" biocrises, near its termination[1]. During the GOBE, which spanned the early Tremadocian (Early Ordovician) to middle Katian (Late Ordovician) with an early Darriwilian peak[2,3], the numbers of marine invertebrate species and genera increased by factors of ~6× and 3×, respectively. A key aspect of this diversification event was the proliferation of marine plankton (e.g., primary producers such as photosynthetic cyanobacteria), representing the "Ordovician Plankton Revolution"[4]. The LOME, which had a ~70–80% species-level extinction rate, occurred in two phases corresponding to the onset and termination of the Hirnantian Glaciation: LOME-1 in the early Hirnantian, which eliminated many tropical taxa, and LOME-2 during the mid-Hirnantian, which decimated the cool-water *Hirnantia* Fauna[5]. A precursor biocrisis (LOME-0) has recently been identified in the mid-Katian[2,6]. The driving mechanisms of the Ordovician marine biological radiations and extinctions have long been speculated upon without arriving at a consensus[7,8].

Despite their complex internal structure and chemical composition[9], the oxygen isotopic composition of conodonts ($\delta^{18}O_{conodont}$) is a valuable tool for study of sea-surface temperatures (SSTs) in the Paleozoic Era[10]. Conodont $\delta^{18}O$ data have established that the GOBE coincided with a long-term global cooling trend with a total magnitude of ~10 °C[10–12], a pattern confirmed by clumped isotope analysis of carbonates[13]. Climatic cooling is postulated to have been a direct driver of evolution among marine invertebrates[10], as

[1]College of Marine Science and Technology, China University of Geosciences, Wuhan, China. [2]State Key Laboratory of Geological Processes and Mineral Resources, China University of Geosciences, Wuhan, China. [3]State Key Laboratory of Geomicrobiology and Environmental Changes, China University of Geosciences, Wuhan, China. [4]Department of Geosciences, University of Cincinnati, Cincinnati, OH, USA. [5]State Key Laboratory of Oil and Gas Reservoir Geology and Exploitation, Chengdu University of Technology, Chengdu, China. [6]National Key Laboratory of Deep Oil and Gas, China University of Petroleum (East China), Qingdao, Shandong, China. ✉e-mail: zhanglei_cug@sina.com

have rising oxygen levels in the atmospheric-oceanic system[7], although the pattern and trigger of the Ordovician climatic cooling remain contentious. For example, it is debated whether this cooling trend was a continuous long-term event or an episodic, multistage process[10–12]. Atmospheric $CO_2$ levels, a key determinant of the long-term climate evolution of the Earth, are thought to have fallen through enhanced silicate weathering[14] linked to the spread of the earliest land plants[15], or possibly to elevated marine primary productivity[16], which are not necessarily mutually exclusive mechanisms. Moreover, the patterns of change in climate and biodiversity in South China (i.e., the most intensively studied craton, and one that may have been the cradle of diversification) versus at a global scale are significantly different[2], complicating an understanding of the causal relationship between Ordovician climatic cooling and contemporaneous bioevolutionary developments[17]. Long-term (>50 Myr) greenhouse-icehouse climatic oscillations were commonly driven by fluctuations in dominance between continental and island-arc volcanism during geological history[18]. The protracted interval of Ordovician climatic cooling (leading to the Hirnantian Glaciation) is speculated to have been induced by volcanic activity[19], offsetting transient warming associated with greenhouse gas emissions from volcanic eruptions[16]. However, integrated studies of volcanic and paleotemperature proxies are lacking to test this hypothesis, leaving the role of volcanism in the contemporaneous long-term cooling uncertain[14].

Volcanism is the principal natural source of mercury (Hg) in the Earth-surface system[20]. Volcanic eruptions can emit substantial quantities of elemental mercury having a residence time of a few years (~1-2 yr) in the atmosphere, which facilitates its long-range transport and eventual deposition in both marine and terrestrial sediments[21,22]. Massive inputs of Hg during major volcanic events may exceed the absorption capacity of organic matter at the sediment-water interface, resulting in positive anomalies of the ratio of total mercury to total organic carbon (TOC) (Hg/TOC) in sedimentary successions that are signals of volcanic Hg inputs[20]. The isotopic composition of Hg serves as an additional powerful tool to decipher sources and mechanisms of Hg enrichment (e.g., Zhao et al.[22]). Hg isotopes can reveal both mass-dependent fractionation (MDF; $\delta^{202}Hg$) and mass-independent fractionation (MIF; $\Delta^{199}Hg$, $\Delta^{200}Hg$, $\Delta^{201}Hg$). Critically, MIF arises predominantly through aqueous or atmospheric photochemical interactions antecedent to Hg sedimentation, and its isotopic signals are resilient against postdepositional diagenetic alteration. Mercury emissions from subaerial volcanoes typically exhibit near-zero MIF values with minimal variance[20,23].

Here, we studied three biostratigraphically well-constrained Ordovician marine successions at Huanghuachang (n.b., the Global Stratotype Section and Point (GSSP) of the Lower/Middle Ordovician boundary), Chenjiahe (n.b., the global auxiliary stratotype section of the same boundary), and Wangjiawan (n.b., the GSSP for the base of the Hirnantian Stage), which are closely spaced (<15 km apart) near Yichang City (Fig. 1 and Supplementary Figs. 1-3). We generated paired high-resolution profiles of oxygen isotopes for conodont bioapatite ($\delta^{18}O_{conodont}$) as paleotemperature proxies, Hg-system chemistry (Hg/TOC, $\Delta^{199}Hg$ and $\delta^{202}Hg$) as volcanic proxies, and strontium isotopes for conodont bioapatite ($^{87}Sr/^{86}Sr_{conodont}$) as a continental weathering proxy, with compilation of global published datasets, and additional constrains of Ordovician time framework based on published biostratigraphic and newly generated carbonate carbon isotope ($\delta^{13}C_{carb}$) chemostratigraphic data. These data allowed us to evaluate the patterns and test volcanism as the main cause of Ordovician climatic cooling, providing key insights regarding life-environment co-evolution during this period.

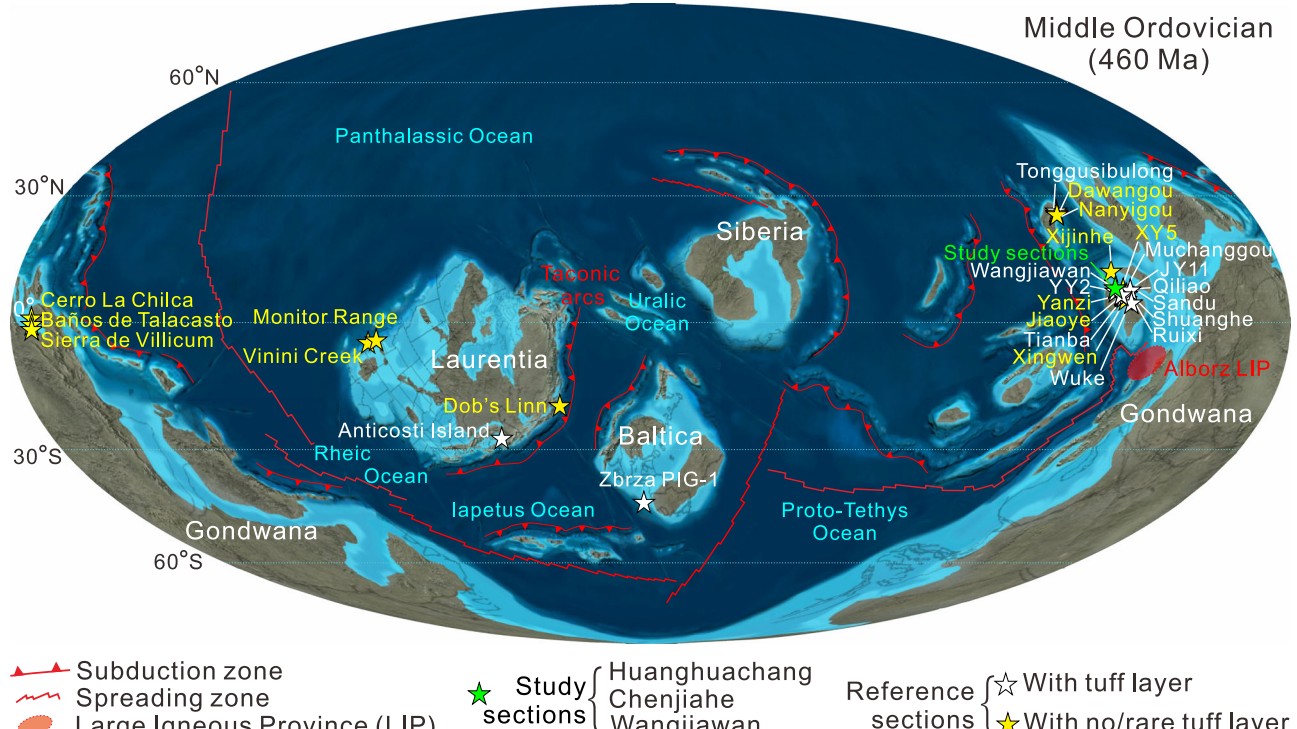

**Fig. 1 | Location of study and reference sections.** The Middle Ordovician paleogeographic map is from https://deeptimemaps.com (© 2016 Colorado Plateau Geosystems Inc). Subduction and spreading zones are based on Longman et al.[16]. Alborz large igneous provinces (LIP) in northern Iran is from Derakhshi et al.[38]. Reference sections provided Hg content, ratio of Hg content to total organic carbon content (Hg/TOC), Hg isotopes, and temporal distribution of volcanic tuff layers for the Middle Ordovician to lower Silurian interval. To consolidate their paleogeographic display, some Upper Ordovician-lower Silurian stratigraphic sections are shown on this Middle Ordovician paleogeographic map.

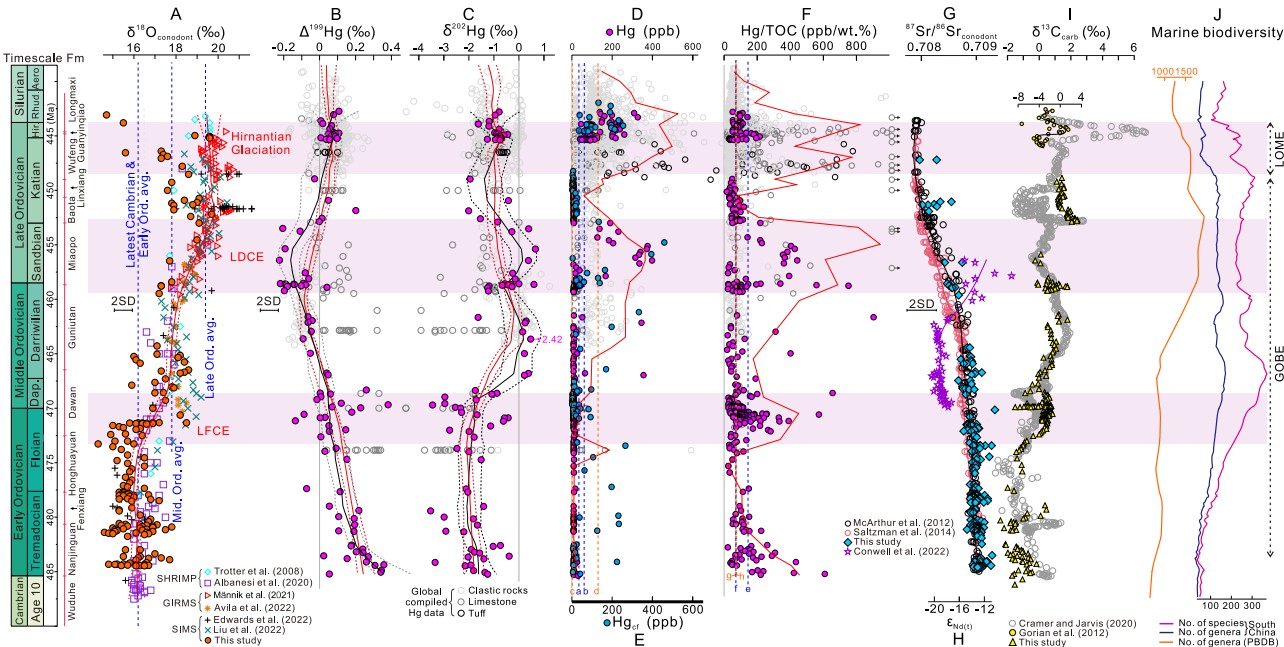

**Fig. 2 | Integrated geochemical data (this study and published) and major biological events of the Ordovician. A** Conodont oxygen isotopes ($\delta^{18}O_{conodont}$). **B**, **C** Mercury isotopes ($\Delta^{199}Hg$ and $\delta^{202}Hg$). **D** Whole rock Hg content. **E** Carbonate-free Hg content ($Hg_{cf}$). **F** Ratio of Hg content to total organic carbon content (Hg/TOC). **G** Strontium isotopes of conodont ($^{87}Sr/^{86}Sr_{conodont}$). **H** Neodymium isotopic ratio ($^{143}Nd/^{144}Nd$, as $\varepsilon_{Nd(t)}$) of bulk carbonate. **I** Bulk-carbonate carbon isotopes ($\delta^{13}C_{carb}$). **J** Marine biodiversity curves. In panels **B**–**I**, newly generated data, and published Hg data and C-isotopes at Wangjiawan section[52,70], are shown in solid circle, diamond or triangle. In **A**–**C**, LOWESS analysis of both newly generated and published data are represented by red curves (dashed and solid fits), while analyses performed specifically on newly generated data are shown as black curves (dashed and solid fits). All LOWESS analysis were performed using a fixed smoothing factor of 0.2. In **D**, **F**, the red curves represent the means of the highest five values within each bin (using 2-Myr bins at 485.4-450 Ma and 1-Myr bins at 450–438.5 Ma). In **F**, the black arrows represent values >1000 ppb/wt%. In panel A, average $\delta^{18}O_{conodont}$ values of all newly and published data for the latest Cambrian-Early Ordovician (~16.2‰) and Middle Ordovician (~17.8‰) are used as baseline values (dashed blue lines) for identification of the cooling episodes during the Late Floian cooling event (LFCE) and the Late Darriwilian cooling event (LDCE). In **D**–**F**, average Hg content of Phanerozoic carbonate rocks (line a, 34 ppb) and shale (line b, 62 ppb)[20], median Hg content of our studied carbonate rocks (line c, 2.4 ppb) and organic-rich clastic rocks (line d, 131 ppb), and average Hg/TOC ratio of Phanerozoic sedimentary rocks (line e, 144 ppb/wt%) and sedimentary rocks of TOC > 0.2% (line f, 72 ppb/wt%)[20], and median Hg/TOC ratios of our studied carbonate rocks (line g, 70 ppb/%) and organic-rich clastic rocks (line h, 76 ppb/%) are adopted as baseline values (dashed vertical lines) for jointly evaluation of Hg enrichment episodes. In **A**–**F**, $\delta^{18}O_{conodont}$ and Hg geochemical data are archived in Figshare. In **G**–**J**, published $^{87}Sr/^{86}Sr_{conodont}$ profiles are from Saltzman et al.[24] (red line) and McArthur et al.[25] (black line), $\varepsilon_{Nd(t)}$ from Conwell et al.[71], $\delta^{13}C_{carb}$ from Cramer and Jarvis[72] and Gorjan et al.[37], and biodiversity data from Deng et al.[2] and Kröger et al.[3]. The Ordovician timescale is from Goldman et al.[28]. Light pink fields represent cooling episodes during the LFCE, LDCE, and Hirnantian Glaciation. Fm = Formation; Ord. avg. = Ordovician average; Dap. = Dapingian; Hir. = Hirnantian; Rhud. = Rhuddanian; Aero. = Aeronian; PBDB = Paleobiology Database; SHRIMP = Sensitive high resolution ion microprobe; GIRMS = Gas isotope ratio mass spectrometry; SIMS = Secondary ion mass spectrometry; GOBE = Great Ordovician Biodiversification Event; LOME = Late Ordovician Mass Extinction; LFCE = Late Floian cooling event; LDCE = Late Darriwilian cooling event.

## Results and discussion

### Sample analysis and global data compilation

Carbonate carbon isotopes ($\delta^{13}C_{carb}$) exhibit major positive excursions from ~−2‰ to ~0‰ in the middle to upper Tremadocian, ~−2‰ to ~+1‰ in middle Floian to lower Dapingian, ~0‰ to ~+1‰ in the Darriwilian, and ~0‰ to ~+2‰ in the middle Sandbian to lower Katian (Fig. 2). These major excursions of $\delta^{13}C_{carb}$ are well preserved and comparable to representative global $\delta^{13}C_{carb}$ profiles (Supplementary Figs. 4-5). Together with intercalibrated biostratigraphic data, they provide a high-resolution temporal framework for the present study (Supplementary Note S1).

Our conodont in-situ oxygen isotope record ($\delta^{18}O_{conodont}$) shows a long-term secular increase from ~+15-17‰ in the Lower Ordovician Nanjinguan Formation (Fm) to ~+16-19‰ in the Middle Ordovician Dawan to Guniutan Fms, and to ~+18-20‰ in the Upper Ordovician Wufeng Fm (Supplementary Fig. 6). Diagenetic and taxon-related effects on $\delta^{18}O_{conodont}$, latitudinal changes of South China, and their impacts on SST reconstruction are discussed in Supplementary Note S2. Overall, these $\delta^{18}O_{conodont}$ data document three major cooling episodes within the Ordovician in South China (Fig. 2): (1) the late Floian cooling event (LFCE), during the *Prioniodus honghuayuanensis* to *Baltoniodus navis* zones of the late Floian to middle Dapingian

stages (upper Honghuayuan to lower Dawan Fms, ~200-215 m), (2) the late Darriwilian cooling event (LDCE), during the *Yangtzeplacognathus protoramosus* to *Amorphognathus ordovicicus* zones of the late Darriwilian to early Katian stages (upper Guniutan to lower Baota Fms, ~270-280 m), and (3) the Hirnantian Glaciation, during the *Normalograptus extraordinarius-N. ojsuensis* to *N. persculptus* zones of the late Katian to Hirnantian stages (upper Lingxiang to middle Guanyinqiao Fms, ~305–315 m).

To establish a comprehensive global SST record, we integrated the 170 newly generated $\delta^{18}O$ values with 293 conodont $\delta^{18}O$ data points from the literature representing multiple other locales (archived in Figshare, Fig. 2). In this compiled dataset, average $\delta^{18}O$ values for the latest Cambrian ( + 16.2 ± 0.3‰), and the Lower, Middle and Upper Ordovician ( + 16.2 ± 0.8‰, +17.8 ± 0.7‰ and +19.4 ± 0.9‰, respectively) were used as baseline values for evaluation of cooling trends (Fig. 3). Relative to these baseline values, intervals of rapidly rising $\delta^{18}O_{conodont}$, representing stepwise cooling events, occurred during the late Floian (LFCE) and the late Darriwilian to early Katian (LDCE) (Fig. 2). Although the Hirnantian Glaciation interval was roughly documented in the newly generated data, it was not clearly evident in the globally compiled LOWESS curve (Supplementary Note S2). To evaluate the global representativeness of the reconstructed

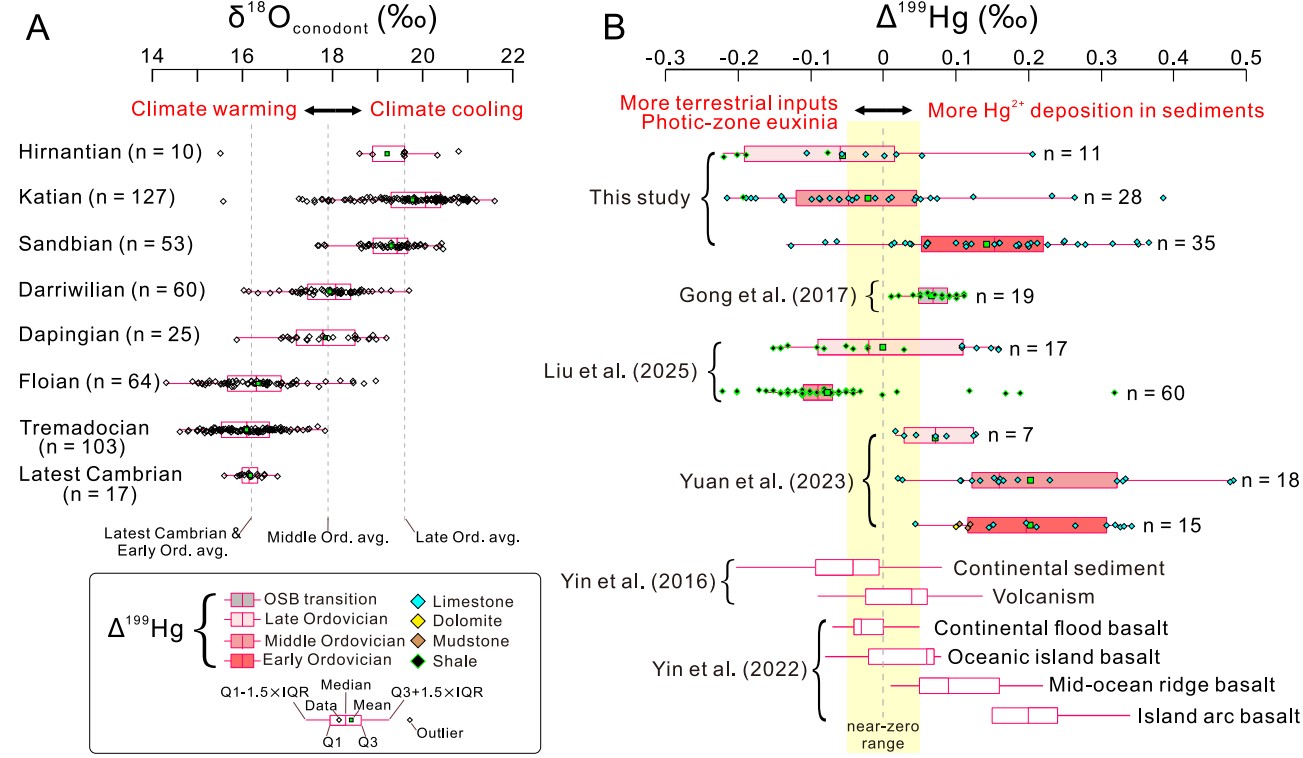

**Fig. 3 | Bar-and-whisker plots for conodont oxygen isotopes ($\delta^{18}O_{conodont}$) and mass independent fractionation of Hg isotopes ($\Delta^{199}Hg$) in Ordovician marine sedimentary rock and other major reservoirs.** Data sources: Yin et al.[23,49]; Gong et al.[52]; Liu et al.[53]; Yuan et al.[54]. Both the newly generated and compiled data shown in Fig. 1A-B are used in these plots. Abbreviations: Ord. avg. = Ordovician average; OSB = Ordovician-Silurian boundary; Q1 = First quartile; Q3 = Third quartile; IQR = Interquartile range.

SST curves and further explore the duration of the Hirnantian Glaciation, we compared them to oxygen-isotope temperature records derived from conodonts, brachiopods, and bulk carbonates (Supplementary Fig. 7). The comparison reveals similar patterns of secular variation in SSTs, particularly during the LFCE and LDCE, as well as the onset of the Hirnantian glaciation around the mid-Katian and its peak near the end of the Hirnantian. These consistent trends support our inference of three stepwise cooling events during the Ordovician.

Conodont in-situ strontium isotopes ($^{87}Sr/^{86}Sr_{conodont}$) exhibit a slow decrease from ~0.7090 in the lower Tremadocian to ~0.7085 in the middle Darriwilian, followed by a more rapid decrease to ~0.7080 in the upper Katian (Supplementary Fig. 6). Our $^{87}Sr/^{86}Sr_{conodont}$ data exhibit trends similar to a Laurentian profile[24] as well as a globally compiled dataset[25] (Fig. 2).

Mercury content ([Hg]) is mostly below Phanerozoic average values (i.e., 34 ppb and 62 ppb for carbonate- and shale-dominant successions, respectively[20], which were adopted as baseline values), interspersed with [Hg] plateaus (as high as ~200–400 ppb) above baseline values in the Sandbian and Hirnantian to basal Silurian (Supplementary Fig. 6). Carbonate-free Hg content ([Hg$_{cf}$]), which is widely used to compensate for dilution of Hg caused by increasing carbonate deposition[26], shows plateaus above the baseline value mostly in the middle Tremadocian to upper Floian, Sandbian, and Hirnantian to basal Silurian (Fig. 2). Moreover, the ratio of Hg to total organic carbon (Hg/TOC) exhibits plateaus above Phanerozoic average values (i.e., 144 ppb/wt% for all sedimentary rocks and 72 ppb/wt% for sedimentary rocks of TOC > 0.2%, which were adopted as baseline values[20]) (Fig. 2). Enrichment factors of Hg (Hg$_{EF}$) show high values above a threshold[27] of 3× (Supplementary Fig. 8) in the lower Tremadocian, upper Floian, Sandbian, and Hirnantian to basal Silurian.

Evaluation of diagenesis suggests good preservation of Hg and validity of the Hg/TOC proxy in the study sections

(Supplementary Note S3). Joint analysis using multiple Hg proxies (Supplementary Note S4) shows [Hg], [Hg$_{cf}$], Hg/TOC and Hg$_{EF}$ mostly below baseline/threshold values in the Tremadocian, indicating a lack of Hg enrichment in that stage. Pronounced Hg enrichment is manifested in the upper Floian, Sandbian, and Hirnantian to basal Silurian in the study sections.

Ratios of mercury to aluminum (Hg/Al), iron (Hg/Fe) and manganese (Hg/Mn) show roughly parallel variations (Supplementary Fig. 9), marked by general decrease trends (from ~50 to <0.1 ppb/wt%, ~50 to 0.1 ppb/wt% and 0.1 to 0.001 ppb/ppm, respectively) in the Nanjinguan to Dawan Fms, followed by increases to maxima (~100 ppb/wt%, ~300 ppb/wt% and ~3 ppb/ppm, respectively) in the upper Guniutan to Miaopo Fms. After decreases to ~0.1 ppb/wt%, ~0.3 ppb/wt% and ~0.001 ppb/ppm, respectively, in the Baota and Lingxiang Fms, these Hg ratios rebound to pre-excursion values in the Wufeng and Longmaxi Fms.

$\Delta^{199}Hg$ exhibits overall positive values with a gradually decreasing trend from ~+0.3‰ to near-zero values (~–0.05 to +0.05 ‰) in the Nanjinguan to Dawan Fms (Supplementary Fig. 6), with a few positive values (~+0.1 to +0.3 ‰) in the Honghuayuan and Dawan Fms. $\Delta^{199}Hg$ mostly exhibits negative values in the Guniutan and Miaopo Fms (~–0.2 to ~+0.05‰), and variation from ~–0.05‰ to +0.2‰ in the Baota and Lingxiang Fms. $\delta^{202}Hg$ mostly shows negative values (~–3 to 0 ‰) throughout the study sections, except for weakly positive values (~+0.2 to +0.7‰) in the Guniutan Fm (~467-462 Ma) and the Miaopo Fm (~459-454 Ma) (Supplementary Fig. 6), and it is negatively correlated with $\Delta^{199}Hg$ (Supplementary Fig. 10). Diagenesis evaluations suggest good preservation of both $\Delta^{199}Hg$ and $\delta^{202}Hg$ signals in the study sections (Supplementary Note S3).

To accurately identify Hg anomalies within the study sections and delineate globally representative enrichment intervals, we integrated our newly generated dataset with published Hg geochemical data from

22 globally distributed Ordovician sections/cores in South China, Tarim, Baltica, Laurentia and Gondwana (Fig. 1), including 1433 Hg content ([Hg]), 1361 Hg/TOC ratios, 397 $\Delta^{199}$Hg values, and 391 $\delta^{199}$Hg values (archived in Figshare, Supplementary Fig. 8). These data derive from clastic rocks (mainly shales, 75%), limestones (24%), and volcanic ash layers (1%), of mainly Late Ordovician age. The newly generated and published Hg content and isotope data are in good agreement throughout the Ordovician (Supplementary Note S5). Multiple baseline parameters of [Hg], [$Hg_{cf}$], Hg/TOC and $Hg_{EF}$ were further adopted to accurately identify Hg anomalies in the compiled dataset (Supplementary Note S4). Using a unified timescale[28], the compiled datasets confirmed Hg enrichment intervals as revealed by our newly generated data, jointly depicting plateau values of [$Hg_{cf}$], Hg/TOC and $Hg_{EF}$ and, thus, abnormal Hg enrichments during the late Floian, Sandbian, middle Katian, and late Katian to early Silurian (Fig. 2).

LOWESS-smoothed $\Delta^{199}$Hg and $\delta^{202}$Hg trends based on our study sections alone closely mirror that of the global composite trend based on both new and published data (Supplementary Note S5) (Fig. 2). Therefore, the identified $\Delta^{199}$Hg and $\delta^{202}$Hg signals through the Ordovician in the study sections represent a global pattern. Accordingly, we subdivided the compiled Ordovician $\Delta^{199}$Hg profile into four intervals, including (1) a decreasing trend of $\Delta^{199}$Hg values during the early Tremadocian to middle Floian (~485-473 Ma), (2) a dominance of near-zero $\Delta^{199}$Hg values during the late Floian, followed by a decreasing trend from positive to near-zero values during the earliest to early Dapingian (i.e., the LFCE) (~473-468 Ma), (3) a gradual decreasing trend to slightly negative values (~−0.05 to −0.1 ‰) through the Darriwilian (~468-459 Ma) until more negative $\Delta^{199}$Hg values (~−0.3 to −0.2 ‰) by the middle Sandbian, followed by a rebound to near-zero values in the late Sandbian (i.e., the LDCE) (~459-453 Ma), and (4) mostly near-zero $\Delta^{199}$Hg values followed by both near-zero and slightly positive values (~+0.05 to +0.1 ‰) during the early Katian to Hirnantian (~453-443 Ma).

## Volcanism through the Ordovician Period

Volcanism, along with other factors, e.g., submarine hydrothermal activity, seawater redox conditions, terrestrial fluxes, and sediment accumulation rates, may collectively influence temporal fluctuations in sedimentary [Hg][21,22]. In seawater, Hg complexes with organic matter to form methylmercury, and it reacts with sulfide to form mercuric sulfide (HgS), yielding compounds that commonly serving as major repositories of Hg[20]. However, mercury can also adsorb onto the surfaces of phosphate, clay minerals, and Fe-Mn oxides[20]. The concentration of phosphorus ([P]) can be utilized as an indicator of the relative abundance of phosphate minerals, and the concentration of aluminum ([Al]) can reflect the relative content of clay minerals such as kaolinite and illite, whereas the concentrations of iron ([Fe]) and manganese ([Mn]) indicate the presence of Fe-Mn oxides. In the study units, [Hg] exhibits pronounced positive covariation with TOC in the Chenjiahe ($r = +0.77$, $n = 50$, $p < 0.001$) and Wangjiawan sections ($r = +0.49$, $n = 52$, $p < 0.001$), but no significant covariation with [Al], [Mn], [Fe], or TS (Supplementary Fig. 11), implying that organic matter serves as the principal host of mercury. In comparison, [Hg] shows weak positive covariation with TOC, [Al] and [Fe] ($r = +0.35$, $n = 266$, $p < 0.001$; $r = +0.41$, $n = 82$, $p < 0.05$; $r = +0.35$, $n = 83$, $p < 0.05$, respectively), whereas no significant covariation with [Mn] and TS in the Huanghuachang section, suggesting the presence of Hg in multiple host phases (organic matter, clay minerals and Fe-oxides).

Sources of sedimentary Hg other than volcanic activity are possible. Hydrothermal activity generally results in lower $^{87}$Sr/$^{86}$Sr signals for seawater and sedimentary rock (i.e., closer to the mantle end-member, ~0.7035)[25], and, therefore, conodont $^{87}$Sr/$^{86}$Sr ratio may proxy for hydrothermal activity. Previous studies have suggested that a majority of Sr in the studied conodont specimens (avg. ~13,000 ppm) was sourced from diagenetic fluids instead of bulk carbonate (avg. ~300 ppm)[29], which means that large quantities of hydrothermally

sourced Sr in contemporaneous seawater should be detectable by analysis of conodonts. In the study sections, the range of secular variation in $^{87}$Sr/$^{86}$Sr$_{conodont}$ (~0.7080 to 0.7090) is consistent with a primary marine Sr isotope signal (Fig. 2)[24,25], providing no evidence of hydrothermal influence.

Seawater redox conditions commonly play a pivotal role in sedimentary Hg accumulation, mostly through modulation of microbial activity and the interplay of the C-S biogeochemical cycles, thus determining the speciation and concentration of mercury within the aquatic environment[30,31]. Therefore, redox oscillations can have a profound effect on Hg diffusion at the sediment-water interface, directly influencing its enrichment in sediment. To evaluate marine redox variation through the Ordovician, we used molybdenum and uranium-enrichment factors ($Mo_{EF}$ and $U_{EF}$), $C_{org}$/P ratios, and the cerium anomaly of the carbonate fraction ($Ce/Ce^*_{carb}$), for either clastic or carbonate successions (Supplementary Fig. 9; Supplementary Note S6). Overall, [Hg] shows no significant correlation to $Mo_{EF}$, $U_{EF}$, $C_{org}$/P and $Ce/Ce^*_{carb}$, and the stratigraphic distribution of [Hg] peaks are not always within intervals of more reducing watermass conditions (i.e., lower $Ce/Ce^*_{carb}$, and higher $Mo_{EF}$, $U_{EF}$ and $C_{org}$/P) (Supplementary Figs. 9 and 11). However, a few high-[Hg] samples (e.g., limestone lenses in Miaopo Fm shale) correspond to more reducing conditions (i.e., lower $Ce/Ce^*_{carb}$), implying that reducing conditions promoted Hg accumulation in limited intervals of the study sections, probably through elevated net burial rates of organic matter and sulfide[20].

Terrestrial fluxes serve as a major source of Hg to the ocean. The overall decrease in calcium carbonate content ([$CaCO_3$]) and increase in [Al] observed in the study sections suggest local increases in the content of non-carbonate components (e.g., terrestrial materials) during the Ordovician (Supplementary Fig. 9). However, [Hg] exhibits no correlation to [$CaCO_3$] or [Al] in the study sections (Supplementary Fig. 11), suggesting that the overall increasing trend of [Hg] through the Ordovician cannot be ascribed to Hg absorption by specific lithologic components.

Sedimentation rate can exert a significant influence on Hg uptake in sedimentary rocks by modulating their physical, chemical, and biogeochemical attributes. A low sedimentation rate facilitates the uptake of hydrogenous species (such as Hg complexes) by the sediment. In the present study, however, average linear sedimentation rate (LSR) shows no correlation with [Hg] at the substage level of resolution (Supplementary Fig. 12). Although condensed intervals with low sedimentation rates generally favor Hg accumulation, similar average [Hg] values (~2-3 ppb) are observed in the rapidly deposited Tremadocian (~24 m/Myr), the intermediate-rate Dapingian (~12 m/Myr), and the slowly deposited Floian (~4 m/Myr) stages at Huanghuachang (Supplementary Figs. 9, 12), suggesting that variation in sedimentation rates was not the dominant cause of observed [Hg] peaks.

In sedimentary successions, the Hg/TOC ratio is usually relatively stable around a background value[20]. Volcanically sourced Hg may be released directly during eruptions or emitted through magmatic heating of organic-rich sediment (e.g., coal beds)[32], or directly to seawater through ocean-crustal hydrothermal activity. The erupted Hg is predominantly in a gaseous form and is subject to global atmospheric transport over a brief timeframe (~0.5 to 2 yr), during which it partakes in the global ocean's biogeochemical cycles, thus leading to anomalously high Hg/TOC ratios (i.e., relative to the baseline value). Sedimentary rock Hg/TOC records have been widely used to evaluate secular variation in volcanic activity at a regional or global scale[20]. While Hg/TOC normalization is suitable for tracing volcanic Hg anomalies in the Chenjiahe and Wangjiawan sections, it is inappropriate for the Huanghuachang section, in which organic matter, clay minerals and Fe-oxides all serve as Hg hosts, necessitating the use of Hg/Al and Hg/Fe as complementary proxies[32]. In both study sections as well as other successions of the same age globally, plateau values of [$Hg_{cf}$], Hg/TOC, and $Hg_{EF}$ are above baseline levels (Fig. 2 and

Supplementary Fig. 12), and the corresponding plateau/peak values in both Hg/Al and Hg/Fe ratios (Supplementary Fig. 9) indicate major volcanic Hg inputs during the late Floian, Sandbian, middle Katian, and late Katian to early Silurian (Supplementary Note S5).

Apart from Hg anomalies, volcanic tuff layers constitute the most overt manifestation of regional volcanism. The multiple tuff layers present in the Middle and Upper Ordovician of the study sections, along with widespread (~0.5 Mkm²) tuffs of trachyandesitic to rhyodacitic composition across the Yangtze Platform (Supplementary Figs. 1-2), especially in Hirnantian successions, were likely derived from a continental volcanic arc associated with arc magmatism along the margin of the South China Craton during the convergence of the Yangtze and Cathaysia blocks[33] (Supplementary Note S7). A global compilation of the spatiotemporal distribution of volcanic tuff layers (Supplementary Note S1) shows sparse occurrences in the Lower Ordovician, with increasing frequency in the Middle and Upper Ordovician of the USA, southern China, Europe, Argentina and Tarim Basin, China (Supplementary Fig. 13)[34]. These global occurrences have been attributed to continental arc volcanism linked to closure of the Iapetus Ocean[35]. Furthermore, these volcanic tuff layers are typically associated with elevated [Hg] and Hg/TOC ratios exceeding baseline values[36], near-zero $\Delta^{199}$Hg values[37], and host formations that display overall Hg anomalies (Supplementary Fig. 8). Although continental arc volcanism (persistently active during oceanic crust subduction) contributed Hg, the spatiotemporal distribution of volcanic tuff layers remains inconsistent with globally documented multiphase Hg enrichment intervals in Ordovician sedimentary successions. Therefore, continental arc volcanism was not the primary driver of the multiphase Hg enrichment intervals in the Ordovician.

Only a few continental large igneous provinces (LIP) of Ordovician age are currently known. The recently identified Alborz LIP in northern Iran[38] has an estimated basalt volume of ~0.5 Mkm³. This LIP originated from the initial rifting of Cimmerian terranes from the northern Gondwanan margin to form the Paleotethys Ocean. Accompanied by equatorward drift, its paleolatitude shifted from ~30°S in the Early Ordovician to ~10°S by the Late Ordovician. The Alborz LIP has been broadly dated to the Middle to Late Ordovician (~469 to 451 Ma), but the existing radiometric ages are insufficient to narrow its timing or to define a pattern of episodic activity[38] (Fig. 2). Additionally, recent studies have identified a major tectono-magmatic event in the Proto-Qiangtang Ocean within peri-Gondwanan terranes (adjacent to South China) that extended from the late Cambrian to the late Middle Ordovician (~510–460 Ma) based on modest age constraints from zircon U-Pb dating[39]. This event (Pinghe) is regarded as a potential continental LIP with a volume of ~2.5 Mkm³, although its classification remains insecure given that its composition of dominant S-type and subordinate A-type granites is also consistent with subduction-zone magmatism. Other as yet-unidentified Ordovician LIPs may have existed. Moreover, high rates of plate motion and oceanic lithosphere formation during the Ordovician, as evidenced by abundant ophiolites[40] and extensive basaltic lavas (e.g., in the British Isles, northern Iran and West Junggar) (e.g., Parnell et al.[41]), exceed those of other periods of the Paleozoic Era[42] (Supplementary Fig. 14). These factors likely contributed to contemporaneous global peaks in the production rates of volcanogenic massive sulfide deposits, sulfidic shales and ironstones[39]. Volcanism related to continental LIPs as well as subduction of oceanic lithosphere may have led to globally elevated basalt weathering fluxes, especially in (sub)tropical regions[14], resulting in a prolonged negative shift of $^{87}$Sr/$^{86}$Sr (by 0.0008) during the Darriwilian to Sandbian, followed by stabilization in the Katian[43] (Fig. 2). LIP volcanism is commonly the primary driver of anomalous Hg enrichment in marine sediments during major geological events such as mass extinctions[20,27]. The current clues regarding Ordovician LIPs suggest that such volcanism may have contributed to the global sedimentary Hg enrichments, although temporal links between LIP activity and Hg enrichment are supported only for limited intervals (~469 Ma, early Dapingian)[38] (Fig. 2).

Potentially significant is the inference of a superplume event during the Middle to Late Ordovician[17,44,45]. Although not well documented, it is supported by multiple lines of indirect evidence including carbon cycle and weathering modeling[46,47], a secular decrease of seawater $^{87}$Sr/$^{86}$Sr[43], and a lack of geomagnetic reversals (i.e., a polarity superchron)[45] (Supplementary Fig. 14). The hypothetical Ordovician superplume is comparable to the Cretaceous superplume in coinciding with the continental-rift/drift stage of a Wilson cycle. The Cretaceous superplume is known to have been associated with pulsed oceanic crustal production, formation of large-scale oceanic plateaus (e.g., Ontong Java), intensified magmato-tectonic activity, reduced geomagnetic field reversal frequency (i.e., the Cretaceous Normal Polarity Superchron), and increased hydrothermalism, as revealed from various sedimentary geochemical records (e.g., $^{87}$Sr/$^{86}$Sr), and similar features characterize the interval of the Ordovician superplume.

Processes not directly associated with volcanism may also lead to MIF and MDF of mercury isotopes. Hg emanating from remote volcanoes experiences extended atmospheric transport, potentially leading to isotopic fractionations due to atmospheric redox processes (e.g., photoreduction). These processes produce positive Hg-MIF compositions (e.g., $\Delta^{199}$Hg) and negative Hg-MDF compositions (e.g., $\delta^{202}$Hg) in atmospheric $Hg^{2+}$, as well as in sediments dominated by atmospheric $Hg^{2+}$ accumulation[30]. Furthermore, photochemical reduction of aqueous $Hg^{2+}$ to $Hg^0$ in the photic zone of euxinic waters, as observed during the end-Permian mass extinction[30], can give negative $\Delta^{199}$Hg and positive $\delta^{202}$Hg in sedimentary rock[31,48]. Additionally, recent studies have shown that mid-ocean ridge basalt (MORB) and island arc basalt (IAB) exhibit positive $\Delta^{199}$Hg values (-0.1 to >0.3‰), whereas oceanic island basalt (OIB) and continental flood basalt (CFB) have near-zero $\Delta^{199}$Hg values[49] (Fig. 3). Therefore, weathering of basalts generally causes Hg isotopes in sedimentary rocks to shift toward values associated with the basaltic endmember (i.e., near-zero to positive $\Delta^{199}$Hg and negative $\delta^{202}$Hg values[50]).

Our integrated dataset of Hg concentration and isotopic data reveals three major episodes (e.g., the LFCE, LDCE and middle Katian to Hirnantian) of intense volcanism during the Ordovician (Fig. 4). During the LFCE (~473–468 Ma, Early-Middle Ordovician transition), volcanic Hg from both continental arcs and, possibly, continental LIPs dominated Hg fluxes and its enrichment in sedimentary rocks. Volcanism intensified during the first half of this interval (~473–470 Ma), which was marked by relatively constant near-zero to slight positive $\Delta^{199}$Hg values, negative $\delta^{202}$Hg values, and positive excursions of [Hg$_{cf}$], Hg/TOC and Hg$_{EF}$ above the baseline/threshold value (Fig. 2). The second half (~470–468 Ma) was characterized by weakened volcanic activity and a rise in the proportion of atmospheric $Hg^{2+}$, marked by low [Hg$_{cf}$], Hg/TOC and Hg$_{EF}$, and increased $\Delta^{199}$Hg values, and which may have been related to a rise in atmospheric oxygen concentrations. The weak volcanism perpetuated into the late Dapingian to Darriwilian (~468–459 Ma), which led to only sporadic Hg enrichments, although higher primary productivity and intensive global oceanic anoxia[51] may also result in greater Hg drawdown[20].

During the LDCE (~459–453 Ma), episodically intensified continental LIP and continuously active continental arc volcanism-controlled Hg enrichments in sedimentary rock (n.b., almost all [Hg$_{cf}$], Hg/TOC and Hg$_{EF}$ values are above the baseline/threshold). Meanwhile, Hg isotopic fractionation recorded in sediment rocks no longer controlled by volcanism (i.e., near-zero to slight positive $\Delta^{199}$Hg values), but rather overlaid by redox-controlled signals (i.e., negative $\Delta^{199}$Hg, positive $\delta^{202}$Hg) under reducing water masse (i.e., photic-zone euxinia).

Following an interval of limited volcanism at ~453–449 Ma, continental LIP and/or continental arc volcanism intensified again,

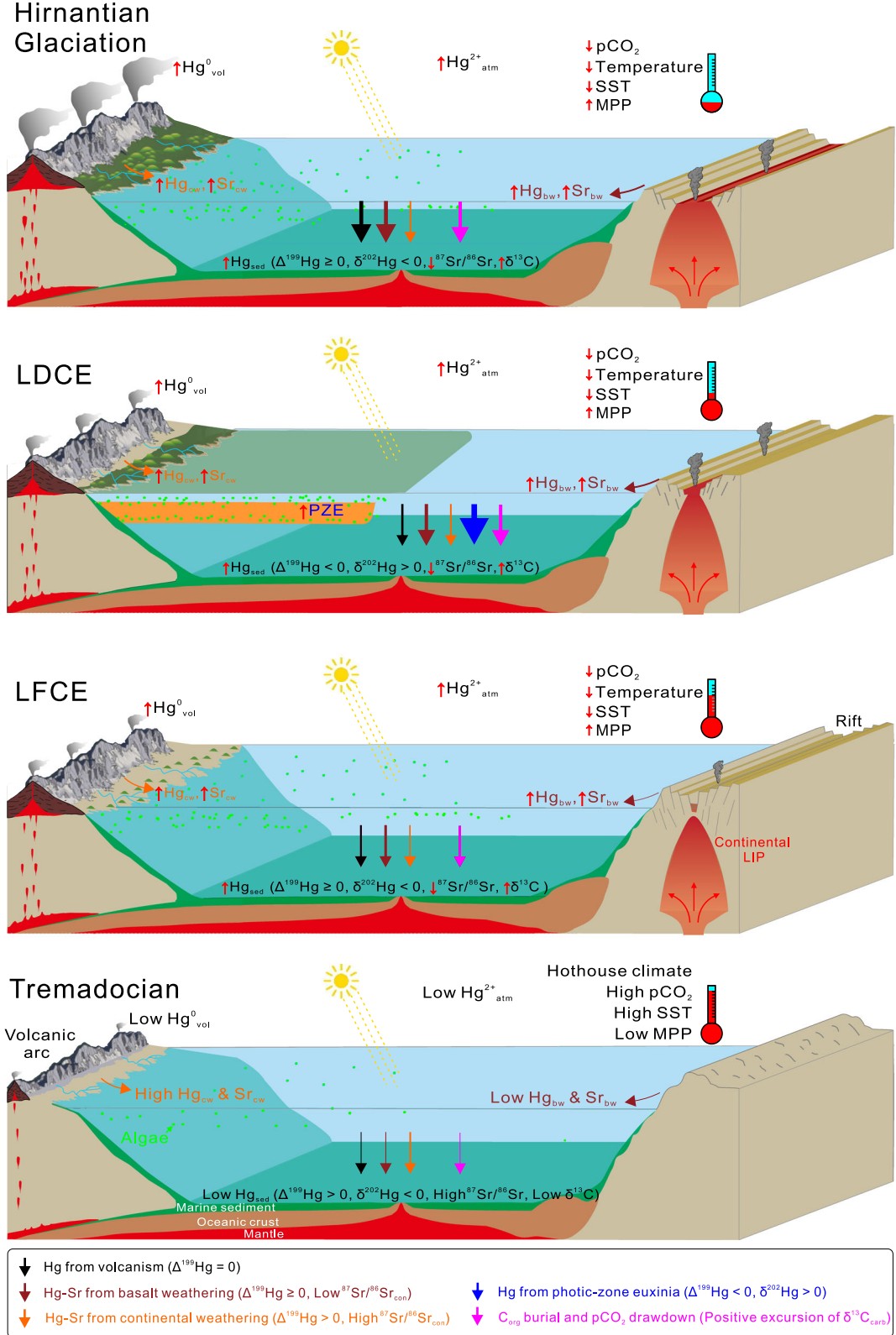

**Fig. 4 | Schematic scenarios of volcanic-climate impacts and marine sedimentary geochemical responses through the Ordovician.** Upward red arrow indicates elevated influxes/intensities or increased values, while downward red arrow indicates the opposite pattern. vol = Volcanism; con = Continent; sed = Sediment; atm = Atmosphere; cw = Continental weathering; bw = Basalt weathering; $C_{org}$ = Organic matter; MPP = Marine primary productivity; SST = Sea-surface temperature; PZE = Photic-zone euxinia; LFCE = Late Floian cooling event; LDCE = Late Darriwilian cooling event.

especially during the middle Katian (~448–447 Ma) and Hirnantian (~445–444 Ma), which drove Hg fractionation and its enrichment in sedimentary rock (mostly near-zero $\Delta^{199}$Hg values and peak [Hg$_{cf}$], Hg/TOC and Hg$_{EF}$ values above the baseline/threshold[52]). The two pulses of volcanism are interrupted during the ~447–445 Ma, resulting in drops of [Hg$_{cf}$], Hg/TOC and Hg$_{EF}$ values mostly below the baseline/threshold (Fig. 4).

## Pulsed episodes of continental weathering and marine photic-zone euxinia

The integrated $\Delta^{199}$Hg, $\delta^{202}$Hg and $^{87}$Sr/$^{86}$Sr$_{conodont}$ indicates episodes of enhanced weathering of volcanic rocks (basalt) (Supplementary Note S8) and marine photic-zone euxinia (PZE) during the Ordovician (Fig. 4). At ~485–473 Ma, an interval during which regional continental arc volcanism was weak, the initial phase of the Taconic Orogeny promoted weathering of mafic rocks (e.g., ophiolite and tropical arc detritus)[14] along with Hg inputs in low-latitude regions (Fig. 4). This event likely dominated Hg isotopic fractionation in the sediment, as revealed by relatively high $^{87}$Sr/$^{86}$Sr$_{conodont}$ and $\Delta^{199}$Hg values and scattered [Hg$_{cf}$] peaks, whereas Hg/TOC and Hg$_{EF}$ mostly fell below baseline/threshold values (Fig. 2 and Supplementary Fig. 8). During the LFCE (~473–468 Ma), relatively stable $^{87}$Sr/$^{86}$Sr$_{conodont}$ values (decreasing by only 0.0002)[24] indicate an overall balance in Sr inputs from mantle/mafic crustal sources (e.g., basalt) and felsic continental weathering (e.g., granite). Over the entire ~485–468 Ma interval, the prolonged decrease in $\Delta^{199}$Hg (from >0.3‰ to ~0‰) and paired negative $\delta^{202}$Hg values with minor variation (around −2‰) cannot be explained by Hg photoreduction in photic-zone euxinic/anoxic watermasses[30] but, rather, dominated by continental weathering.

At ~468–459 Ma, PZE dominated marine Hg fractionation (i.e., negative $\Delta^{199}$Hg, positive $\delta^{202}$Hg) because of enhanced nutrient inputs during volcanic rock weathering, e.g., in tropical arc region (as recorded by decreased $^{87}$Sr/$^{86}$Sr$_{conodont}$ and increased $\epsilon_{Nd(t)}$ values; Fig. 2). During the LDCE (~459–453 Ma), prolonged PZE dominated marine Hg fractionation (i.e., strongly negative $\Delta^{199}$Hg, shift to positive $\delta^{202}$Hg), which was caused mainly by elevated nutrient influx due to intensified weathering of volcanic rocks (especially in tropical areas[14]), as evidenced by sharp decreases in $^{87}$Sr/$^{86}$Sr$_{conodont}$ and increases in $\epsilon_{Nd(t)}$ (Fig. 2). Currently, Middle Ordovician PZE has been reported from the Tarim Basin and other localities in South China[53], but Hg isotope signatures are contrasting between nearshore sections with PZE such as Nangyigou ($\Delta^{199}$Hg: +0.14 ± 0.08‰; $\delta^{202}$Hg: −2.21 ± 0.98‰) and offshore sections such as Xijinhe, Dawangou, Huanghuachang (negative $\Delta^{199}$Hg and positive $\delta^{202}$Hg values)[54], likely due to non-uniform Hg cycling processes in different reservoirs[30]. At ~453–449 Ma, PZE weakened and continental weathering dominated marine Hg fluxes and isotopic fractionation in sedimentary rocks, as indicated by increases of $\Delta^{199}$Hg to mostly near-zero values, and declining $\delta^{202}$Hg to negative values (Fig. 2 and Supplementary Fig. 8).

## Global volcanism and basalt weathering drivers the LFCE and LDCE

Long-term Ordovician cooling has been attributed to various mechanisms, including orogenic uplift (i.e., Taconic Orogeny) and weathering of silicate rocks[55], or a shift of tectonic plates into more humid regions and subsequent intensified weathering of fresh volcanic rocks[56]. The general problem with such explanations is that there have been many orogenies, volcanic events, and shifts in tectonic plates of similar or greater magnitude during the Phanerozoic that have not led to markedly enhanced weathering. In the present study, global compilations of volcanic tuffs indicate an overall intensification of continental arc volcanism due to subduction of oceanic lithosphere throughout the Ordovician (Supplementary Fig. 13). Although continental arc volcanism can potentially draw down CO$_2$ levels through enhanced chemical weathering, our global compilation of volcanic

tuffs shows no increase in such volcanism during the three stepwise cooling phases of the Ordovician. Significantly, continental arc volcanism is more commonly associated with warming events in Earth history, through reworking of the crustal carbonate reservoir and degassing of CO$_2$, rather than with cooling events (Supplementary Note S7).

Alternatively, episodic intense continental LIP volcanism followed by prolonged enhanced weathering of basalt represents a more likely explanation for the multistage character of a long-term cooling event, e.g., during major geological events[57] and in the Ordovician[14], although such volcanism may have caused transient warming by releasing CO$_2$, acting as immediate effect of active eruptions[32]. In addition, the linkage of continental LIP volcanism to the Ordovician climatic cooling is evidenced by the concurrent, episodically intensified low-latitude Alborz LIP[38] (Fig. 2). However, owing to the limited number of documented continental LIPs and the lack of high-resolution geochronological constraints, the timing of episodic continental LIP volcanism and its relationship with the three stepwise cooling events remain to be further verified. In any case, chemical weathering of basalt could sequester atmospheric CO$_2$ during the Ordovician[56], predominantly through secular silicate minerals reacting with CO$_2$ to produce carbonates.

Globally intensified continental LIP volcanism and weathering of volcanic rocks during the Ordovician is likely to have been the trigger for climatic cooling. This relationship is implied by concurrent positive excursions of $\delta^{18}$O$_{conodont}$, and plateau [Hg$_{cf}$], Hg/TOC and Hg$_{EF}$ above baseline/threshold during the LFCE, LDCE and Hirnantian Glaciation, as well as plateau $\delta^{18}$O$_{conodont}$ and sporadic Hg enrichments between the three major cooling events (Fig. 2 and Supplementary Fig. 8). Furthermore, the negative shift in $\Delta^{199}$Hg to near-zero values and absence of Hg enrichment in sedimentary rocks indicate weakened oceanic lithospheric subduction and continental LIP quiescence during the Early Ordovician, concurrent with a relatively warmer climate. These relationships suggest that the Ordovician climatic cooling (e.g., LFCE and LDCE) was primarily a volcanically driven, stepwise cooling event spread out over a ~40-Myr-long interval that culminated in the Hirnantian Glaciation. The inferred temporal concurrence of volcanism and the Ordovician climatic cooling have previously been ascribed to weathering of volcanic rocks[14,46,55].

As a pivotal adjunct, the volcanic cooling effect was further amplified by enhanced marine primary productivity and organic carbon burial (e.g., peak TOC values in the LDCE interval, Supplementary Fig. 9), driven by elevated nutrient fluxes (e.g., phosphorus) from the weathering of terrestrial volcanic rocks[16,43]. This is supported by the long-term Ordovician climatic cooling and concurrent of positive excursions of $\delta^{13}$C$_{carb}$ (Fig. 2). These considerations establish connections between the GOBE, the Ordovician climatic cooling, and contemporaneous volcanism (e.g., Albanesi and Barnes[58]; Miller et al.[59]).

As important supplements to volcanism and basalt weathering, several other processes may have collectively accelerated continental volcanic rock weathering and pCO$_2$ drawdown, hastening the development of the Late Ordovician icehouse. The Ordovician Period was an active phase of continental drift and rift formation. The entire ensemble of continents moved ~1500–2000 km northwards at a moderate rate (~5.0 cm per year), along with counterclockwise rotation (e.g., by 30° for the Laurentia and 45° for the Baltica), apart from Gondwana, which moved the slowest (at a speed of ~2.5 cm per year) and remained situated over the South Pole[60]. Meanwhile, continental rifting persisted throughout the Ordovician until the onset of collision in the Late Ordovician, causing reduced volume of ocean basins, progressively rose in sea levels, and a maximum sea level during the Late Ordovician (excluded a precipitous sea level fall during Hirnantian Glaciation) which is comparable to that in the mid-Cretaceous[61]. Active plate movement promotes changes in the land surface topography, leading to the formation of rifted margins (as well as island arcs, zones

of plate collision, etc.), thus generations of more weatherable volcanic rocks. The abundance of passive continental margin reflects the scale of continental breakup and rifted margins, which are consistently high during the Cambrian greenhouse and Ordovician icehouse periods (and plateau levels of the Early Paleozoic)[62], indicating that the changes of rifted margins probably not the cause of the Ordovician cooling.

In addition, our present study shows major global volcanism and the onset of Ordovician climatic cooling during the middle Floian (~ 475Ma), which is ~5 Myr earlier than the earliest embryophytes in the Dapingian (~470 Ma) (Supplementary Note S9). Therefore, the delayed evolution of land plants, as well as small biomass of early land plants, and/or incomplete fossil record do not support terrestrialization as a trigger of the Ordovician cooling event. With the expansion of land plants during the Middle to Late Ordovician[63], they would have played an increasingly important role in the climatic cooling process[15]. However, the exact causal relationship between land plant evolution and Ordovician cooling remains unclear due to a lack of definitive fossil evidence, which may be addressed in the future through more terrestrial plant fossil records and refined age constraints.

Overall, secular climatic cooling from the LFCE to the Hirnantian Glaciation was probably driven by continental LIP volcanism and basalt weathering, and may be further amplified by elevation of marine productivity (thus PZE). Concurrent with the three-phase climatic cooling, Hg enrichments in sedimentary rock were mainly driven by episodically intensified volcanism related to continental LIP under background of prolonged continental arc volcanism. Hg fractionation was largely driven by continental weathering (especially the basalt) and the following oceanic PZE in the ~485–473 Ma and ~468–448 Ma interval, and by volcanism in the LFCE and Hirnantian Glaciation.

## Methods

### Geological background and sample preparation

The Huanghuachang, Chenjiahe and Wangjiawan sections are situated in the Yiling District, Yichang City, South China (Supplementary Fig. 1). The paleogeography and lithostratigraphy of the study sections have been fully documented in prior studies[29]. Additional bio- and $\delta^{13}C_{carb}$ chemo-stratigraphy (Supplementary Figs. 3 and 5), and their inter-calibration for the Ordovician time framework are discussed detailly in Supplementary Note S1. Here, we provide a concise summary focused on key stratigraphic features essential to this analysis. Paleogeographically, the study area was located on the north-central Yangtze Platform, which accumulated shallow-marine carbonate sediment that graded into argillaceous sands in inner-shelf settings to the northwest and slope facies to the southeast[64]. The study area exposure the Wuduhe Fm of the Upper Cambrian, and overlying 11 formations/beds (from base to top, the Nanjinguan, Fenxiang, Honghuayuan, Dawan, Guniutan, Miaopo, Baota, Lingxiang and Wufeng formations, Guanyinqiao Bed, and Longmaxi Fm) ranging continuously in age from the earliest to latest Ordovician and Ordovician-Silurian transition[64] (Supplementary Figs. 2-3). Limestone, bioclastic limestone and muddy limestone are dominant in the study successions, except for thin (few meters) black shale intervals with limestone lenses in the Miaopo Fm, and black shales in the Wufeng and Longmaxi Fms. Lithological compositions of the successions were fully described in Zhang et al.[29]. Besides, multiple volcanic tuff depositions yielded, including a ~ 4 cm thick tuff layer in the upper Dawan Fm and a ~ 10 cm thick tuff layer in the middle Miaopo Fm at Huanghuachang, and at least five tuff layers of ~1 to 4 cm thick in the upper Wufeng Fm at Wangjiawan (Supplementary Fig. 2). Overall, the study successions record a gradual shift from carbonate platform facies in the Lower Ordovician to neritic facies in the Middle Ordovician and deep basinal facies in the Upper Ordovician, reflecting a long-term sea-level rise before the Hirnantian glacio-eustatic fall at the end of the Ordovician[29].

The bulk carbonate rocks and few shales were collected from the study sections. Weathered surfaces and diagenetic veins in bulk rocks were removed, and the remaining sample was cleaned, air-dried and crushed. An aliquot of each sample was powdered using a rock mill to <200 mesh for bulk-rock geochemical analyses. To extract conodont specimens, lightly crushed samples were dissolved in 10% acetic acid for days, after which conodont elements (from a few to dozens in absolute frequency) were recovered from the insoluble residue using a binocular microscope. The color alteration index (CAI) of the study specimens ranges from 1 to 3, indicating probable preservation of primary environmental signals in the conodont apatite (Supplementary Note S2). Each conodont specimen was identified, including *Ansella* sp., *Drepanodus* sp., *Paroistodus* sp., *Periodon* sp., *Oneotodus* sp., *Scolopodus* sp., *Drepanoistodus* sp., *Baltoniodus* sp., *Tripodus* sp., *Serratognathus* sp., and *Triangulodus* sp. The samples used in this study represent the same suite that was used in Zhang et al.[9,29].

Conodont samples were embedded in resin, followed by grinding and polishing to produce a flat surface suitable for in-situ oxygen isotope analysis. The densest albid crown was targeted for in-situ O isotope analyses in this study to minimize tissue-related effect, reduce biases in reconstructed temperature curve, and facilitate cross-case comparisons (Supplementary Note S2). Other batch of conodonts were affixed to double-sided adhesive tape on a silica glass for in-situ Sr-isotopic analysis. Albid crown is also thought to yield the most faithful record of (near-)primary seawater Sr isotope signals[24], and it was targeted for the in-situ Sr isotope analyses in this study.

### In-situ conodont oxygen isotopes and paleotemperature

In-situ oxygen isotope measurements were made using a Cameca IMS 1280 secondary ion mass spectrometry (SIMS) at the Institute of Geology and Geophysics, Chinese Academy of Sciences. Oxygen isotopes were measured using the multi-collection mode on two off-axis Faraday cups. The intensity of $^{16}O$ was typically $1 \times 10^9$ cps. The nuclear magnetic resonance probe was used to control stability of the magnetic field. A single analysis took ~5 min consisting of pre-sputtering (~120 s), automatic beam centering (~60 s) and integration of oxygen isotopes (20 cycles × 4 s, total 80 s). SIMS measurements were performed using identical laser parameters (20 μm spot diameter, 80 s ablation duration) to minimize systematic biases.

The instrumental mass fractionation factor (IMF) was corrected using the Durango apatite standard. Measured $^{18}O/^{16}O$ ratios were normalized to Vienna Standard Mean Ocean Water compositions (V-SMOW, $^{18}O/^{16}O = 0.0020052$), and then corrected for the instrumental mass fractionation factor (IMF) as follows:

$$\left( \delta^{18}O \right)_M = \left( \frac{(^{18}O/^{16}O)_M}{0.0020052} - 1 \right) \times 1000_{(‰)} \tag{1}$$

$$IMF = \left( \delta^{18}O \right)_{M(standard)} - (\delta^{18}O)_{VSMOW} \tag{2}$$

$$(\delta^{18}O)_{sample} = (\delta^{18}O)_M - IMF \tag{3}$$

Replicate analyses of the Durango apatite standard (which was analyzed after every five sample measurements) yielded an average value of +9.40 ± 0.16 ‰ (2 SD; $n = 78$), which is indistinguishable within analytical uncertainty from the reported value of +9.4 ± 0.3 ‰ (2 SD)[10]. An average of ~5 measurements of $\delta^{18}O$ were obtained for 1 to 3 specimen(s) from each stratum, yielding average 2 SD variance of 0.8 ‰. All oxygen-isotope measurements were made during a single analytical

session, during which the measured values of the Durango standard did not show any significant drift.

To estimate temperatures of conodont apatite precipitation, we made use of the equations of Pucéat et al.[65] updated by Zhang et al.[9] (Eq. 4):

$$T = 118.7 - 4.22 \times (\delta^{18}O_{phos} + 0.9 - \delta^{18}O_{water}) \qquad (4)$$

where T is the temperature of precipitation in °C, $\delta^{18}O_{phos}$ is the measured oxygen-isotope composition of conodont apatite, and $\delta^{18}O_{water}$ is the oxygen-isotope composition of the primary or diagenetic fluid with which oxygen in conodont apatite equilibrated. Fluid $\delta^{18}O_{water}$ of the Ordovician was roughly set at –4.0 ‰ SMOW, based on constrains from carbonate clumped isotopes[13]. According to the Eq. 4, a 1‰ increase in $\delta^{18}O_{conodont}$ indicates a decrease in sea-surface temperature by ~ 4 °C.

## In-situ conodont strontium isotopes

In-situ $^{87}Sr/^{86}Sr$ ratios of conodont albid crown ($^{87}Sr/^{86}Sr_{conodont}$) were measured by laser-ablation multi-collector inductively coupled plasma-mass spectrometry (LA-MC-ICP-MS) analysis at the State Key Laboratory of Geological Processes and Mineral Resources (GPMR) at the China University of Geosciences (Wuhan)[66]. A 193-nm ArF-excimer laser was used during the measurement, with a laser beam diameter of 60 μm. Three $^{87}Sr/^{86}Sr$ measurements were obtained for each specimen, yielding average 2 SD of ~0.0006. Two natural apatite standards, Slyudyanka and MAD (Madagascar apatite), were used to monitor the accuracy of LA-MC-ICP-MS measurements, yielding average values 0.70776 ± 0.00017 (2 SD) and 0.71176 ± 0.00016 (2 SD), respectively, which are consistent with the reference $^{87}Sr/^{86}Sr$ values for Slyudyanka (0.70769 ± 0.00015 (2 SD)) and MAD (0.71180 ± 0.00011 (2 SD)).

## Mercury content and isotopes

Mercury concentration was measured using a LECO AMA254 mercury analyzer in the GPMR, and its isotopes were determined using a MC-ICP-MS with high sensitivity X skimmer cone at the Institute of Geochemistry, Chinese Academy of Sciences, Guiyang, China[21].

For mercury concentration analyses, all samples were freeze-dried to prevent the decomposition of Hg. About 100 mg for mudstone or shales and 150–200 mg for limestone were analyzed. Data reliability was ensured by analysis of international standard coal samples 502–685 (40.5 ppb) after every 12 unknowns, then followed by a repeat, yielding reproducibility of sample concentrations being within 10%. Long-term monitoring of 502–685 ($n = 90$) yield analytical uncertainty 2.5 ppb (1 SD) for the analysis interval.

Pyrolysis method was used to extract Hg for its isotopic analyses. An international standard NIST SRM 997 Tl was used for simultaneous instrumental mass bias correction of Hg and 4 ng/mL SnCl$_2$ solution was used to generate elemental Hg$^0$ before being introduced into the plasma. International standard NIST SRM 3133 was measured after every 3 unknowns to monitor the stability of the instrument. We also analyzed NIST SRM 3177 after every 10 unknowns to examine the instrument accuracy. Hg concentrations of ~2 ng/mL or 1 ng/mL of NIST SRM 3133 and NIST SRM 3177 solutions were prepared for matching measured sample solutions to reduce the matrix dependent mass bias. Hg isotopic composition is reported in $\delta^{202}Hg$ notation in units of per mille (‰) relative to the NIST SRM 3133 Hg standard:

$$\delta^{202}Hg(\permil) = [(^{202}Hg/^{198}Hg_{sample})/(^{202}Hg/^{198}Hg_{standard}) - 1] \times 1000 \qquad (5)$$

Mass independent fractionation (MIF) of Hg isotopes is expressed in $\Delta$ notation ($\Delta^{xxx}Hg$), which describes the difference between the measured $\delta^{xxx}Hg$ and the theoretically predicted $\delta^{xxx}Hg$ value, using the following equations:

$$\Delta^{199}Hg \approx \delta^{199}Hg - (\delta^{202}Hg \times 0.2520) \qquad (6)$$

$$\Delta^{200}Hg \approx \delta^{200}Hg - (\delta^{202}Hg \times 0.5024) \qquad (7)$$

$$\Delta^{201}Hg \approx \delta^{201}Hg - (\delta^{202}Hg \times 0.7520) \qquad (8)$$

Replicate analyses of the NIST 3177 Hg isotope reference standard ($n = 4$) yielded the following: $\delta^{202}Hg = -0.42 \pm 0.06\permil$ (2 SD); $\Delta^{199}Hg = -0.04 \pm 0.02\permil$ (2 SD); $\Delta^{200}Hg = +0.01 \pm 0.04\permil$ (2 SD); $\Delta^{201}Hg = +0.01 \pm 0.04\permil$ (2 SD).

## Carbonate carbon and oxygen isotopes

The analyses of $\delta^{13}C_{carb}$ and $\delta^{18}O_{carb}$ using a Thermo Fisher 253 Plus mass spectrometer at the GPMR. The analytical precision was better than 0.04‰ for $\delta^{13}C_{carb}$ and $\delta^{18}O_{carb}$ based on duplicate analyses of the national reference standards GBW-04416 ($\delta^{13}C_{carb} = +1.61\permil$, $\delta^{18}O_{carb} = -11.59\permil$) and GBW-04417 ($\delta^{13}C_{carb} = -6.06\permil$, $\delta^{18}O_{carb} = -24.12\permil$).

## Elemental and total organic carbon contents

Major- and trace-element concentrations (e.g., Al, Fe and Mn, and Mo, U and Sr) of bulk-rock samples were measured using wavelength-dispersive XRF and inductively coupled plasma mass spectrometry (ICP-MS), respectively in the GPMR. Average analytical uncertainty is better than 5% (RSD−relative standard deviation) for major elements based on repeated analysis of national standards GBW07132, GBW07133, and GBW07407, and better than 2% (RSD) for trace elements based on international standards AGV-2, BHVO-2, BCR-2, and GSR-1.

In the same laboratory, total organic carbon (TOC) was measured using an Elementar Vario Micro Cube analyzer with a detection limit of 0.03% for carbon content. Powdered samples were digested with an excess of 2-N HCl at 50 °C for 12 h, with periodic stirring to ensure complete digestion and inorganic carbon removal. Initial weight ($m_0$) and residue weight ($m_1$) were recorded post-digestion. The residue was dried, ground, and analyzed for carbon content (C%) via elemental analyzer. Total organic carbon (TOC) was calculated as: TOC = C% × ($m_1/m_0$). A standard sample and a repeat were analyzed after every 12 unknowns. Data quality was assessed through multiple analyses of standard sample GSS-8 (C% = 1.97%). Long-term monitoring of GSS-8 ($n = 51$) yield analytical uncertainty 0.05% (1 SD) for the analysis interval. Since all analyses were performed consecutively during the same period, analytical uncertainty is consistent across samples. Thus, while absolute TOC values may carry systematic offsets (in Hg/TOC normalization), temporal trends remain robust.

Total sulfur content (TS) was measured using a Thermo Fisher FlashSmart at Chengdu University of Technology. About 3-4 mg of bulk-rock powder was pyrolyzed, and the generated SO$_2$ was purified through a chromatographic column before entering the elemental analyzer. Multiple analyses of standard sample GSS-1a (S% = 0.0726 wt.%) yielded a standard deviation of ~0.02%.

Rare earth and trace element concentrations of the carbonate fraction (e.g., La$_{carb}$, Ce$_{carb}$, Nd$_{carb}$, Th$_{carb}$) were conducted in the Wuhan SampleSolution Analytical Technology Co., Ltd., using ICP-MS[66]. About 100 mg sample powder was digested with 5% acetic acid at 60 °C (24 h, sealed bath). After centrifugation, ~5 mL supernatant was evaporated to dryness (~105 °C). Then, 0.5 mL concentrated HNO$_3$ was added and evaporated, followed by 0.3 mL HNO$_3$ and 5 mL ultra-pure H$_2$O (2 h heating). The solution was diluted to 100 g for ICP-MS. Duplicates analyzed after every 10 samples yielded relative errors of <5%.

## Data availability

The geochemical data generated in this study are archived in Figshare[67–69].

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

## Acknowledgements

We thank Zhihong Li, Yang Li, Fei Guo and Pingan Yan for help in the field, and Guangyi Sun, Yunjie Wu, Zhenfeng Luo for their assistance with the laboratory work. This research is supported by the NSFC grants (Nos. 42488101 to F.H., 42530205 to H.Z., 42488201 to H.Z., 42477215 to H.Z.), the "CUG Scholar" Scientific Research Funds at China University of Geosciences (No. 2023081) to H.Z., the Postdoctoral Fellowship Program of CPSF under Grant Number GZC20232474 and 2024M753028 to H.Z., the "MOST" Special Fund from State Key Laboratory of Geological Processes and Mineral Resources, China University of Geosciences (No. MSFGPMR2024-104) to H.Z.

## Author contributions

F.H., H.Z. and L.Z. conceived and designed the study; H.Z., L.Z., T.J.A., Z.Y.L., X.D.W. and F.H. jointly made contributions in the sample collection, data analysis and interpretation, writing, editing and review of the manuscript.

## Competing interests

The authors declare no competing interests.
