## [Transparent Peer Review file · Nature Communications]

Volcanism and basalt weathering drove Ordovician climatic cooling

Corresponding Author: Professor Lei Zhang

Version 0:

Reviewer comments:

Reviewer #1

(Remarks to the Author)

See letter to the editor and reviewed document attached.

Reviewer #2

(Remarks to the Author)

Review of the manuscript entitled "Volcanism and basalt weathering drove Ordovician climatic cooling" submitted to NC by Zhao et al. (MS: NCOMMS-25-31073)

A series of remarkable geological events occurred during the Ordovician, including the Great Ordovician Biodiversification Event (GOBE), the Late Ordovician mass extinction (LOME), and the Hirnantian glaciation. The coupling relationships among climate, marine environment, and biological evolution have long been a central focus of Ordovician research. A general cooling trend dominated the Ordovician climate, albeit with short-term fluctuations, and was accompanied by a continuous decline in seawater $87\text{Sr}/86\text{Sr}$ ratios. Debates on the linkage between Ordovician volcanism and climate change have persisted for years; however, progress over the past two decades has been limited due to the poor preservation of volcanic eruption records. In recent years, from the perspective of sedimentary geochemistry, especially the interpretation of Hg chemical strata, the eruption records, time and scale of volcanic events have gradually been identified, but these studies have mainly focused on the area near the Ordovician–Silurian (O/S) boundary. In this manuscript, the authors, for the first time, present Hg–Sr–O isotope chemostratigraphy spanning the entire Ordovician from South China. They argue for a correlation between volcanic activity and global temperature changes, as well as potential driving mechanisms. The data appear reasonable and are well interpreted within the context of existing studies. Overall, I consider this work to be original and believe it enhances our understanding of Ordovician climate change. However, I have some concerns regarding the data processing and interpretation in the manuscript. The main issues are as follows:

(1) Biostratigraphy

My first major concern, and perhaps the most critical one, pertains to the stratigraphic framework of this study. The authors selected three distinct sections from the Yichang area and ultimately assembled them into a composite Ordovician stratigraphic column. A robust biostratigraphic framework is essential for this study, particularly as the authors aim to correlate their geochemical data with global Ordovician events and to calculate chronostratigraphic ages and sedimentation rates. Although Figure 1 provides a rough correlation of the biostratigraphy, it is clearly oversimplified. As the authors mentioned, these sections are well-studied. To my knowledge, detailed biostratigraphic data are available for each individual section, and the authors should independently present the conodont/ graptolite biozones for each profile. Such information is a necessary prerequisite for accurate interpretation of geochemical proxies. Moreover, given that the authors themselves have conducted extensive conodont analysis, establishing a detailed biostratigraphic framework should be readily achievable.

(2) Do the Hg chemostratigraphic signals reflect global or local events?

Hg concentration profiles in chemostratigraphy can be influenced not only by global large-scale volcanic eruptions but also by local volcanic activity, redox conditions, sedimentation rates, intensity of chemical weathering, and lithofacies variations. The studied sections show substantial lithofacies heterogeneity. As observed: (1) Hg concentrations differ by 1–2 orders of magnitude between black shales and carbonates; and (2) high Hg content tends to occur in intervals with low sedimentation rates (Fig. S7). Thus, using a uniform background value for Hg concentrations and Hg/TOC ratios to identify volcanic events warrants further scrutiny. Although the authors employ Hg isotopes to distinguish sources, it should be noted that many

Hg/TOC anomalies occur within black shale intervals near the Sandbian–Katian (S/K) and Ordovician–Silurian (O/S) boundaries. The influence of anoxic deposition on Hg isotope signatures is uncertain, potentially yielding either negative or positive $\Delta^{199}\text{Hg}$ values—especially when volcanic eruptions coincide with black shale deposition. In fact, two major volcanic ash layers (bentonites) have been recorded in the Ordovician stratigraphy, corresponding to the S/K and O/S boundaries (Huff et al., 1996; Su et al., 2009), seemingly consistent with the intervals of Hg/TOC anomalies reported by the authors. I recommend that the authors further test the geochemical response of their sections against these known volcanic events. Regarding the background levels of Hg, I suggest establishing lithofacies-specific baselines or calculating Hg burial fluxes as a more robust alternative.

(3) Redox proxies

The authors use proxies such as MoEF, UEF, and C/P ratios to infer local redox conditions. These indicators may be valid for shales (TOC >1%), but likely become unreliable in carbonate-dominated intervals. In particular, the hydrodynamic conditions of the basin can significantly affect the reliability of redox-sensitive trace metal proxies. I recommend supplementing the dataset with Ce anomaly curves for the carbonate samples, which would help better constrain the redox conditions. Moreover, as the authors state, there is no correlation between redox proxies and Hg concentrations. However, the manuscript also reports a positive correlation between Hg and TOC contents. This is contradictory, as elevated TOC content is commonly associated with anoxic conditions. This section of the manuscript requires major revision to clarify and resolve these inconsistencies.

(4) Interpretation of basalt weathering based on Hg-Sr isotopes

During basalt weathering, lighter Hg isotopes are preferentially released (Gao et al., 2023), theoretically causing seawater $\delta^{202}\text{Hg}$ values to decrease and $\Delta^{199}\text{Hg}$ values to increase. While this pattern may be briefly reflected in the Late Darrivilian Cooling Event (LDCE), it is important to note that the long-term decline in seawater $87\text{Sr}/86\text{Sr}$ is a more gradual and sustained trend. Furthermore, volcanic ash layers near the S/K and O/S boundaries typically exhibit arc-type geochemical signatures, with source magmas likely evolving from mantle- to crust-derived compositions (Huff, 2008; Yang et al., 2022). More importantly, the long-term evolution of chemical weathering intensity during the Ordovician remains poorly constrained. Therefore, it is not advisable for the authors to attribute the sustained decrease in seawater Sr isotopic ratios solely to enhanced basalt weathering. While Hg isotopes may offer partial insights, they are insufficient to fully resolve this complex issue.

(5) TOC data and normalization

The authors do not provide sufficient details on the pretreatment procedures for TOC analysis in the methods section. Furthermore, the majority of the limestone samples have TOC contents below 0.1%. For such low-TOC samples, what are the detection limits and analytical uncertainties of the instrumentation used? These directly affect the reliability of Hg/TOC normalization. Although the authors claim that normalization does not artificially amplify natural anomalies, a systematic assessment of the reliability of this approach is still necessary. In many elemental analyzers used for C-S measurements, an error margin of $\pm 0.1\%$ is common, and this should be explicitly addressed. In previous studies, strict criteria have been applied to eliminate false positives in Hg anomalies; for example, see Racki et al. (2019, *Geology*).

Other comments

Figure 1

(1) In South China, the boundary between the Wufeng and Linxiang formations is regionally very consistent and lies at the base of the *Dicellograptus* complexus biozone (Zhang et al., 2019), with a GTS 2020 age of approximately 449 Ma (Goldman et al., 2020).

(2) In recent years, several studies worldwide have reported Middle–Late Ordovician Hg chemostratigraphic records, all underpinned by robust biostratigraphic frameworks. By converting the sample collection height or depth into deposition age, these data could be integrated into Figure 1. Such an approach would be highly appropriate for distinguishing Hg anomalies driven by local versus global volcanic events.

Figure 2

After adding the published Hg data to Figure 1, panel 2B could be removed and replaced with a schematic of Hg isotope fractionation mechanisms. This substitution would aid readers in quickly grasping the key processes discussed in the text.

Lines 95-99: To improve readability and logical flow, this paragraph should be moved to follow line 81.

Lines 255-228: These two sentences are unclear and lack a coherent logical connection to the preceding text.

Lines 234-237: These redox proxies may not be valid for carbonate lithologies.

Lines 238-242: These conclusions contradict the previously noted correlation between Hg and TOC. Clearly, anoxic black shales are enriched in Hg, so the claim that redox conditions have no effect on Hg concentrations is not supported.

Lines 247-253: Sedimentation rate and trace element concentration are not typically linearly related; however, condensed intervals with low sedimentation rates do favor element accumulation.

Lines 261-264: Lithofacies exert a strong control on Hg concentrations. Using a single threshold for anomaly identification is not appropriate. I recommend adopting lithofacies-specific thresholds and a rigorous discrimination protocol to identify volcanic Hg events (see Racki et al., 2018).

Lines 265-275: The volcanic ash deposition record (photographs, thicknesses, frequencies) should be described in detail within the geological background and lithostratigraphic columns. These data are critical for your arguments, please include them in the Supplementary Materials.

Line 296: $\delta^{202}\text{Hg}$

Lines 325-336: Basalt weathering produces significant Hg isotope fractionation; please describe its specific impact on sedimentary Hg isotope signatures (see Gao et al., 2023).

Lines 344-355: As the authors note, bottom-water anoxia and photic-zone euxinia expanded during the Mid–Late Ordovician (Zhang et al., 2022; Liu et al., 2025). Although anoxic deep-basin sediments typically record negative $\Delta^{199}\text{Hg}$ and positive $\delta^{202}\text{Hg}$ signatures, the studied sections lie on a shallow-water platform where redox proxies point to oxic deposition. From a mass-balance standpoint, expanding deep-water anoxia should drive opposite Hg-isotope trends in shallow settings. The authors should therefore discuss this apparent discrepancy.

Lines 362-378: Since the preceding discussion is based entirely on Hg/TOC anomalies, I suggest relocating this paragraph

to precede the main Discussion section.

Ref.

- Gao et al., 2023. Tracing the source and transport of Hg during pedogenesis in strongly weathered tropical soil using Hg isotopes. *Geochimica et Cosmochimica Acta*, 361, 101–112.
- Goldman et al., 2020. The Ordovician Period. In: Gradstein, F.M., Ogg, J.G., Schmitz, M.D., Ogg, G.M. (Eds.), *Geologic Time Scale 2020*. Amsterdam, Netherlands, Elsevier, pp. 631–694.
- Huff et al., 1996. Large-magnitude Middle Ordovician volcanic ashfalls in North America and Europe dimensions, emplacement and post-emplacement characteristics. *Journal of Volcanology and Geothermal Research*, 73, 285–301.
- Huff, 2008. Ordovician K-bentonites: Issues in interpreting and correlating ancient tephra. *Quaternary International*, 178, 276–287.
- Racki et al., 2018. Mercury enrichments and the Frasnian-Famennian biotic crisis: A volcanic trigger proved? *Geology*, 46, 543–546.
- Shen et al., 2019. Mercury in marine Ordovician/Silurian boundary sections of South China is sulfide-hosted and non-volcanic in origin. *Earth and Planetary Science Letters*, 511, 130–140.
- Su et al., 2009. K-bentonite, black-shale and flysch successions at the Ordovician–Silurian transition, South China: Possible sedimentary responses to the accretion of Cathaysia to the Yangtze Block and its implications for the evolution of Gondwana. *Gondwana Research*, 15, 111–130.
- Zhang et al., 2019. Ordovician integrative stratigraphy and timescale of China. *Science China Earth Sciences*, 62, 61–88.
- Zhang et al., 2022. Progressive expansion of seafloor anoxia in the Middle to Late Ordovician Yangtze Sea: Implications for concurrent decline of invertebrate diversity. *Earth and Planetary Science Letters*, 598, 117858.

Reviewer #3

(Remarks to the Author)

This manuscript presents coupled $\delta^{18}\text{O}$ conodont, $\delta^{87}\text{Sr}/\delta^{87}\text{Sr}$ conodont and Hg system chemistry data (Hg/TOC, $\Delta^{199}\text{Hg}$ and $\delta^{202}\text{Hg}$) from the South China GSSP sections. These data are highly significant for reconstructing volcanism and climate change during the Ordovician period. While the scientific hypothesis is reasonable, the storyline and data interpretation require significant improvement to meet the standards of *Nature Communications*. I therefore recommend major revisions to enhance the global significance of the environmental data in this study and strengthen the interpretation of the data.

Detailed comments are provided below.

1. Attention should be paid to the limitations and multiple solutions of the Hg system. Volcanism identified during the Ordovician period lacks matched geological records, such as bentonite or volcanic ash (Huff, 2008). Furthermore, Hg isotope data from other continents is lacking for comparison purposes. Therefore, it is unclear whether the identified volcanism represents a global pattern, a point which the authors should emphasize carefully. The gradual decrease in $\Delta^{199}\text{Hg}$ correlates negatively with long-term global cooling (485–455 Ma), but this correlation disappears during the Late Ordovician. The authors should provide a clear and concise explanation of the relationship between Hg isotopes and climate.

2. With regard to weathering, I would advise the authors to incorporate published $\delta^{87}\text{Sr}/\delta^{87}\text{Sr}$ data from Laurentia into their discussion of changes in oceanic $\delta^{87}\text{Sr}/\delta^{87}\text{Sr}$ and the mechanisms behind them during the Ordovician period. Very limited data is available from South China during the key transition period of the rapid decrease in $\delta^{87}\text{Sr}/\delta^{87}\text{Sr}$. Except for basalt and volcanic rock, the tropical arc would contribute to global weatherability. I therefore recommend that the authors integrate these published data to demonstrate changes in global weatherability and its relationship with climate change (Macdonald et al., 2019, *Science*; Conwell et al., 2022, *Geology*).

3. I have a few further comments regarding the temperature reconstruction. To improve the reliability of the data, the authors should provide a detailed analytical method for the $\delta^{18}\text{O}$ conodont, given that $\delta^{18}\text{O}$ values within the same conodont can vary considerably. Furthermore, the location of South China gradually shifted from $\sim 60^\circ\text{S}$ to the Equator during the Ordovician period (Jin JS et al., 2020, *Geology*). Temperature data from South China may be affected by latitude and temperature gradients, which could cause deviations from the global average temperature. Therefore, I suggest that the authors integrate well-studied published temperature or $\delta^{18}\text{O}$ data for comparison (Goldberg et al., 2021, *PNAS*; Scotese et al., 2021, *ESR*; Edwards et al., 2022, *GSAB*). This would establish a global pattern of climate change during the Ordovician period and make an important contribution to climate reconstruction during the Ordovician.

4. While the weathering of basalt and the expansion of early land plants could explain the changes in Hg and Sr isotopes to some extent, the evidence is insufficient to support these mechanisms. I therefore recommend that the authors focus on the three global cooling events that occurred during the Ordovician period, strengthen the evidence of global volcanism, and establish the long-term relationship between volcanism and climate change.

Additional minor comments:

1) The relationship between Hg and TOC is weak ($r = 0.35$) in the Huanghuachang section, which covers the most of the Ordovician period in this study. Therefore, organic matter may not be the primary host for Hg in this section. Furthermore, the gradual increase in Hg content upwards may be due to an increase in water depth and mudstone content, as evidenced by the lithological succession. Changes in Hg content may indicate that more non-carbonate material is absorbing Hg, rather

than reflecting changes in regional or global Hg flux. I therefore suggest adding carbonate content data to the discussion of the mechanism of Hg changes.

2) Carbon isotopes are an important tool in chemostratigraphy. However, this study did not report carbon isotope data for the GSSP sections, instead using the GTS 2020 curve. To enable more precise comparisons with the GTS 2020 curve, as well as to refine the stratigraphy of the sections in this study, I would recommend that the authors include carbonate and oxygen isotope data for these sections.

3) Hg data from a single section can be interpreted in various ways and has limited global significance. If possible, I would advise the authors to conduct Hg isotope analyses from other continents (e.g., Laurentia) to demonstrate that Hg isotope changes in South China are a global phenomenon. This would greatly enhance the reliability and quality of the manuscript.

Reviewer #4

(Remarks to the Author)

Version 1:

Reviewer comments:

Reviewer #1

(Remarks to the Author)

He Zhao et al., authors of the manuscript entitled "Basalt volcanism and erosion caused Ordovician climate cooling," have thoroughly revised the original contribution in accordance with the reviewers' corrections and evaluations. The revision addresses in detail each of the issues raised by the referees. In particular, my specific comments have been fully taken into account. It should be noted that the incorporation of the information recently processed by the authors further enriches the original version of the manuscript and clearly makes it a more robust and almost new contribution, suitable for publication in Nature Communications. The conclusions are significant and supported by the data, inferences, and discussions on each topic addressed in relation to the progressive cooling of the Ordovician oceans, namely volcanism, weathering, and episodic anoxia. The authors lean toward massive volcanism as the main factor driving the cooling trend throughout the Ordovician and its relationship to the variable development of faunas over time.

Based on my understanding, I suggest publishing this article in Nature Communications, mediating the interpretation and final editorial decision.

Reviewer #2

(Remarks to the Author)

Review of the revised manuscript "Volcanism and basalt weathering drove Ordovician climatic cooling" submitted to NC by Zhao et al. (MS: NCOMMS-25-31073A)

General comments

Thank you for providing me with the opportunity to read through this revised version of this manuscript. The authors have made significant improvements to the original version and have obviously given good consideration to comments of the reviewers. In this study, the authors present a detailed Ordovician Hg-O-Sr isotopic chemostratigraphy and convincingly demonstrate the important role of volcanic activity in driving climate change during the Ordovician. Unlike previous studies that primarily focused on volcanic activity near the Ordovician–Silurian (O/S) boundary, the novelty of this work lies in the first-time identification of two possible large-scale volcanic events during the Early to Middle Ordovician, along with an exploration of their coupling with global cooling episodes. This finding offers a new perspective on the mechanisms driving Ordovician climate change and provides important context for understanding biological evolution during this period. I believe the study not only enhances our understanding of Ordovician climate dynamics but also contributes valuable insights into the interactions between early Earth's environment and the evolution of life. As such, I recommend acceptance after minor revisions.

Other comments

Fig.2: The bottom age of the Wufeng Formation is approximately ~499 Ma. Please mark the major biological evolution events (GOBE, LOME) on the biodiversity curve (I).

L276-278: Delete "However, the lack of high-resolution age models for the study sections precludes detailed examination of the relationship between sedimentation rate and Hg enrichment."

Reviewer #3

(Remarks to the Author)

This revised manuscript presents new evidence (e.g., carbonate-free Hg content, Hg_{cf}) and expands the discussion on the potential sources and causes of Hg. This supports the idea that the Hg in this study originates primarily from volcanism. Furthermore, the authors compiled geological records of Hg and bentonite from global sections, enabling a more accurate comparison between geochemical data and geological records. This new data is crucial for establishing Hg as a reliable indicator of global volcanism during the Ordovician period. The authors also compiled published Sr-C-Nd isotope and temperature data to consolidate the stratigraphic framework, verify changes in global temperature, and enhance the explanation of Hg changes. They also fully considered changes in the paleolatitude of South China during the Ordovician period. I believe that the new dataset and its interpretation in the current version are much more robust and reliable, and that most of my concerns have been addressed. Therefore, I believe it could be accepted for publication after very minor revisions.

Figure 4 in the main text lacks discussion. The data in the different panels is not explained well either. I suggest removing it, as it is not key evidence for your discussion.

I am slightly concerned about the lithology of the bentonite that you identified in Figure S2E of the Dawan Formation. To substantiate the presence of bentonite in your study sections of the Dawan Formation, you should provide more solid evidence, such as photographs of the thin section. I recently found published data on bentonite from the Darriwilian Kuniutan Formation in Wangjiawan (YW2 core), close to the sections studied in this research (see link below). This data provides detailed isotope ages and lithological features that could strengthen argument for the presence of bentonite during the Middle Ordovician period.

Reference: Globally synchronous meteorite rain during the Middle Ordovician.

Reviewer #4

(Remarks to the Author)

Responses to review comments

Reviewer #1 (Remarks to the Author):

C1-1: I have reviewed the paper by He Zhao et al. entitled “Volcanism and basalt weathering drove Ordovician climatic cooling”. This study is a classic chemostratigraphic work on isotopy (O, Hg, Sr) under biostratigraphic control of conodonts that integrates paleontological data on eutracheophyte precursors, and Total Organic Carbon analysis in the Ordovician of South China. Its purpose is understanding the cause of the cooling of the Ordovician Period and the subsequent diversification of life. The manuscript is well done and suitable for this journal.

The article is well written, almost free of typographical errors (indicated in post scriptum). The manuscript consists of an abstract, an introductory part, results on O, Sr and Hg isotopy, discussion on volcanic activity and long-term global cooling, including early evolution of land plants and TOC implications. In addition, a section on methods is developed. References, 3 composite figures and supplementary material are necessary complements to the text, in good shape and quality for publication.

R1-1: We greatly appreciate that the reviewer acknowledged the novelty of this paper and provided valuable suggestions. We have carefully addressed these concerns. Please see our detailed responses to each comment below. Additionally, we have made numerous revisions throughout the manuscript to improve various aspects of the manuscript. Please refer to the revised manuscript for details. Below we numbered the reviewer comments as “C1-1, C1-2 ...” and our responses as “R1-1, R1-2 ...”.

C1-2: The study is correct to my knowledge, recording a series of isotopic curves in 3 sections of S China under previously established conodont biostratigraphic control, whose references should be added. The authors argue that the Ordovician global cooling could have been caused by a complex interplay of CO₂ uptakes by early plant propagation and weathering of basalts after intense volcanism. In the specialized part of the article that concerns my knowledge, I urge the author to correct the names of several genera of conodonts, as indicated in the post scriptum and in the attached reviewed paper. Also, it is suggested to add a number of significant references throughout the manuscript.

R1-2: Very helpful point. In accordance with the reviewers’ comments, we carefully examined previously published papers on biostratigraphy from the three study sections (Wang et al.,^{1,2}; Chen et al.,³; Wang et al.,⁴). We have added these relevant citations for the biostratigraphy of the study sections in Supplementary Note S1, and newly plotted the distribution of biozones for each section in Supplementary Figure 2 (as suggested by

the 2nd reviewer).

Furthermore, to enhance the reliability of stratigraphic division and correlation, we conducted new inorganic carbon isotope analysis ($\delta^{13}\text{C}_{\text{carb}}$, as suggested by the 3rd reviewer), and compared the $\delta^{13}\text{C}_{\text{carb}}$ curve from our sections with global Geological Time Scale data (Cramer and Jarvis⁵) (e.g., Line 112-117, Supplementary Note S1, Supplementary Figure 5). The comparison identified several globally correlatable positive excursions of $\delta^{13}\text{C}_{\text{carb}}$, for example, during the Floian, Mid-Darriwilian (known as the Mid-Darriwilian inorganic carbon excursion - MDICE), and late Sandbian (known as the Guttenberg inorganic carbon excursion - GICE).

Then, we integrated conodont and graptolite biostratigraphic data and $\delta^{13}\text{C}_{\text{carb}}$ from the three study sections. This integrated dataset was compared with the Ordovician temporal stratigraphic framework of South China (Zhang et al.⁶), and the global Ordovician carbon isotope chemostratigraphic framework (Supplementary Figure 5) (Cramer and Jarvis⁵). Based on this analysis, we established the Ordovician stratigraphic and time frameworks in study sections, and discussed the basis for defining the boundaries of various Ordovician Stages and cited relevant literatures, as well as an evaluation of carbon isotope preservation during diagenesis (Supplementary Note S1).

Regarding the names of conodont genera, we acknowledge the reviewers' professionalism and corrections of conodont genera/species and are particularly grateful for this guidance. Following the reviewers' suggestions and referencing relevant literature, we have corrected the conodont species names in the main text (Line 578-581), Supplementary Note S2, and Supplementary Data S3.

C1-3: I consider that the manuscript would be ready for publication after a **moderate revision**, on typographical errors and figure formats, and especially revising the nomenclature of conodont genera. In addition, take care of the references to be added as suggested, following the indications of the post scriptum and the attached reviewed document. This contribution to Nature Communications is an original and valuable work that could be considered for publication after present review and decision of editorial evaluation.

Sincerely,

Guillermo L. Albanesi

R1-3: Thank you for providing a standalone PDF file with your comments clearly marked. This was particularly helpful in allowing us to quickly and accurately identify areas for improvement and enhance the quality of the manuscript.

As detailed in our responses above, we have revised the nomenclature of conodont genera and added relevant citations concerning the biostratigraphy of the study sections and conodont data, in accordance with your suggestions.

Additionally, we have carefully checked for typographical errors and standardized figure formats, reformatted the entire manuscript according to Nature Communications' Formatting Instructions. These refinements ensure our manuscript now aligns more closely with the journal's standards.

PS: Suggestions and corrections:

C1-4: - Line 81: See also: Barnes, C.R., 2004, Ordovician oceans and climate, in Webby, B.D., Paris, F., Droser, M.L., and Percival, I.G., eds., *The Great Ordovician Biodiversification Event*: New York, Columbia University Press, p. 72–76.

R1-4: Cited in Lines 66 and 366.

C1-5:- Lines 102-103: add the references of the papers related to the conodont biostratigraphy of the 3 stratigraphic sections analyzed.

R1-5: We have added new citations for conodont biostratigraphic studies of the Huanghuachang and Chenjiahe sections (Wang et al., ^{1,2}; Wang et al.,⁴) in Supplementary Note S1. Regarding the Wangjiawan section, as it is dominated by clastic sedimentary rocks, graptolites are more abundant than conodonts, and graptolite biostratigraphic data better constrained the time framework of the Ordovician-Silurian boundary transition (Chen et al.,³). Therefore, we have supplementally cited graptolite biostratigraphic data for Wangjiawan section.

C1-6: - Line 183: Floian.

R1-6: Revised.

C1-7: - Line 191: Fm,

R1-7: Revised.

C1-8: - Line 275: Also well reported in the Argentine Precordillera: Huff, W.D., Bergström, S.M., Kolata, D.R., Cingolani, C.A. and Astini, R.A. 1998. Orrlovician K-bentonites in the Argentine Precorrillera: relations to Gondwana margin evolution. In: Pankhurst, R.I. and Rapela, C.W. (eds.) *The ProtoAndean Margin of Gondwana*. Geological Society, London, Special Publications, 142, 107-]26.

R1-8: Cited in Line 329.

C1-9: - Line 278: See also: Barnes, C.R., 2004, Was there an Ordovician superplume event?, in Webby, B.D., Paris, F., Droser, M.L., and Percival, I.G., eds., *The Great Ordovician Biodiversification Event*: New York, Columbia University Press, p. 77-80.

R1-9: Thank you for recommending the reference on the Middle to Late Ordovician superplume event. We have added the citation.

C1-10: - Line 311: Add a reference for the Ordovician System divisions and chrono-biostratigraphy, e.g.

Goldman, D., Leslie, S. A., Liang, Y. and Bergström, S.M., 2022. Ordovician biostratigraphy: index fossils, biozones and correlation. In: Harper, D. A. T., Lefebvre, B., Percival and I. G., Servais, T. (eds.), A Global Synthesis of the Ordovician System: Part 1. Geological Society, London, Special Publications, 532, 31–62.

Also, clarify that, aside on the left margin of the panel A, the formations are named.

R1-10: We have cited Goldman et al. (2022) in the caption of Figure 1. We have newly labeled "Fm" in the left column of Figure 1.

C1-11: - Line 429: See also: Ramos, V.A., Escayola, M., Mutti, D. and Vujovich, G.I., 2000. Proterozoic Paleozoic ophiolites of the Andean basement of southern South America. In: Dilek, Y., Moores, E.M., Elthon, D. and Nicolas, A. eds. Ophiolites and Oceanic Crust: New insights from field studies and the Ocean Drilling Program: Boulder, Colorado, Geological Society of America, Special Paper 349: 331-349.

R1-11: Thank you for recommending the reference on Ordovician ophiolites. We have cited the paper in Line 352.

C1-12: - Line 448: See also: Albanesi, G.L. and Barnes, C.R., 2000, Subspeciation within a punctuated equilibrium evolutionary event: Phylogenetic history of the Lower-Middle Ordovician *Paroistodus originalis*–*P. horridus* complex (Conodonta). *Journal of Paleontology*, v. 74, p. 492–502, doi: 10.1666/0022-3360(2000)074<0492:SWAPEE>2.0.CO;2.

R1-12: Thank you for recommending the reference on Ordovician volcanism and its associated biotic-environmental events. We have cited the paper in Line 498.

C1-13: - Line 491: delete "is".

R1-13: Revised.

C1-14: - Line 530: The amount of conodont elements (absolute frequency) per sample should be indicated. Also indicate how many pelagic conodont elements per sample were used for geochemical analyses.

R1-14: We have briefly shown absolute frequency of extracted conodont specimen (from a few to dozens) in Line 574-576, and number of pelagic conodont elements (1 to 3) per stratum used for oxygen isotope analyses in Line 608-610. More detailed numbers of the extracted conodont specimen are documented in Supplementary Data S3.

C1-15: - Lines 531-533: Caution: several genera of conodonts are misspelled and/or their identification are doubtfuls: *Belodella* (probably *Ansella*), *Pasoistodus* (probably *Paroistodus*), *Pesiodus* (?*Periodon*), *Sessagtognathus* (probably *Serratognathus*), *Trianglodus* (probably *Triangulodus*).

In supplementary material you should correct: *Conurodus* (= *Cornuodus*), *Protopandesodus*

(=Protopanderodus), Nususguathus (= Nasusgnathus), Belodena (= Belodella), Juanogn (=Juanognathus).

R1-15: We sincerely appreciate your specific revision suggestions, which were crucial for strengthening this study. In accordance with your recommendations, we have updated the nomenclature of conodont genera and species throughout the manuscript (Line 578-581; Supplementary Note S2; Supplementary Data S3).

C1-16: Other comments/suggestions in attached PDF file.

R1-16: Based on your comments marked in the PDF manuscript, we made additional revises, including: (1) We incorporated briefly comparative analysis of our Sr isotope curve with seawater Sr isotope curves from Laurentia (Figure 1F, Line 154-155), with supplementary citations; (3) We annotated conodont sampling levels adjacent to the stratigraphic columns in Supplementary Figure 2.

Response: Thank you again!

Reviewer #2 (Remarks to the Author):

C2-1: Review of the manuscript entitled "Volcanism and basalt weathering drove Ordovician climatic cooling" submitted to NC by Zhao et al. (MS: NCOMMS-25-31073)

A series of remarkable geological events occurred during the Ordovician, including the Great Ordovician Biodiversification Event (GOBE), the Late Ordovician mass extinction (LOME), and the Hirnantian glaciation. The coupling relationships among climate, marine environment, and biological evolution have long been a central focus of Ordovician research. A general cooling trend dominated the Ordovician climate, albeit with short-term fluctuations, and was accompanied by a continuous decline in seawater $^{87}\text{Sr}/^{86}\text{Sr}$ ratios. Debates on the linkage between Ordovician volcanism and climate change have persisted for years; however, progress over the past two decades has been limited due to the poor preservation of volcanic eruption records. In recent years, from the perspective of sedimentary geochemistry, especially the interpretation of Hg chemical strata, the eruption records, time and scale of volcanic events have gradually been identified, but these studies have

mainly focused on the area near the Ordovician–Silurian (O/S) boundary. In this manuscript, the authors, for the first time, present Hg-Sr-O isotope chemostratigraphy spanning the entire Ordovician from South China. They argue for a correlation between volcanic activity and global temperature changes, as well as potential driving mechanisms. The data appear reasonable and are well interpreted within the context of existing studies. Overall, I consider this work to be original and believe it enhances our understanding of Ordovician climate change.

However, I have some concerns regarding the data processing and interpretation in the manuscript. The main issues are as follows:

R2-1: We greatly appreciate that the reviewer acknowledged the novelty of this paper and provided valuable suggestions. We have carefully addressed these concerns. Please see our detailed responses to each comment below. We have also made numerous revisions throughout the manuscript to improve various aspects of the manuscript.

Please refer to the revised manuscript for details. Below we numbered the reviewer comments as “C1-1, C1-2 ...” and our responses as “R1-1, R1-2 ...”.

C2-2:

(1) Biostratigraphy

My first major concern, and perhaps the most critical one, pertains to the stratigraphic framework of this study. The authors selected three distinct sections from the Yichang area and ultimately assembled them into a composite Ordovician stratigraphic column. A robust biostratigraphic framework is essential for this study, particularly as the authors aim to correlate their geochemical data with global Ordovician events and to calculate chronostratigraphic ages and sedimentation rates. Although Figure 1 provides a rough correlation of the biostratigraphy, it is clearly oversimplified. As the authors mentioned, these sections are well-studied. To my knowledge, detailed biostratigraphic data are available for each individual section, and the authors should independently present the conodont/graptolite biozones for each profile. Such information is a necessary prerequisite for accurate interpretation of geochemical proxies. Moreover, given that the authors themselves have conducted extensive conodont analysis, establishing a detailed biostratigraphic framework should be readily achievable.

R2-2: Good point. Accordingly, we carefully examined previously published biostratigraphic papers of the three study sections (Wang et al.,^{1,2}; Chen et al.,³; Wang et al.,⁴). We have newly plotted the distribution of biozones for each section in Supplementary Figure 2), and added these relevant citations therein.

Furthermore, to enhance the reliability of stratigraphic division and correlation, we conducted new inorganic carbon isotope analysis ($\delta^{13}\text{C}_{\text{carb}}$, as suggested by the 3rd reviewer), and compared the $\delta^{13}\text{C}_{\text{carb}}$ curve from our sections with global Geological Time Scale data (Cramer and Jarvis⁵) (e.g., Line 112-117, Supplementary Note S1, Supplementary Figure 5). The comparison identified several globally correlatable positive excursions of $\delta^{13}\text{C}_{\text{carb}}$, for example, during the Floian, Mid-Darriwilian (known as the Mid-Darriwilian inorganic carbon excursion - MDICE), and late Sandbian (known as the Guttenberg inorganic carbon excursion - GICE).

Then, we integrated conodont and graptolite biostratigraphic data and $\delta^{13}\text{C}_{\text{carb}}$ from the three study

sections. This integrated dataset was compared with the Ordovician temporal stratigraphic framework of South China (Zhang et al.⁶), and the global Ordovician carbon isotope chemostratigraphic framework (Supplementary Figure 5) (Cramer and Jarvis⁵). Based on this analysis, we established the Ordovician stratigraphic and time frameworks for the study sections, and discussed the basis for defining the boundaries of various Ordovician Stages and cited relevant literatures, as well as an evaluation of carbon isotope preservation during diagenesis (Supplementary Note S1).

Collectively, the integration of published biostratigraphic data from our study sections with new inorganic carbon isotope datasets, supplemented by regional and global comparative analyses, has enabled us to establish a detailed stratigraphic and time frameworks for the study sections.

C2-3: (2) Do the Hg chemostratigraphic signals reflect global or local events?

Hg concentration profiles in chemostratigraphy can be influenced not only by global large-scale volcanic eruptions but also by local volcanic activity, redox conditions, sedimentation rates, intensity of chemical weathering, and lithofacies variations. The studied sections show substantial lithofacies heterogeneity. As observed: (1) Hg concentrations differ by 1–2 orders of magnitude between black shales and carbonates; and (2) high Hg content tends to occur in intervals with low sedimentation rates (Fig. S7). Thus, using a uniform background value for Hg concentrations and Hg/TOC ratios to identify volcanic events warrants further scrutiny. Although the authors employ Hg isotopes to distinguish sources, it should be noted that many Hg/TOC anomalies occur within black shale intervals near the Sandbian–Katian (S/K) and Ordovician–Silurian (O/S) boundaries. The influence of anoxic deposition on Hg isotope signatures is uncertain, potentially yielding either negative or positive $\Delta^{199}\text{Hg}$ values—especially when volcanic eruptions coincide with black shale deposition. In fact, two major volcanic ash layers (bentonites) have been recorded in the Ordovician stratigraphy, corresponding to the S/K and O/S boundaries (Huff et al., 1996; Su et al., 2009), seemingly consistent with the intervals of Hg/TOC anomalies reported by the authors. I recommend that the authors further test the geochemical response of their sections against these known volcanic events. Regarding the background levels of Hg, I suggest establishing lithofacies-specific baselines or calculating Hg burial fluxes as a more robust alternative.

R2-3: Good point. (1) We now use multiple Hg proxies (e.g., Hg content, carbonate-free Hg content (Hg_{cf}), Hg/TOC ratio, Hg enrichment factor (Hg_{EF})) and adopt multiple baseline values and thresholds (for carbonate rocks, shales, high TOC samples) to constrain Hg enrichment intervals in study sections (and in global compiled curves). We give a detailed description of these proxies as well as their baseline values and thresholds in the new Supplementary Note S4, and partly shown here:

“First, because carbonates exclude Hg during formation (relative to TOC in sediment), increasing CaCO_3 concentration can dilute the concentration of Hg in sediments, therefore, a carbonate-free Hg content ($\text{Hg}_{\text{cf}} = \text{Hg}/(1-\text{CaCO}_3/100)$) is expected to reflect more real Hg enrichment during deposition⁷. In study sections, CaCO_3 content varies considerably between 0.1% and 99.4%, and there is a roughly increasing trend in Hg content and decrease trend in CaCO_3 content from Lower to Upper Ordovician (even there is no significant correlation between CaCO_3 and Hg), suggesting a potential carbonate dilution of Hg in study samples. Therefore, calculation of Hg_{cf} is applied to jointly evaluate Hg enrichment in study successions.

Second, following previous approach (e.g., Racki et al., 2018), we used median Hg content and Hg/TOC

ratios of study sections (Huanghuachang, Chenjiahe and Wangjiawan) as background values, which are 131.2 ppb and 76.5 ppm/%, respectively, for organic-rich clastic rocks (designated as baselines ③, ④), and 2.4 ppb and 70.0 ppm/%, respectively, for carbonate rocks (designated as baselines ⑦, ⑧) (Figure 1D-E).

Third, given our study spans the whole Ordovician Period, we supplemented these with more broadly representative Phanerozoic-scale baselines derived from >3,500 samples (Grasby et al., 2019), in which average Hg content are 62.4 ppb for shale and 34.3 ppb for carbonate rocks (baselines ①, ②), and 144 ppb/wt% for all sedimentary rock, and 71.9 ppb/wt% only for samples with TOC>0.2% (baselines ⑤, ⑥) (Figure 1D-E).

Fourth, we further implemented Hg enrichment factor (Hg_{EF}) calculations following Racki et al. (2018), defined as $Hg_{EF} = (Hg/TOC)_{sample} / (Hg/TOC)_{baseline}$. Hg/TOC_{baseline} was set as 144.0 ppb/wt% for TOC≤0.2%, and 71.9 ppb/wt% for TOC>0.2% following Grasby et al. (2019). The recurrent of Hg_{EF}>3 is used as threshold to indicate "truly enriched" Hg in sedimentary rock (cf. Racki et al., 2018)."

(2) To enhance the accurate identification of Hg anomalies within our study sections and delineate a globally representative Ordovician Hg enrichment intervals, we integrated our newly generated data to published Ordovician Hg geochemical data from 22 globally distributed sections/cores in South China, Tarim, Baltica, Laurentia and Gondwana (Supplementary Figure 1), including paired 1469 Hg content ([Hg]), 1401 Hg/TOC ratios, 397 Δ¹⁹⁹Hg values and 391 δ¹⁹⁹Hg values (and tuff (or called bentonite/volcanic ash) layers if available) (Supplementary Figure 8).

These data derive from clastic rocks (mainly shales, 74.7%), limestones (23.9%), and volcanic ash layers (1.4%), and predominantly belong to the Upper Ordovician. The new and published Hg content and isotopes are comparable throughout the Ordovician (Supplementary Note S5). Multiple baseline parameters of [Hg], [Hg_{cf}], Hg/TOC and Hg_{EF} were further adopted to identify Hg anomalies in compiled datasets and minimize false Hg enrichment (Supplementary Note S4). Using a unified timescale⁸, the compiled datasets further confirmed Hg enrichment intervals as revealed from our newly generated data, jointly depict plateau values of [Hg_{cf}], Hg/TOC and Hg_{EF} thus abnormal Hg enrichment during the late Floian, Sandbian, and middle Katian, and late Katian to early Silurian.

(3) To evaluate the impacts of regional volcanism to Hg chemostratigraphic signals in study sections, we newly added discussions on paleotectonic setting and volcanic background of study units (Supplementary Note S7).

Supplementary Note S7: "The study units were situated within the platform-slope-basin depositional system located between the Yangtze Platform of South China and the adjacent Cathaysia Block during the Ordovician. Along the margin of the Cathaysia Block proximal to the Jiangnan Slope, a series of NE-SW-trending (in present-day orientation here and below) arc belts developed. This platform-slope-basin system extensively records the deposition of potassium-rich bentonites (K-bentonites, that is, volcanic tuff) especially during the Late Ordovician Hirnantian stage, while reports of K-bentonites from other Ordovician epochs are relatively scarce (Su et al.⁹; this study). This disparity may be related to differences in geological preservation and research intensity. Paleogeographic reconstruction based on bed thickness measurements of the K-bentonites from multiple sections across the Yangtze region indicates that Hirnantian K-bentonite deposition was nearly ubiquitous throughout the Yangtze region. The affected area is roughly estimated to have covered ~0.5 million km²⁹. Geochemical analyses reveal that these K-bentonites originated from volcanic eruptions in

a collisional or accretionary zone tectonic setting^{10,11}. The composition of the parent magma ranges from trachyandesite to rhyodacite, with some rhyolite, further indicating an origin in volcanic-arc and syn-collision to within-plate tectonomagmatic settings along an active continental margin. Their distribution correlates with cratonward-migrating black shales and flysch during a tectophase cycle, directly linking K-bentonite formation to northwestward arc-continent collision and accretion between the Cathaysia Block and Yangtze Craton during the Ordovician-Silurian boundary transition^{9,12}. Our study units are located approximately ~600–800 km from the volcanic vents along the margin of the Cathaysia Block, placing them at the periphery of the K-bentonite depositional extent.

Regional volcanic arc systems can generate global climatic effects. Global chemical weathering of continental volcanic arcs plays as rapid drawdown of CO₂ tied to arc weathering stabilizes surface temperatures over geological time¹³. Besides, probably act as one of the most important trajectories, reworking of accreted carbonate platforms located in mature continental arcs (i.e., crustal carbonate) is an important source of volcanic carbon during supercontinent formation and breakup^{14,15}. Lee et al.¹⁶ proposed climatic cooling effect of island arc (e.g., Neogene) and warming effect of continental arc (e.g., Cretaceous-Paleogene), which were associated with carbon fixation/release sequestered by global crustal carbonate reservoirs during assembly/dispersal of continents, therefore, long-term (>50 Myr) greenhouse-icehouse oscillations may be linked to fluctuations between continental- and island arc-dominated states. McKenzie et al.¹⁷ further demonstrated continental volcanic arc emissions of CO₂ as the principal driver of long-term (multimillion-year time scales) icehouse-greenhouse variability over the past ~720 Myr, in which widespread continental arcs correspond with prominent early Paleozoic and Mesozoic greenhouse climates, whereas reduced continental arc activity corresponds with icehouse climates of the Cryogenian, Late Ordovician, late Paleozoic, and Cenozoic. Other evidence supports the mechanism, for example, extinction of Neo-Tethyan volcanic arcs (i.e., continental volcanic arc) is largely synchronous with phases of CO₂ reduction during climate cooling throughout the early to middle Cenozoic; enhanced volcanism and CO₂ emissions due to unloading of active magmatic provinces on continents during the last deglaciation¹⁸.”

(4) To examine Ordovician Hg geochemical responses to known volcanic events from a global perspective, we examined spatiotemporal distributions of volcanic tuff (or bentonite/volcanic ash) layers, which is the most overt manifestation of volcanism, in study sections and other published sections/areas.

First, we described the occurrence of tuff layers in study sections (Line 563-566: “Besides, multiple volcanic tuff depositions yielded, including a ~4 cm thick tuff layer in the upper Dawan Fm and a ~10 cm thick tuff layer in the middle Miaopo Fm at Huanghuachang section, and at least five tuff layers of ~1 to 4 cm thick in the upper Wufeng Fm at Wangjiawan section (Supplementary Fig. 2).”). Then, we added their field photographs into the Supplementary Information (Supplementary Figures 2E-H and 3) with further descriptions in figure captions: “(E) A tuff layer (~4 cm thick) within calcareous mudstone in the middle part of the Dawan Formation, Huanghuachang section. (F-H) Five tuff layers (numbered 1 to 5 from bottom to top, with thicknesses of ~2 cm, ~2 cm, ~4 cm, ~2.5 cm, and ~1 cm, respectively) within the Wufeng Formation in the Wangjiawan section.”

Second, we additionally compiled previous published volcanic tuff data from 22 globally distributed sections/cores, predominantly from the Middle and Upper Ordovician, where paired Hg measurements had been figured out (Line 192-217; Supplementary Figure 8G; Supplementary Note S5).

Third, we compared above tuff records to global review data as documented in Huff (2008) in Supplementary Note S5: “We noticed that Huff et al.¹⁹ once reported volcanic tuff records across the Lower to Upper Ordovician in multiple continentals, however, the corresponded Hg enrichment is unclear. To enhance global representativeness, we compared our compiled data to volcanic tuff distribution dataset¹⁹ (Supplementary Figure 13). The compilation of these published data shows the most active period of volcanic activity and tuff deposition during the Late Ordovician, moderate during the Middle Ordovician, while minimal during the Early Ordovician. The volcanic tuff layers preserved in the study sections, i.e., one layer in the upper Dawan Fm, one layer in the Miaopo Fm, and five layers in the upper Wufeng Fm (Supplementary Fig. 2), is overall comparable to the global pattern. Moreover, the temporal distribution of the volcanic tuff layer corresponds well with globally prominent Hg anomalies in the Late Ordovician, which further support volcanic origin Hg enrichment in sedimentary rock during the Ordovician.”

Fourth, we conducted additional discussions between Hg records and known volcanic events (continental arc volcanism and LIPs) in Line 320-364: “Apart from Hg anomalies, volcanic tuff layers constitute the most overt manifestation of regional volcanism. The multiple tuff layers present in the Middle and Upper Ordovician of the study sections, along with widespread (~0.5 Mkm²) tuffs of trachyandesitic to rhyodacitic composition across the Yangtze Platform (Supplementary Figs. 1-2), especially in Hirnantian successions, were likely derived from a continental volcanic arc associated with arc magmatism along the margin of the South China Craton during the convergence of the Yangtze and Cathaysia blocks within it⁹ (Supplementary Note S7). A global compilation of the spatiotemporal distribution of volcanic tuff layers (Supplementary Note S5) shows sparse occurrences in the Lower Ordovician, with increasing frequency in the Middle and Upper Ordovician of the USA, southern China, Europe, Argentina and Tarim Basin, China (Supplementary Fig. 13)²⁰. These global occurrences have been attributed to continental arc volcanism linked to closure of the Iapetus Ocean¹⁹. Furthermore, these volcanic tuff layers are typically associated with elevated [Hg] and Hg/TOC ratios exceeding baseline values²¹, near-zero $\Delta^{199}\text{Hg}$ values²², and host formations that display overall Hg anomalies (Supplementary Fig. 8). Although continental arc volcanism (persistently active during oceanic crust subduction) contributed Hg, the spatiotemporal distribution of volcanic tuff layers remains inconsistent with globally documented multiphase Hg enrichment intervals in Ordovician sedimentary successions. Therefore, continental arc volcanism was not the primary driver of the multi-phase Hg enrichment intervals in the Ordovician.

Only a few continental large igneous provinces (LIP) of Ordovician age are currently known. The recently identified Alborz LIP in northern Iran²³ has an estimated basalt volume of ~0.5 Mkm³. This LIP originated from the initial rifting of Cimmerian terranes from the northern Gondwanan margin to form the Paleotethys Ocean. Accompanied by equatorward drift, its paleolatitude shifted from ~30°S in the Early Ordovician to ~10°S by the Late Ordovician. The Alborz LIP has been broadly dated to the Middle to Late Ordovician (~469 to 451 Ma), but the existing radiometric ages are insufficient to narrow its timing or to define a pattern of episodic activity²³ (Fig. 2). Additionally, recent studies have identified a major tectono-magmatic event in the Proto-Qiangtang Ocean within peri-Gondwanan terranes (adjacent to South China) that extended from the late Cambrian to the late Middle Ordovician (~510-460 Ma) based on modest age constraints from zircon U-Pb dating²⁴. This event (Pinghe) is regarded as a potential continental LIP with a volume of ~2.5 Mkm³, although its classification remains insecure given that its composition of dominant S-type and subordinate A-type granites is also consistent with subduction-zone magmatism. Other as yet-identified Ordovician LIPs may have existed. Moreover, high rates of plate motion and oceanic lithosphere formation during the Ordovician, as evidenced by abundant ophiolites²⁵ and extensive basaltic lavas (e.g., in the British Isles, northern Iran and West Junggar) (e.g., Parnell et al.²⁶), exceed those of other periods of the Paleozoic Era²⁷. These factors likely contributed to contemporaneous global peaks in the production rates of volcanogenic massive sulfide deposits, sulfidic shales and ironstones²⁴. Volcanism related to continental LIPs as well as subduction of oceanic lithosphere may have led to globally elevated basalt weathering fluxes,

especially in (sub)tropical regions²⁸, resulting in a prolonged negative shift of $^{87}\text{Sr}/^{86}\text{Sr}$ (by 0.0008) during the Darriwilian to Sandbian, followed by stabilization in the Katian²⁹ (Fig. 2). LIP volcanism is commonly the primary driver of anomalous Hg enrichment in marine sediments during major geological events such as mass extinctions^{30,27}. The current clues regarding Ordovician LIPs suggest that such volcanism may have contributed to the global sedimentary Hg enrichments, although temporal links between LIP activity and Hg enrichment are supported only for limited intervals (~469 Ma, early Dapingian)²³ (Fig. 2).”

(5) We added additional discussions in the main text to explain the influence of seawater redox on Hg isotope signatures ($\Delta^{199}\text{Hg}$ and $\delta^{202}\text{Hg}$) in study sections, and different behaviors of Hg isotope signatures among different settings during the Ordovician in Line 430-443:

“At ~468-459 Ma, PZE dominated marine Hg fractionation (i.e., negative $\Delta^{199}\text{Hg}$, positive $\delta^{202}\text{Hg}$) because of enhanced nutrient inputs during both volcanic rock and tropical arc weathering (as recorded by decreased $^{87}\text{Sr}/^{86}\text{Sr}_{\text{conodont}}$ and increased $\epsilon_{\text{Nd}(t)}$ values; Fig. 2). During the LDCE (~459-453 Ma), prolonged PZE dominated marine Hg fractionation (i.e., strongly negative $\Delta^{199}\text{Hg}$, shift to positive $\delta^{202}\text{Hg}$), which was caused mainly by elevated nutrient influx due to intensified weathering of volcanic rocks (especially in tropical areas²⁸), as evidenced by sharp decreases in $^{87}\text{Sr}/^{86}\text{Sr}_{\text{conodont}}$ and increases in $\epsilon_{\text{Nd}(t)}$ (Fig. 2). Currently, Middle Ordovician PZE has been reported from Tarim Basin and other localities in South China³¹, however, Hg isotope signatures are contrast between nearshore section with PZE such as Nangyigou ($\Delta^{199}\text{Hg}$: $+0.14 \pm 0.08\%$; $\delta^{202}\text{Hg}$: $-2.21 \pm 0.98\%$) and offshore sections such as Xijinhe, Dawangou, Huanghuachang (negative $\Delta^{199}\text{Hg}$ and positive $\delta^{202}\text{Hg}$ values)³², likely due to distinct Hg mass balance processes operating in different reservoirs³³. At ~453-449 Ma, PZE weakened and continental weathering dominated marine Hg fluxes and isotopic fractionation in sedimentary rocks, as indicated by increases of $\Delta^{199}\text{Hg}$ to mostly near-zero values, and declining $\delta^{202}\text{Hg}$ to negative values (Fig. 2 and Supplementary Fig. 8). ”

Collectively, our refined methodology and global review studies provide enhanced temporal constraints on global sedimentary Hg enrichments during the Ordovician, argue against local events (e.g., continental arc volcanism) be marine driver of the Hg enrichment, and further proposed LIPs probably be the main driver of sedimentary Hg enrichment.

C2-4: (3) Redox proxies

The authors use proxies such as MoEF, UEF, and C/P ratios to infer local redox conditions. These indicators may be valid for shales (TOC >1%), but likely become unreliable in carbonate-dominated intervals. In particular, the hydrodynamic conditions of the basin can significantly affect the reliability of redox-sensitive trace metal proxies. I recommend supplementing the dataset with Ce anomaly curves for the carbonate samples, which would help better constrain the redox conditions.

Moreover, as the authors state, there is no correlation between redox proxies and Hg concentrations. However, the manuscript also reports a positive correlation between Hg and TOC contents. This is contradictory, as elevated TOC content is commonly associated with anoxic conditions. This section of the manuscript requires major revision to clarify and resolve these inconsistencies.

R2-4: Reasonable point. Thank you for pointing out this issue that we have overlooked.

Generally, MoEF and U_{EF} proxies were developed primarily based on organic-rich marine sediments under

varying redox conditions in modern oceans. Consequently, they are best suited for organic-rich clastic rocks (e.g., shales) in deep-time stratigraphic records (Algeo and Tribovillard³⁴; Algeo and Li³⁵). Similarly, the C_{org}/P proxy is calibrated mainly for organic-rich facies (TOC > 1%) (Algeo and Ingall³⁶) and is not exclusive to shales, thus it remains valid for organic-rich carbonate sequences. Given that specific intervals within our study sections (e.g., Miaopo Formation, Wufeng-Longmaxi Formation) are dominated by organic-rich shales, we retain MO_{EF} , U_{EF} , and C_{org}/P ratios to reconstruct paleoredox conditions for these strata.

Additionally, for carbonate-dominated intervals, we newly conducted trace element analyses of carbonate fraction to reconstruct seawater redox conditions using the Ce anomaly, and made additional discussions in Supplementary Information (Supplemental Note S6 and Supplementary Figure 9J), partly shown as below:

“In the present study, Ce/Ce^*_{carb} shows an overall decreasing trend from ~1 in the Nanjinguan to Fenxiang Fms, to ~0.8-1 in the Honghuayuan to Guniutan formations, before drop to ~0.6-0.8 in the Miaopo to Linxiang Fms (Supplementary Fig. 9), indicating suboxic-anoxic conditions during the Tremadocian, and suboxic to oxic seawater during the Floian to Katian Stages. The overall pattern of oceanic oxygenation during the Early to Late Ordovician is further supported by a decreasing trend in C_{org}/P profile from ~50 to <10 mol/mol in the Nanjinguan to Honghuayuan Fms, and overall low values <10 mol/mol in the Dawan to Linxiang Fms. The lower U_{EF} and MO_{EF} values (mostly < 3) in the Miaopo Fm confirmed oxic seawater conditions during the Middle-Late Ordovician boundary transition. Both U_{EF} and MO_{EF} values show larger fluctuations, ~5–60 and ~5–120, respectively, at the Ordovician-Silurian transition, suggesting a more reducing seawater to anoxic-sulfidic conditions during the LOME. ”

Then, we explored the relationship of Hg content in the samples to seawater redox changes, and found that redox effects cannot be excluded, even though their impact may be limited to a few reducing intervals. Accordingly, we added new Ce/Ce^* data and prepared a more rigorous analysis in Lines 247-260:

“Seawater redox conditions commonly play a pivotal role in sedimentary Hg accumulation, mostly through modulation of microbial activity and the interplay of the C-S biogeochemical cycles, thus determining the speciation and concentration of mercury within the aquatic environment^{33,37}. Therefore, redox oscillations can have a profound effect on Hg diffusion at the sediment-water interface, directly influencing its enrichment in sediment. To evaluate marine redox variation through the Ordovician, we used molybdenum and uranium-enrichment factors (MO_{EF} and U_{EF}), C_{org}/P ratios, and the cerium anomaly of the carbonate fraction (Ce/Ce^*_{carb}), for either clastic or carbonate successions (Supplementary Fig. 9; Supplementary Note S6). Overall, [Hg] shows no significant correlation to MO_{EF} , U_{EF} , C_{org}/P and Ce/Ce^*_{carb} , and the stratigraphic distribution of [Hg] peaks are not always within intervals of more reducing watermass conditions (i.e., lower Ce/Ce^*_{carb} , and higher MO_{EF} , U_{EF} and C_{org}/P) (Supplementary Figs. 9 and 11). However, a few high-[Hg] samples (e.g., limestone lenses in Miaopo Fm shale) correspond to more reducing conditions (i.e., lower Ce/Ce^*_{carb}), implying that reducing conditions promoted Hg accumulation in limited intervals of the study sections, probably through elevated net burial rates of organic matter and sulfide³⁰. ”

C2-5: (4) Interpretation of basalt weathering based on Hg-Sr isotopes

During basalt weathering, lighter Hg isotopes are preferentially released (Gao et al., 2023), theoretically causing seawater $\delta^{202}Hg$ values to decrease and $\Delta^{199}Hg$ values to increase. While this pattern may be briefly reflected in the Late Darriwilian Cooling Event (LDCE), it is important to note that the long-term decline in

seawater $^{87}\text{Sr}/^{86}\text{Sr}$ is a more gradual and sustained trend. Furthermore, volcanic ash layers near the S/K and O/S boundaries typically exhibit arc-type geochemical signatures, with source magmas likely evolving from mantle- to crust-derived compositions (Huff, 2008; Yang et al., 2022). More importantly, the long-term evolution of chemical weathering intensity during the Ordovician remains poorly constrained. Therefore, it is not advisable for the authors to attribute the sustained decrease in seawater Sr isotopic ratios solely to enhanced basalt weathering. While Hg isotopes may offer partial insights, they are insufficient to fully resolve this complex issue.

R2-5: Good point. We appreciate the reviewer's comments concerning the critical relationships between Hg, Sr isotope variations and continental weathering. Based on global review study of the Ordovician volcanic tuff layers, we realized that influences of continental arc volcanism are larger enough to impact continental weathering. Therefore, we suggested that continental weathering related to continental arc volcanism, together with basalt weathering, may also contribute to the sustained decrease in seawater Sr isotopic ratios. In the revised manuscript, we more carefully discussed continental weathering and responses of Hg-Sr isotopes in an independent section (Line 416-443):

“The integrated $\Delta^{199}\text{Hg}$, $\delta^{202}\text{Hg}$ and $^{87}\text{Sr}/^{86}\text{Sr}_{\text{conodont}}$ indicates episodes of enhanced weathering of volcanic rocks (basalt) (Supplementary Note S8) and marine photic-zone euxinia (PZE) during the Ordovician (Fig. 5). At $\sim 485\text{-}473$ Ma, an interval during which regional continental arc volcanism was weak, the initial phase of the Taconic Orogeny promoted weathering of mafic rocks (e.g., ophiolite and tropical arc detritus)²⁸ along with Hg inputs in low-latitude regions (Fig. 5). This event likely dominated Hg isotopic fractionation in the sediment, as revealed by relatively high $^{87}\text{Sr}/^{86}\text{Sr}_{\text{conodont}}$ and $\Delta^{199}\text{Hg}$ values and scattered $[\text{Hg}_{\text{cf}}]$ peaks, whereas Hg/TOC and Hg_{EF} mostly fell below baseline/threshold values (Fig. 2 and Supplementary Fig. 8). During the LFCE ($\sim 473\text{-}468$ Ma), relatively stable $^{87}\text{Sr}/^{86}\text{Sr}_{\text{conodont}}$ values (decreasing by only 0.0002)³⁸ indicate an overall balance in Sr inputs from mantle/mafic crustal sources (e.g., basalt) and felsic continental weathering (e.g., granite). Over the entire $\sim 485\text{-}468$ Ma interval, the prolonged decrease in $\Delta^{199}\text{Hg}$ (from $>0.3\text{‰}$ to $\sim 0\text{‰}$) and paired negative $\delta^{202}\text{Hg}$ values with minor variation (around -2‰) cannot be explained by Hg photoreduction in photic-zone euxinic/anoxic watermasses³³ but, rather, dominated by continental weathering.

At $\sim 468\text{-}459$ Ma, PZE dominated marine Hg fractionation (i.e., negative $\Delta^{199}\text{Hg}$, positive $\delta^{202}\text{Hg}$) because of enhanced nutrient inputs during both volcanic rock and tropical arc weathering (as recorded by decreased $^{87}\text{Sr}/^{86}\text{Sr}_{\text{conodont}}$ and increased $\epsilon_{\text{Nd}(t)}$ values; Fig. 2). During the LDCE ($\sim 459\text{-}453$ Ma), prolonged PZE dominated marine Hg fractionation (i.e., strongly negative $\Delta^{199}\text{Hg}$, shift to positive $\delta^{202}\text{Hg}$), which was caused mainly by elevated nutrient influx due to intensified weathering of volcanic rocks (especially in tropical areas²⁸), as evidenced by sharp decreases in $^{87}\text{Sr}/^{86}\text{Sr}_{\text{conodont}}$ and increases in $\epsilon_{\text{Nd}(t)}$ (Fig. 2). Currently, Middle Ordovician PZE has been reported from Tarim Basin and other localities in South China³¹, however, Hg isotope signatures are contrast between nearshore section with PZE such as Nangyigou ($\Delta^{199}\text{Hg}$: $+0.14 \pm 0.08\text{‰}$; $\delta^{202}\text{Hg}$: $-2.21 \pm 0.98\text{‰}$) and offshore sections such as Xijinhe, Dawangou, Huanghuachang (negative $\Delta^{199}\text{Hg}$ and positive $\delta^{202}\text{Hg}$ values)³², likely due to distinct Hg mass balance processes operating in different reservoirs³³. At $\sim 453\text{-}449$ Ma, PZE weakened and continental weathering dominated marine Hg fluxes and isotopic fractionation in sedimentary rocks, as indicated by increases of $\Delta^{199}\text{Hg}$ to mostly near-zero values, and declining $\delta^{202}\text{Hg}$ to negative values (Fig. 2 and Supplementary Fig. 8).”

C2-6: (5) TOC data and normalization

The authors do not provide sufficient details on the pretreatment procedures for TOC analysis in the methods section. Furthermore, the majority of the limestone samples have TOC contents below 0.1%. For such low-TOC samples, what are the detection limits and analytical uncertainties of the instrumentation used? These directly affect the reliability of Hg/TOC normalization. Although the authors claim that normalization does not artificially amplify natural anomalies, a systematic assessment of the reliability of this approach is still necessary. In many elemental analyzers used for C-S measurements, an error margin of $\pm 0.1\%$ is common, and this should be explicitly addressed. In previous studies, strict criteria have been applied to eliminate false positives in Hg anomalies; for example, see Racki et al. (2019, Geology).

R2-6: Good point. Now we expanded our description of TOC pretreatment procedures in the "Methods" section, Line 679-690: "In the same laboratory, total organic carbon (TOC) was measured using an Elementar Vario Micro Cube analyzer with a detection limit of 0.03% for carbon content. Powdered samples were digested with an excess of 2 N HCl at 50°C for 12 hours, with periodic stirring to ensure complete digestion and inorganic carbon removal. Initial weight (m_0) and residue weight (m_1) were recorded post-digestion. The residue was dried, ground, and analyzed for carbon content (C%) via elemental analyzer. Total organic carbon (TOC) was calculated as: $\text{TOC} = \text{C}\% \times (m_1/m_0)$. A standard sample and a repeat were analyzed after every 12 unknowns. Data quality was assessed through multiple analyses of standard sample GSS-8 (C% = 1.97%). Long-term monitoring of GSS-8 ($n = 51$) yield analytical uncertainty 0.05% (1σ) for the analysis interval. Since all analyses were performed consecutively during the same period, analytical uncertainty is consistent across samples. Thus, while absolute TOC values may carry systematic offsets (in Hg/TOC normalization), temporal trends remain robust."

Regarding the TOC analytical uncertainties, our TOC analyses were performed using an Elementar Vario Micro Cube instrument (Thermo Fisher Scientific) with a carbon detection limit of 0.03%. Analytical uncertainty for the analysis period was 0.05% (1SD) based on long-term monitoring of reference material GSS-8 ($1.97 \pm 0.05\%$, see figure below) across the analysis period. Crucially, while the instrument measures C%, the TOC uncertainty propagates through the (m_1/m_0) ratio, resulting in propagated uncertainties $< 0.05\%$. Now, we have added more details about the TOC analytical uncertainties in Lines 679-690.

We should note that for clastic rocks (shales) in the study sections, the average TOC ($1.7 \pm 1.2\%$) substantially exceeds this uncertainty, confirming data reliability, whereas carbonate rocks exhibited lower average TOC ($0.05 \pm 0.06\%$) approaching the uncertainty threshold. However, since all carbonate rock analyses were performed consecutively during the same period, analytical uncertainty is consistent across samples. Thus, while absolute TOC values may carry systematic offsets, temporal trends remain robust.

With reference to the TOC analytical uncertainties (Line 640-645), we further evaluated the Hg/TOC proxy. Our Hg content analyses showed an analytical uncertainty of 2.5 ppb (1SD) based on long-term monitoring of reference material 502-685 (40.5 ± 2.5 ppb, see figure above) during the analytical period. Clastic rocks in the study sections averaged Hg content 143 ± 100 ppb, well above the uncertainty threshold, confirming data accuracy. For carbonates in the study sections (avg 7.0 ± 31.1 ppb), uncertainty impacts absolute values more significantly. Nevertheless, consecutive analysis of all carbonate samples within a single time frame ensures consistent analytical conditions. Therefore, while absolute Hg concentrations in carbonates may have systematic bias, relative stratigraphic trends remain interpretable.

We further applied stricter criteria to minimize false positives in Hg anomalies (see R2-3), including (1) using additional proxies of carbonate-free Hg content (Hg_{cf}) (Fendley et al.,⁷) and Hg enrichment factor (Hg_{EF}) (Racki et al.,³⁹), as well as Hg content and Hg/TOC ratio in original manuscript (Supplementary Note S4); (2) adopt multiple baseline values and thresholds (for carbonate rocks, shales, high TOC samples) to constrain Hg enrichment intervals of different lithologies in study sections (Supplementary Note S4), instead of single baseline value in original manuscript; (3) we newly conducted a global review study on published Hg content, Hg/TOC ratios of the Ordovician, made a comparison between our new data and published data (Supplementary Note S5), then applied the above multiple Hg proxies, baseline values and thresholds to jointly constrain Hg anomaly periods in compiled curves based on our newly generated and published dataset (Line 192-217).

Through the above evaluations and operations, we can ensure that the Hg anomaly signals we obtain are reliable and of global significance.

C2-7: Other comments

Figure 1

(1) In South China, the boundary between the Wufeng and Linxiang formations is regionally very consistent and lies at the base of the *Dicellograptus complexus* biozone (Zhang et al., 2019), with a GTS 2020 age of approximately 449 Ma (Goldman et al., 2020).

R2-7: Thank you for the suggestion. We have employed the base of the *Dicellograptus complexus* biozone as basis for definition of the Wufeng-Linxiang formation boundary (Supplementary Figure 3).

C2-8: (2) In recent years, several studies worldwide have reported Middle–Late Ordovician Hg chemostratigraphic records, all underpinned by robust biostratigraphic frameworks. By converting the sample collection height or depth into deposition age, these data could be integrated into Figure 1. Such an approach would be highly appropriate for distinguishing Hg anomalies driven by local versus global volcanic events.

R2-8: Very good point. We thank the reviewer for this suggestion. Following the suggestions, in the revised manuscript (e.g., Line 192-217), we integrated our newly generated data to published Ordovician Hg geochemical data from 22 globally distributed sections/cores in South China, Tarim, Baltica, Laurentia and Gondwana (Supplementary Figure 1), including paired 1469 Hg content ($[Hg]$), 1401 Hg/TOC ratios, 397 $\Delta^{199}Hg$ values and 391 $\delta^{199}Hg$ values (and tuff (or called bentonite/volcanic ash) layers if available) (Supplementary Figure 8) (Supplementary Data). These data derive from clastic rocks (mainly shales, 74.7%),

limestones (23.9%), and volcanic ash layers (1.4%), and predominantly belong to the Upper Ordovician. The new and published Hg content and isotopes are comparable throughout the Ordovician (Supplementary Note S5). Multiple baseline parameters of [Hg], [Hg_{cf}], Hg/TOC and Hg_{EF} were further adopted to identify Hg anomalies in the compiled datasets and to minimize false Hg enrichment (Supplementary Note S4). Using a unified timescale, the compiled datasets further confirmed Hg enrichment intervals as revealed from our newly generated data, jointly depicting plateau values of [Hg_{cf}], Hg/TOC and Hg_{EF} revealing abnormal Hg enrichment during the late Floian, Sandbian, middle Katian, and late Katian to early Silurian (Fig. 1).

Please see more details in the main text (e.g., 193-218), Supplementary Information (Supplementary Notes S4-S5), and relevant figures (e.g., Fig. 1, Supplementary Figures 1 and 8)

C2-9: Figure 2: After adding the published Hg data to Figure 1, panel 2B could be removed and replaced with a schematic of Hg isotope fractionation mechanisms. This substitution would aid readers in quickly grasping the key processes discussed in the text.

R2-9: Good point. Following the reviewer's suggestions, we have integrated the published Hg data into Figure 1 by using a unified timescale. We also added a new schematic diagram of Hg isotope fractionation mechanisms through the key cooling intervals of the Ordovician in the new Figure 5. We maintain that Figure 2B still provides a concise and unique representation of the new and published data, complementing Figure 2A to meticulously depict the overarching trends in Ordovician O-Hg isotope evolution. We have therefore retained Figure 2B in the manuscript.

C2-10: Lines 95-99: To improve readability and logical flow, this paragraph should be moved to follow line 81.

R2-10: Revised (Line 66-73).

C2-11: Lines 255-228: These two sentences are unclear and lack a coherent logical connection to the preceding text.

R2-11: The language was improved, and logical issues were revised (Line 238-246).

C2-12: Lines 234-237: These redox proxies may not be valid for carbonate lithologies.

R2-12: As detailed in our response R2-4, we newly incorporated the Ce anomaly proxy for carbonate components alongside existing C_{org}/P and U_{EF}-Mo_{EF} proxies. This multi-proxy approach provides a comprehensive assessment of seawater redox conditions recorded in carbonate-shale successions (Supplemental Note S6 and Supplementary Figure 9).

C2-13: Lines 238-242: These conclusions contradict the previously noted correlation between Hg and TOC.

Clearly, anoxic black shales are enriched in Hg, so the claim that redox conditions have no effect on Hg concentrations is not supported.

R2-13: As detailed in our response R2-4, we additionally explored the relationship of Hg content in sedimentary rocks to seawater redox changes, and found that redox effects cannot be excluded, even their impact may be limited in few reducing intervals. Accordingly, we revised the manuscript by using newly Ce/Ce* and provided a more rigorous expression in Line 251-260.

C2-14: Lines 247-253: Sedimentation rate and trace element concentration are not typically linearly related; however, condensed intervals with low sedimentation rates do favor element accumulation.

R2-14: Thank you for the comments. Now we made revisions to improve the arguments in Lines 267-278: “Sedimentation rate can exert a significant influence on Hg uptake in sedimentary rocks by modulating their physical, chemical, and biogeochemical attributes. A low sedimentation rate facilitates the uptake of hydrogenous species (such as Hg complexes) by the sediment. In the present study, however, average linear sedimentation rate (LSR) shows no correlation with [Hg] at the substage level of resolution (Supplementary Fig. 12). Although condensed intervals with low sedimentation rates generally favor Hg accumulation, e.g., similar average [Hg] values (~2-3 ppb) are observed in the rapidly deposited Tremadocian (~24 m/Myr), the intermediate-rate Dapingian (~12 m/Myr), and the slowly deposited Floian (~4 m/Myr) stages at Huanghuachang (Supplementary Figs. 9, 12), suggesting that variation in sedimentation rates was not the dominant cause of observed [Hg] peaks. However, the lack of high-resolution age models for the study sections precludes detailed examination of the relationship between sedimentation rate and Hg enrichment.”.

C2-15: Lines 261-264: Lithofacies exert a strong control on Hg concentrations. Using a single threshold for anomaly identification is not appropriate. I recommend adopting lithofacies-specific thresholds and a rigorous discrimination protocol to identify volcanic Hg events (see Racki et al., 2018).

R2-15: We appreciate your constructive comments. As we replied in R2-3: (1) we now use multiple Hg proxies (e.g., Hg content, carbonate-free Hg content (Hg_{cf}), Hg/TOC ratio, Hg enrichment factor (Hg_{EF})) and adopt multiple baseline values and thresholds (for carbonate rocks, shales, high TOC samples) to constrain Hg enrichment intervals in study sections (Supplementary Note S4); (2) Additionally, we integrated our newly generated data to published Ordovician Hg geochemical data from 22 globally distributed sections/cores (e.g., Line 192-217; Supplementary Notes S5; Fig. 1; Supplementary Figures 1 and 8), then the above multiple proxies, baseline values and thresholds were jointly applied to identify a global representative sedimentary Hg enrichment intervals during the Ordovician.

C2-16: Lines 265-275: The volcanic ash deposition record (photographs, thicknesses, frequencies) should be described in detail within the geological background and lithostratigraphic columns. These data are critical for your arguments, please include them in the Supplementary Materials.

R2-16: Good point. As above response R2-3, now we described the occurrence of volcanic tuff layers in the study sections (e.g., Line 563-566; Supplementary Figures 2E-H). We also added the distribution of volcanic

tuff layers in lithostratigraphic columns of Supplementary Figures 3, 5, 6, and 9. The paleotectonic and volcanic settings of the study units are described in detail in Supplementary Note S7.

C2-17: Line 296: $\delta^{202}\text{Hg}$

R2-17: Revised. (Line 285)

C2-18: Lines 325-336: Basalt weathering produces significant Hg isotope fractionation; please describe its specific impact on sedimentary Hg isotope signatures (see Gao et al., 2023).

R2-18: We described sedimentary Hg isotope signatures in response to basalt weathering, and cited Gao et al., (2023) in Line 385-387: “Therefore, weathering of basalts generally causes Hg isotopes in sedimentary rocks to shift toward values associated with the basaltic endmember (i.e., near-zero to positive $\Delta^{199}\text{Hg}$ and negative $\delta^{202}\text{Hg}$ values⁴⁰).”

C2-19: Lines 344-355: As the authors note, bottom-water anoxia and photic-zone euxinia expanded during the Mid–Late Ordovician (Zhang et al., 2022; Liu et al., 2025). Although anoxic deep-basin sediments typically record negative $\Delta^{199}\text{Hg}$ and positive $\delta^{202}\text{Hg}$ signatures, the studied sections lie on a shallow-water platform where redox proxies point to oxic deposition. From a mass-balance standpoint, expanding deep-water anoxia should drive opposite Hg-isotope trends in shallow settings. The authors should therefore discuss this apparent discrepancy.

R2-19: Good point. Our study sections, located on a shallow-water platform with predominantly oxic to suboxic conditions (Supplementary Note S6), likely resided below photic-zone euxinic waters but above bottom-water anoxia during the Mid–Late Ordovician transition. Photochemical decomposition of HgS within photic-zone euxinia preferentially liberates lighter Hg^{2+} isotopes, producing residual aqueous Hg with positive $\delta^{202}\text{Hg}$ (kinetic fractionation) and negative $\Delta^{199}\text{Hg}$ (mass-independent fractionation) (Zheng et al.³⁷). This isotopically distinct Hg subsequently binds to organic matter and clay minerals, depositing as the observed negative $\Delta^{199}\text{Hg}$ and positive $\delta^{202}\text{Hg}$ signals in our sections.

Concurrently, photochemical reactions emit gaseous $\text{Hg}(0)$ bearing positive $\Delta^{199}\text{Hg}$ and negative $\delta^{202}\text{Hg}$ to the atmosphere. After atmospheric transport, this $\text{Hg}(0)$ deposits in distal marine or terrestrial reservoirs (Sun et al.,³³). Mass balance models thus predict contrasting Hg isotope signatures between reservoirs (Sun et al.,³³). While our section records low $\Delta^{199}\text{Hg}$ and high $\delta^{202}\text{Hg}$, coeval strata elsewhere may exhibit high $\Delta^{199}\text{Hg}$ /low $\delta^{202}\text{Hg}$.

Supporting this, nearshore Nangyigou section (Tarim) limestones (Yuan et al.³²) exhibit opposite signals ($\Delta^{199}\text{Hg}$: $+0.14 \pm 0.08\%$; $\delta^{202}\text{Hg}$: $-2.21 \pm 0.98\%$) versus offshore sections (e.g., Xijinhe, Dawangou, Huanghuachang-Chenjiahe) showing negative $\Delta^{199}\text{Hg}$ and positive $\delta^{202}\text{Hg}$ during the Mid–Late Ordovician transition (Liu et al.³¹; this study).

We have clarified this mechanism in the main text (Line 436-441) to preclude ambiguity: “Currently, Middle Ordovician PZE has been reported from Tarim Basin and other localities in South China³¹, however, Hg isotope signatures are contrast between nearshore section with PZE such as Nangyigou ($\Delta^{199}\text{Hg}$: $+0.14 \pm$

0.08‰; $\delta^{202}\text{Hg}$: $-2.21 \pm 0.98\%$) and offshore sections such as Xijinhe, Dawangou, Huanghuachang (negative $\Delta^{199}\text{Hg}$ and positive $\delta^{202}\text{Hg}$ values)³², likely due to distinct Hg mass balance processes operating in different reservoirs³³.”

C2-20: Lines 362-378: Since the preceding discussion is based entirely on Hg/TOC anomalies, I suggest relocating this paragraph to precede the main Discussion section.

R2-20: We have relocated this paragraph to Supplementary Note S3.

C2-21: Ref.

Gao et al., 2023. Tracing the source and transport of Hg during pedogenesis in strongly weathered tropical soil using Hg isotopes. *Geochimica et Cosmochimica Acta*, 361, 101–112.

Goldman et al., 2020. The Ordovician Period. In: Gradstein, F.M., Ogg, J.G, Schmitz, M.D., Ogg, G.M. (Eds.), *Geologic Time Scale 2020*. Amsterdam, Netherlands, Elsevier, pp. 631–694.

Huff et al., 1996. Large-magnitude Middle Ordovician volcanic ashfalls in North America and Europe dimensions, emplacement and post-emplacement characteristics. *Journal of Volcanology and Geothermal Research*, 73, 285–301.

Huff, 2008. Ordovician K-bentonites: Issues in interpreting and correlating ancient tephras. *Quaternary International*, 178, 276–287.

Racki et al., 2018. Mercury enrichments and the Frasnian-Famennian biotic crisis: A volcanic trigger proved? *Geology*, 46, 543–546.

Shen et al., 2019. Mercury in marine Ordovician/Silurian boundary sections of South China is sulfide-hosted and non-volcanic in origin. *Earth and Planetary Science Letters*, 511, 130–140.

Su et al., 2009. K-bentonite, black-shale and flysch successions at the Ordovician–Silurian transition, South China: Possible sedimentary responses to the accretion of Cathaysia to the Yangtze Block and its implications for the evolution of Gondwana. *Gondwana Research*, 15, 111–130.

Zhang et al., 2019. Ordovician integrative stratigraphy and timescale of China. *Science China Earth Sciences*, 62, 61–88.

Zhang et al., 2022. Progressive expansion of seafloor anoxia in the Middle to Late Ordovician Yangtze Sea: Implications for concurrent decline of invertebrate diversity. *Earth and Planetary Science Letters*, 598, 117858.

R2-21: Thank you for providing these key references. We have cited all of them.

Reviewer #3 (Remarks to the Author):

C3-1: This manuscript presents coupled $\delta^{18}\text{O}$ conodont, $\delta^{87}\text{Sr}/\delta^{87}\text{Sr}$ conodont and Hg system chemistry data (Hg/TOC, $\Delta^{199}\text{Hg}$ and $\delta^{202}\text{Hg}$) from the South China GSSP sections. These data are highly significant for reconstructing volcanism and climate change during the Ordovician period. While the scientific hypothesis is reasonable, the storyline and data interpretation require significant improvement to meet the standards of Nature Communications. I therefore recommend major revisions to enhance the global significance of the environmental data in this study and strengthen the interpretation of the data.

R3-1: We greatly appreciate that the reviewer acknowledged the novelty of this paper and provided valuable

suggestions. We have carefully addressed these concerns. Please see our detailed responses to each comment below. Additionally, we have made numerous revisions throughout the manuscript to improve various aspects of the manuscript. Please refer to the revised manuscript for details. Below we numbered the reviewer comments as “C1-1, C1-2 ...” and our responses as “R1-1, R1-2 ...”.

Detailed comments are provided below.

C3-2: 1. Attention should be paid to the limitations and multiple solutions of the Hg system. Volcanism identified during the Ordovician period lacks matched geological records, such as bentonite or volcanic ash (Huff, 2008). Furthermore, Hg isotope data from other continents is lacking for comparison purposes. Therefore, it is unclear whether the identified volcanism represents a global pattern, a point which the authors should emphasize carefully. The gradual decrease in $\Delta^{199}\text{Hg}$ correlates negatively with long-term global cooling (485-455 Ma), but this correlation disappears during the Late Ordovician. The authors should provide a clear and concise explanation of the relationship between Hg isotopes and climate.

R3-2: Very good point. We thank the reviewer for this constructive suggestion. We made major revisions both in the main text and Supplementary Information, and summarized here:

(1) We fully discuss diagenetic preservation of Hg system (Supplementary Note S3) and host phase of Hg in the study samples (e.g., Line 230-237), for example:

In Supplementary Note S3: “Regarding preservations of Hg in sedimentary rock, weathering can remove Hg signals in organic-rich sediment, for example, leading to loss of Hg up to ~90% in highly weathered shales, while degradation may alter the type and quality of organic matter, especially for samples with low hydrogen and high oxygen index values (e.g., Type II, equivalent to burial temperatures of ~60-180 °C), thereby affecting the Hg/TOC ratio⁴¹. Our study successions consist largely of carbonate rocks that underwent low degrees of weathering based on field observations, therefore, weathering is unlikely to have been a dominant influence on Hg/TOC ratios. In addition, the color alteration indices (CAI) of the extracted conodont specimens range from 1 to 3, equivalent to burial temperatures of ~60-200 °C⁴², suggesting degradation of organic matter may have been an influence on Hg/TOC ratios. However, there is no clear difference in the CAI of conodonts between younger and older strata, indicating a relatively uniform level of thermal alteration throughout the study sections. In addition, TOC is strongly correlated with [Hg] at Chenjiahe ($r = +0.77$; $n = 50$; $p < 0.001$) and Wangjiawan sections ($r = +0.49$; $n = 52$; $p < 0.001$) (Supplementary Fig. 11), suggesting degradation of organic matter and/or uncertainty of TOC analysis had a limited effect on Hg/TOC ratio in the two sections. At Huanghuachang section, the correlation between TOC and [Hg] is weaker ($r = +0.35$; $n = 266$; $p < 0.001$), implying potential impacts on Hg/TOC ratios due to degradation of organic matter and/or uncertainty of TOC analysis in carbonate rocks (e.g., higher CaCO_3 content of carbonate rocks in the Tremadocian). Overall, positive excursions and peak values in the Hg/TOC profiles were not caused by low TOC content ($< 0.2 \text{ wt.}\%$)³⁰, suggesting validity of Hg/TOC ratio in tracing Hg enrichment intervals in study sections.”

Line 230-237: “In the study units, [Hg] exhibits pronounced positive covariation with TOC in the Chenjiahe ($r = +0.77$, $n = 50$, $p < 0.001$) and Wangjiawan sections ($r = +0.49$, $n = 52$, $p < 0.001$), but no

significant covariation with [Al], [Mn], [Fe], or TS (Supplementary Fig. 11), implying that organic matter serves as the principal host of mercury. In comparison, [Hg] shows weak positive covariation with TOC, [Al] and [Fe] ($r = +0.35$, $n = 266$, $p < 0.001$; $r = +0.41$, $n = 82$, $p < 0.05$; $r = +0.35$, $n = 83$, $p < 0.05$, respectively), whereas no significant covariation with [Mn] and TS in the Huanghuachang section, suggesting the presence of Hg in multiple host phases (organic matter, clay minerals and Fe-oxides). ”

(2) We applied stricter criterions to evaluate Hg enrichment and minimize false positives of Hg anomalies in sedimentary rock, by using more Hg derived proxies, baseline values and thresholds (Supplementary Note S4). For example:

We use additional proxies of carbonate-free Hg content (Hg_{cf}) (Fendley et al.⁷) and Hg enrichment factor (Hg_{EF}) (Racki et al.³⁹), as well as Hg content and Hg/TOC ratio in original manuscript.

We adopt multiple baseline values and thresholds (for carbonate rocks, shales, and high TOC samples) to constrain Hg enrichment intervals of different lithologies in study sections, instead of single baseline value in original manuscript.

(3) We newly conducted a global review study on published Hg content, Hg/TOC ratios, $\Delta^{199}Hg$ and $\delta^{199}Hg$ values of the Ordovician, and made a comparison between our new data and published data (Supplementary Note S5), and confirmed global significance of our Hg content and Hg isotope data. Then, we applied the above multiple Hg proxies, baseline values and thresholds to jointly constrain Hg anomaly periods in compiled curves based on our newly generated and published dataset. These analyses make the reconstructed sedimentary Hg enrichments globally significant. For example:

In Line 192-217: “To accurately identify Hg anomalies within the study sections and delineate globally representative enrichment intervals, we integrated our newly generated dataset with published Hg geochemical data from 22 globally distributed Ordovician sections/cores in South China, Tarim, Baltica, Laurentia and Gondwana (Fig. 1), including 1433 Hg content ([Hg]), 1361 Hg/TOC ratios, 397 $\Delta^{199}Hg$ values, and 391 $\delta^{199}Hg$ values (Supplementary Fig. 8; Supplementary Data). These data derive from clastic rocks (mainly shales, 75%), limestones (24%), and volcanic ash layers (1%), of mainly Late Ordovician age. The newly generated and published Hg content and isotope data are in good agreement throughout the Ordovician (Supplementary Note S5). Multiple baseline parameters of [Hg], [Hg_{cf}], Hg/TOC and Hg_{EF} were further adopted to accurately identify Hg anomalies in the compiled dataset (Supplementary Note S4). Using a unified timescale⁸, the compiled datasets confirmed Hg enrichment intervals as revealed by our newly generated data, jointly depicting plateau values of [Hg_{cf}], Hg/TOC and Hg_{EF} and, thus, abnormal Hg enrichments during the late Floian, Sandbian, middle Katian, and late Katian to early Silurian (Fig. 2).

LOWESS-smoothed $\Delta^{199}Hg$ and $\delta^{199}Hg$ trends based on our study sections alone closely mirror that of the global composite trend based on both new and published data (Supplementary Note S5) (Fig. 2). Therefore, the identified $\Delta^{199}Hg$ and $\delta^{199}Hg$ signals through the Ordovician in the study sections represent a global pattern. Accordingly, we subdivided the compiled Ordovician $\Delta^{199}Hg$ profile into four intervals, including (1) a decreasing trend of $\Delta^{199}Hg$ values during the early Tremadocian to middle Floian (~485-473 Ma), (2) a dominance of near-zero $\Delta^{199}Hg$ values during the late Floian, followed by a decreasing trend from positive to

near-zero values during the earliest to early Dapingian (i.e., the LFCE) (~473-468 Ma), (3) a gradual decreasing trend to slightly negative values (~-0.05 to -0.1 ‰) through the Darriwilian (~468-459 Ma) until more negative $\Delta^{199}\text{Hg}$ values (~-0.3 to -0.2 ‰) by the middle Sandbian, followed by a rebound to near-zero values in the late Sandbian (i.e., the LDCE) (~459-453 Ma), and (4) mostly near-zero $\Delta^{199}\text{Hg}$ values followed by both near-zero and slightly positive values (~+0.05 to +0.1 ‰) during the early Katian to Hirnantian (~453-443 Ma). ”

In Supplementary Note S5: “LOWESS-smoothed $\delta^{199}\text{Hg}$ curves between the study sections and a compilation with previous published data also shows comparable trends, marked by negative values (from ~-3 to -1‰) with minor variation during the Early Ordovician, followed by a major rise to positive values (from ~0 to +1‰) during the Middle to early Late Ordovician before decreasing to negative values (from ~-2 to 0‰) during the mid-Late Ordovician to OSB transition. Therefore, the identified $\Delta^{199}\text{Hg}$ and $\delta^{199}\text{Hg}$ signals through the Ordovician in the study sections represent a global pattern.”

(4) We also made a more detail description on volcanic tuff layers in study sections (e.g., Line 563-566), newly added discussion on paleotectonic and volcanic setting of study units (Supplementary Note S7), a global review study on spatiotemporal distributions of volcanic tuff (or bentonite/volcanic ash) layers (Supplementary Note S5), and relationship of continental arc volcanism to Hg enrichment and isotopic fractionation during the Ordovician (e.g., Line 320-337). For example:

First, we added more detailed descriptions of the tuff layers in study sections. In Line563-566: “Besides, multiple volcanic tuff depositions yielded, including a ~4 cm thick tuff layer in the upper Dawan Fm and a ~10 cm thick tuff layer in the middle Miaopo Fm at Huanghuachang section, and at least five tuff layers of ~1 to 4 cm thick in the upper Wufeng Fm at Wangjiawan section (Supplementary Fig. 2).”. In Supplementary Figures 2E-H and 3, and figure captions: “(E) A tuff layer (~4 cm thick) within calcareous mudstone in the middle part of the Dawan Formation, Huanghuachang section. (F-H) Five tuff layers (numbered 1 to 5 from bottom to top, with thicknesses of ~2 cm, ~2 cm, ~4 cm, ~2.5 cm, and ~1 cm, respectively) within the Wufeng Formation in the Wangjiawan section.”

Second, we additionally compiled previous published volcanic tuff data from 22 globally distributed sections/cores, predominantly from the Middle and Upper Ordovician, where paired Hg measurements had been figured out (Line 192-217; Supplementary Figure 8G; Supplementary Note S5).

Third, we compared above tuff records to global review data as documented in Huff (2008) in Supplementary Note S5: “We noticed that Huff et al.¹⁹ once reported volcanic tuff records across the Lower to Upper Ordovician in multiple continentals, however, the corresponded Hg enrichment is unclear. To enhance global representativeness, we compared our compiled data to volcanic tuff distribution dataset¹⁹ (Supplementary Figure 13). The compilation of these published data shows the most active period of volcanic activity and tuff deposition during the Late Ordovician, moderate during the Middle Ordovician, while minimal during the Early Ordovician. The volcanic tuff layers preserved in the study sections, i.e., one layer in the upper Dawan Fm, one layer in the Miaopo Fm, and five layers in the upper Wufeng Fm (Supplementary Fig. 2), is overall comparable to the global pattern. Moreover, the temporal distribution of the volcanic tuff layer corresponds well with globally prominent Hg anomalies in the Late Ordovician, which further support volcanic origin Hg enrichment in sedimentary rock during the Ordovician.”

Fourth, we newly discussed relationship of continental arc volcanism to Hg enrichment and isotopic fractionation in Line 320-337: “Apart from Hg anomalies, volcanic tuff layers constitute the most overt manifestation of regional volcanism. The multiple tuff layers present in the Middle and Upper Ordovician of the study sections, along with widespread (~0.5 Mkm²) tuffs of trachyandesitic to rhyodacitic composition across the Yangtze Platform (Supplementary Figs. 1-2), especially in Hirnantian successions, were likely derived from a continental volcanic arc associated with arc magmatism along the margin of the South China Craton during the convergence of the Yangtze and Cathaysia blocks within it⁹ (Supplementary Note S7). A global compilation of the spatiotemporal distribution of volcanic tuff layers (Supplementary Note S5) shows sparse occurrences in the Lower Ordovician, with increasing frequency in the Middle and Upper Ordovician of the USA, southern China, Europe, Argentina and Tarim Basin, China (Supplementary Fig. 13)²⁰. These global occurrences have been attributed to continental arc volcanism linked to closure of the Iapetus Ocean¹⁹. Furthermore, these volcanic tuff layers are typically associated with elevated [Hg] and Hg/TOC ratios exceeding baseline values²¹, near-zero $\Delta^{199}\text{Hg}$ values²², and host formations that display overall Hg anomalies (Supplementary Fig. 8). Although continental arc volcanism (persistently active during oceanic crust subduction) contributed Hg influxes, the spatiotemporal distribution of volcanic tuff layers remains inconsistent with globally documented multiphase Hg enrichment intervals in Ordovician sedimentary successions. Therefore, continental arc volcanism was not the primary driver of the multi-phase Hg enrichment intervals in the Ordovician.”

(5) In addition, based on newly generated and globally compilations of Hg geochemical data, we confirmed the global significance of the identified Hg enrichment intervals, and we further discussed their relationship to LIPs.

In Line 315-319: “In both the study sections and other successions globally, plateau values of [Hg_{cf}], Hg/TOC, and Hg_{EF} are above baseline levels (Fig. 2 and Supplementary Fig. 12), and the corresponding plateau/peak values in both Hg/Al and Hg/Fe ratios (Supplementary Fig. 9) indicate major volcanic Hg inputs during the late Floian, Sandbian, middle Katian, and late Katian to early Silurian (Supplementary Note S5).”

In Line 338-364: “Only a few continental large igneous provinces (LIP) of Ordovician age are currently known. The recently identified Alborz LIP in northern Iran²³ has an estimated basalt volume of ~0.5 Mkm³. This LIP originated from the initial rifting of Cimmerian terranes from the northern Gondwanan margin to form the Paleotethys Ocean. Accompanied by equatorward drift, its paleolatitude shifted from ~30°S in the Early Ordovician to ~10°S by the Late Ordovician. The Alborz LIP has been broadly dated to the Middle to Late Ordovician (~469 to 451 Ma), but the existing radiometric ages are insufficient to narrow its timing or to define a pattern of episodic activity²³ (Fig. 2). Additionally, recent studies have identified a major tectono-magmatic event in the Proto-Qiangtang Ocean within peri-Gondwanan terranes (adjacent to South China) that extended from the late Cambrian to the late Middle Ordovician (~510-460 Ma) based on modest age constraints from zircon U-Pb dating²⁴. This event (Pinghe) is regarded as a potential continental LIP with a volume of ~2.5 Mkm³, although its classification remains insecure given that its composition of dominant S-type and subordinate A-type granites is also consistent with subduction-zone magmatism. Other as yet-unidentified Ordovician LIPs may have existed. Moreover, high rates of plate motion and oceanic lithosphere formation during the Ordovician, as evidenced by abundant ophiolites²⁵ and extensive basaltic lavas (e.g., in the British Isles, northern Iran and West Junggar) (e.g., Parnell et al.²⁶), exceed those of other periods of the Paleozoic Era²⁷. These factors likely contributed to contemporaneous global peaks in the production rates of volcanogenic massive sulfide deposits, sulfidic shales and ironstones²⁴. Volcanism related to continental LIPs as well as subduction of oceanic lithosphere may have led to globally elevated basalt weathering fluxes, especially in (sub)tropical regions²⁸, resulting in a prolonged negative shift of ⁸⁷Sr/⁸⁶Sr (by 0.0008) during the Darriwilian to Sandbian, followed by stabilization in the Katian²⁹ (Fig. 2). LIP volcanism is commonly the

primary driver of anomalous Hg enrichment in marine sediments during major geological events such as mass extinctions^{30,27}. The current clues regarding Ordovician LIPs suggest that such volcanism may have contributed to the global sedimentary Hg enrichments, although temporal links between LIP activity and Hg enrichment are supported only for limited intervals (~469 Ma, early Dapingian)²³ (Fig. 2).”

(6) Long-term negative excursions of $\Delta^{199}\text{Hg}$ have been linked to continental weathering at ~485-473 Ma (when climate cooling still not happened), and to oceanic PZE (concurrent with cooling climate, e.g., LDCE) at ~468-448 Ma which was triggered by intensive volcanism and volcanic rock weathering. To more clearly discuss volcanism, weathering, PZE and their relationships to Hg and other proxies, we made major changes on the structure of the “Discussion”, and provide a clearer and more concise summary of the relationship between Hg isotopes and climate by the end. For example:

In Line 536-543: “Overall, secular climatic cooling from the LFCE to the Hirnantian Glaciation was probably driven by continental LIP volcanism and basalt weathering, and may be further amplified by elevation of marine productivity (thus PZE). Concurrent with the three-phase climatic cooling, Hg enrichments in sedimentary rock were mainly driven by episodically intensified volcanism related to continental LIP under background of prolonged continental arc volcanism. Hg fractionation was largely driven by continental weathering (especially the basalt) and the following oceanic PZE in the ~485-473 Ma and ~468-448 Ma interval, and by volcanism in the LFCE and Hirnantian Glaciation.”

C3-3: 2. With regard to weathering, I would advise the authors to incorporate published $\delta^{87}\text{Sr}/\delta^{86}\text{Sr}$ data from Laurentia into their discussion of changes in oceanic $\delta^{87}\text{Sr}/\delta^{86}\text{Sr}$ and the mechanisms behind them during the Ordovician period. Very limited data is available from South China during the key transition period of the rapid decrease in $\delta^{87}\text{Sr}/\delta^{86}\text{Sr}$. Except for basalt and volcanic rock, the tropical arc would contribute to global weatherability. I therefore recommend that the authors integrate these published data to demonstrate changes in global weatherability and its relationship with climate change (Macdonald et al., 2019, Science; Conwell et al., 2022, Geology).

R3-3: Good point. In the revised manuscript, we have compiled published marine $^{87}\text{Sr}/^{86}\text{Sr}$ data and Nd isotopes from the Laurentian continent (Saltzman et al.³⁸; McArthur et al.⁴³; Conwell et al.⁴⁴), and plotted them within a unified Ordovician timescale in Figure 1G-H. Then, we compared published $^{87}\text{Sr}/^{86}\text{Sr}$ data with our new dataset, in which our new data overlap nearly perfectly with published records, demonstrating the global significance of our curve (Line 154-155). Collectively, these integrated Sr-isotope and Nd-isotope datasets provide high-resolution constraints for evaluating continental weathering during the Ordovician.

We newly emphasized tropical arc weathering, together with basalt and volcanic rock weathering in composing global weatherability, for example:

In Line 416-443: “The integrated $\Delta^{199}\text{Hg}$, $\delta^{202}\text{Hg}$ and $^{87}\text{Sr}/^{86}\text{Sr}_{\text{conodont}}$ indicates episodes of enhanced weathering of volcanic rocks (basalt) (Supplementary Note S8) and marine photic-zone euxinia (PZE) during the Ordovician (Fig. 5). At ~485-473 Ma, an interval during which regional continental arc volcanism was weak, the initial phase of the Taconic Orogeny promoted weathering of mafic rocks (e.g., ophiolite and tropical arc detritus)²⁸ along with Hg inputs in low-latitude regions (Fig. 5). This event likely dominated Hg isotopic fractionation in the sediment, as revealed by relatively high $^{87}\text{Sr}/^{86}\text{Sr}_{\text{conodont}}$ and $\Delta^{199}\text{Hg}$ values and scattered

[Hg_{cf}] peaks, whereas Hg/TOC and Hg_{EF} mostly fell below baseline/threshold values (Fig. 2 and Supplementary Fig. 8). During the LFCE (~473-468 Ma), relatively stable ⁸⁷Sr/⁸⁶Sr_{conodont} values (decreasing by only 0.0002)³⁸ indicate an overall balance in Sr inputs from mantle/mafic crustal sources (e.g., basalt) and felsic continental weathering (e.g., granite). Over the entire ~485-468 Ma interval, the prolonged decrease in Δ¹⁹⁹Hg (from >0.3‰ to ~0‰) and paired negative δ²⁰²Hg values with minor variation (around -2‰) cannot be explained by Hg photoreduction in photic-zone euxinic/anoxic watermasses³³ but, rather, dominated by continental weathering.

At ~468-459 Ma, PZE dominated marine Hg fractionation (i.e., negative Δ¹⁹⁹Hg, positive δ²⁰²Hg) because of enhanced nutrient inputs during both volcanic rock and tropical arc weathering (as recorded by decreased ⁸⁷Sr/⁸⁶Sr_{conodont} and increased ε_{Nd(t)} values; Fig. 2). During the LDCE (~459-453 Ma), prolonged PZE dominated marine Hg fractionation (i.e., strongly negative Δ¹⁹⁹Hg, shift to positive δ²⁰²Hg), which was caused mainly by elevated nutrient influx due to intensified weathering of volcanic rocks (especially in tropical areas²⁸), as evidenced by sharp decreases in ⁸⁷Sr/⁸⁶Sr_{conodont} and increases in ε_{Nd(t)} (Fig. 2). Currently, Middle Ordovician PZE has been reported from Tarim Basin and other localities in South China³¹, however, Hg isotope signatures are contrast between nearshore section with PZE such as Nangyigou (Δ¹⁹⁹Hg: +0.14 ± 0.08‰; δ²⁰²Hg: -2.21 ± 0.98‰) and offshore sections such as Xijinhe, Dawangou, Huanghuachang (negative Δ¹⁹⁹Hg and positive δ²⁰²Hg values)³², likely due to distinct Hg mass balance processes operating in different reservoirs³³. At ~453-449 Ma, PZE weakened and continental weathering dominated marine Hg fluxes and isotopic fractionation in sedimentary rocks, as indicated by increases of Δ¹⁹⁹Hg to mostly near-zero values, and declining δ²⁰²Hg to negative values (Fig. 2 and Supplementary Fig. 8). ”

C3-4: 3. I have a few further comments regarding the temperature reconstruction. To improve the reliability of the data, the authors should provide a detailed analytical method for the δ¹⁸O_{conodont}, given that δ¹⁸O values within the same conodont can vary considerably.

R3-4: Thank you for your comments. In the revised manuscript, we have expanded methodological details on in-situ microanalysis within the Methods section, and expanded discussion on preservation of conodont δ¹⁸O values in Supplementary Information, for example:

In Line 582-585: “Conodont samples were embedded in resin, followed by grinding and polishing to produce a flat surface suitable for in-situ oxygen isotope analysis. The densest albid crown was targeted for in-situ O isotope analyses in this study to minimize tissue-related effect, reduce biases in reconstructed temperature curve, and facilitate cross-case comparisons (Supplementary Note S2).”.

In Line 599-612: “SIMS measurements were made using a Cameca IMS-1280 SIMS at the Institute of Geology and Geophysics, Chinese Academy of Sciences. Oxygen isotopes were measured using the multi-collection mode on two off-axis Faraday cups. The intensity of ¹⁶O was typically 1×10⁹ cps. The nuclear magnetic resonance probe was used to control stability of the magnetic field. A single analysis took ~5 min consisting of pre-sputtering (~120 s), automatic beam centering (~60 s) and integration of oxygen isotopes (20 cycles × 4 s, total 80 s). SIMS measurements were performed using identical laser parameters (20 μm spot diameter, 80 s ablation duration) to minimize systematic biases.

The instrumental mass fractionation factor (IMF) was corrected using the Durango apatite standard.

Measured $^{18}\text{O}/^{16}\text{O}$ ratios were normalized to Vienna Standard Mean Ocean Water compositions (V-SMOW, $^{18}\text{O}/^{16}\text{O} = 0.0020052$), and then corrected for the instrumental mass fractionation factor (IMF) as follows:

$$(\delta^{18}\text{O})_M = \left(\frac{(^{18}\text{O}/^{16}\text{O})_M}{0.0020052} - 1 \right) \times 1000 (\text{‰}) \quad (1)$$

$$\text{IMF} = (\delta^{18}\text{O})_{M(\text{standard})} - (\delta^{18}\text{O})_{\text{VSMOW}} \quad (2)$$

$$(\delta^{18}\text{O})_{\text{sample}} = (\delta^{18}\text{O})_M - \text{IMF} \quad (3)$$

Replicate analyses of the Durango apatite standard (which was analyzed after every five sample measurements) yielded an average value of $+9.40 \pm 0.16 \text{‰}$ (2σ ; $n = 78$), which is indistinguishable within analytical uncertainty from the reported value of $+9.4 \pm 0.3 \text{‰}$ (2σ)⁴⁵. An average of ~ 5 measurements of $\delta^{18}\text{O}$ were obtained for 1 to 3 specimen(s) from each stratum, yielding average 2σ variance of 0.8‰ . All oxygen-isotope measurements were made during a single analytical session, during which the measured values of the Durango standard did not show any significant drift. ”.

In Supplementary Note S2: “The color alteration indices (CAI) of the study specimens range from 1 to 3, indicating limited thermal alteration in the bioapatite structure and its oxygen isotopic composition^{29,42,45-49}. Although conodonts are inhomogeneous in structure and chemical composition^{42,50}, conodont albid crown is the densest bioapatite tissue that are more preferences for extract (near-) primary seawater O isotope signals, thus was targeted for in-situ oxygen isotope analyses and reconstruction of Ordovician sea-surface temperatures (SSTs) in previous^{29,42,45-49} and present studies. In addition, the in-situ O isotopic analysis was conducted using identical laser ablation spot diameter ($20 \mu\text{m}$) and ablation duration (80 s) to minimize systematic biases and further improve the reliability of trends in the reconstructed SST curve.”.

C3-5: Furthermore, the location of South China gradually shifted from $\sim 60^\circ\text{S}$ to the Equator during the Ordovician period (Jin JS et al., 2020, Geology). Temperature data from South China may be affected by latitude and temperature gradients, which could cause deviations from the global average temperature. Therefore, I suggest that the authors integrate well-studied published temperature or $\delta^{18}\text{O}$ data for comparison (Goldberg et al., 2021, PNAS; Scotese et al., 2021, ESR; Edwards et al., 2022, GSAB). This would establish a global pattern of climate change during the Ordovician period and make an important contribution to climate reconstruction during the Ordovician.

R3-5: Good point. We appreciate the reviewer's suggestions.

(1) We newly added more discussion on affection of latitude and temperature gradients on our SST curves in Supplementary Note S2:

“The change of paleolatitude in South China may affect the results of the reconstructed SSTs in the present study. Previous studies suggested that the location of South China gradually shifted from $\sim 30^\circ\text{S}$ during the Early Ordovician to the Equator during the Late Ordovician⁵¹. Therefore, the SST curve from South China may be affected by latitude and temperature gradients, which may cause deviations from the global average temperature, thus a much higher absolute in the reconstructed SSTs is expected (Supplementary Fig. 7). ”

(2) According to reviewer’s suggestion, we more detailly compared our $\delta^{18}\text{O}$ data to published $\delta^{18}\text{O}$ data in Supplementary Note S2:

“The three major cooling episodes that we integrated are reinforced through comparing with previously published $\delta^{18}\text{O}_{\text{conodont}}$ data from other individual locations, using either gas isotope ratio mass spectrometry (GIRMS)^{29,48}, sensitive high resolution ion microprobe (SHRIMP)^{45,47} or SIMS^{46,49} (Fig. 2). Previous studies suggest a bias between SHRIMP and GIRMS conodont analyses, in which the former is systematically $\sim 0.6\text{--}1.3\%$ higher than the later⁵², while none systematically bias between SHRIMP and SIMS analyses. To enhance consistency across different data sources during data compilation, we apply a minimum correction of $+0.6\%$ when converting GIRMS results to the SHRIMP standard.

The newly generated $\delta^{18}\text{O}_{\text{conodont}}$ curve exhibits positive excursions of $\sim 2\text{--}3\%$ during the LFCE and LDCE, as well as the Hirnantian Glaciation, which are consistent with global published $\delta^{18}\text{O}_{\text{conodont}}$ data^{46,47,49} (Fig. 2). For example, the LFCE from the Laurentian margins and Argentine Precordillera^{46,47}, the LDCE from Laurentia and Tarim^{46,49} and the Hirnantian cooling from the Laurentia and Gondwana⁴⁵. An offset of ~ 1 to 1.5% between newly generated and previously published data following the LDCE may be attributed to multiple processes, including spatial difference in SSTs which may relate to paleolatitude, taxon-related effects in conodonts (as we discussed above), and/or biases during data compilation. For example, there is a consistent $\sim 4\text{--}6\text{ }^\circ\text{C}$ temperature excess (corresponding to $\delta^{18}\text{O}$ depletion of $\sim 1\text{--}1.5\%$) in South China relative to global averages during the late Early to Late Ordovician, and this discrepancy was minimal during the Floian-Sandbian but intensified during the Katian, likely reflecting stronger latitudinal thermal gradient effects as South China approached equatorial latitudes in the Late Ordovician.

Although the Hirnantian Glaciation interval was documented in the newly generated data, it was not clearly evident in the globally compiled LOWESS curve (Fig. 2A), probably due to the fact that the short-term, pronounced cooling had been obscured by data smoothing techniques. A previous proposed climate warming around the Katian-Hirnantian boundary^{53,54} is not evident in our new records, possibly because of low data resolution in that interval of the compiled data, smoothing of the data masked the transient fluctuations, or the limited duration or geographic extent of the warming event. Two low $\delta^{18}\text{O}_{\text{conodont}}$ values at the top of the study sections indicate temporary warming conditions at the end of the Hirnantian Glaciation⁵⁴. ”

(3) We further made a comparison between our SST curve and well-studied published temperature data. For example:

In Line 613-621: “To estimate temperatures of conodont apatite precipitation, we made use of the equations of⁵⁵ updated by⁴² (Eq. 4):

$$T = 118.7 - 4.22 \times (\delta^{18}\text{O}_{\text{phos}} + 0.9 - \delta^{18}\text{O}_{\text{water}}) \quad (4)$$

where T is the temperature of precipitation in $^\circ\text{C}$, $\delta^{18}\text{O}_{\text{phos}}$ is the measured oxygen-isotope composition of conodont apatite, and $\delta^{18}\text{O}_{\text{water}}$ is the oxygen-isotope composition of the primary or diagenetic fluid with which oxygen in conodont apatite equilibrated. Fluid $\delta^{18}\text{O}_{\text{water}}$ of the Ordovician was roughly set at -4.0% SMOW, based on constrains from carbonate clumped isotopes⁵⁶. According to the Eq. 4, a 1% increase in $\delta^{18}\text{O}_{\text{conodont}}$ indicates a decrease in sea surface temperature by $\sim 4\text{ }^\circ\text{C}$.”

In Supplementary Note S2: “To explore the global significance of reconstructing sea-surface temperature curve and the documented three cooling episodes (i.e., LFCE, LDCE and Hirnantian Glaciation) in this study,

we calculated SSTs following Eq. 4^{42,55}, and compared our SSTs to curves of the global oceanic average temperature⁵⁷ and oxygen-isotope derived temperature⁵⁸ (Supplementary Fig. 6). Based on paleotemperature calculation formula, our newly generated data and global compiled $\delta^{18}\text{O}_{\text{conodont}}$ data depict $\sim 1.5\text{‰}$, $\sim 1.5\text{‰}$ and $\sim 2.5\text{‰}$ increases in $\delta^{18}\text{O}_{\text{conodont}}$, thus drops of SST by $\sim 7\text{°C}$, $\sim 7\text{°C}$ and $\sim 10\text{°C}$ during the LFCE, LDCE and Hirnantian Glaciation, respectively. The major drop in sea-surface temperature by $\sim 7\text{°C}$ during the LFCE is comparable in the timing and magnitude to temperature curve reported by Goldberg et al.⁵⁸. The LFCE do not clearly shown in Scotese et al.⁵⁷ probably because the $\delta^{18}\text{O}_{\text{conodont}}$ -based data mostly derived from low latitude area (~ 0 to 30°S) (e.g., Albanesi et al.⁴⁷, Liu et al.⁵⁹, and this study), whereas the later study intergraded temperatures data from both low and high latitudinal areas. For the LDCE, the revealed $\sim 7\text{°C}$ drop in sea-surface temperature is comparable to synchronously drop by $\sim 4\text{°C}$ in Goldberg et al.⁵⁸ and $\sim 7\text{°C}$ in Scotese et al.⁵⁷. The Hirnantian Glaciation, even not clearly shown in global compiled $\delta^{18}\text{O}_{\text{conodont}}$ curve, was still revealed by four $\delta^{18}\text{O}_{\text{conodont}}$ peak values in the present study documenting a $\sim 10\text{°C}$ drop which correlates well to $\sim 10\text{°C}$ drop reported in Goldberg et al.⁵⁸ and $\sim 5\text{°C}$ drop in Scotese et al.⁵⁷.

Therefore, our newly acquired $\delta^{18}\text{O}_{\text{conodont}}$ data preserve (near-)primary changes in paleotemperature. The positive excursions of $\delta^{18}\text{O}_{\text{conodont}}$ record a protracted (25-Myr-long) climatic cooling trend that aligns with global data (Supplementary Fig. 6). This trend occurred in multiple stages, beginning around the Lower/Middle Ordovician boundary ($\sim 470\text{ Ma}$), reinforced around the Middle/Late Ordovician boundary ($\sim 460\text{ Ma}$), and culminating in the Hirnantian Glaciation ($\sim 445\text{-}444\text{ Ma}$).”.

C3-6: 4. While the weathering of basalt and the expansion of early land plants could explain the changes in Hg and Sr isotopes to some extent, the evidence is insufficient to support these mechanisms. I therefore recommend that the authors focus on the three global cooling events that occurred during the Ordovician period, strengthen the evidence of global volcanism, and establish the long-term relationship between volcanism and climate change.

R3-6: Thank you for your comments. We removed out most discussions on early land plant from the manuscript. Since basalt weathering plays a significant role in linking volcanism to climate cooling, we retained and more carefully discussed its contribution to the cooling process.

As we responded above, through integration of global O–Hg–Sr isotopes, Hg content, and volcanic tuff data (Line 134-149; 192-217; Supplementary Notes 2, 4-5), we have reconstructed a refined evolution history of global paleotemperatures, volcanism history in terms of LIP and arc volcanisms, PZE and continental weathering of basalt, other volcanic rock, and tropical arc detritus. This synthesis enables a deeper examination of the linkages between episodic volcanism and stepwise climatic cooling.

Furthermore, we expanded discussion of potential global climatic response to regional volcanic arc systems, through literature review of all geologic time that is relevant to this issue (Supplementary Note S7). Then, we further evaluated both the continental arc–volcanism and LIPs during the Ordovician and its relationship to climate changes in the revised manuscript (e.g., Line 456-498).

Please see more details in the main text and Supplementary Information.

Additional minor comments:

C3-7: 1) The relationship between Hg and TOC is weak ($r = 0.35$) in the Huanghuachang section, which covers the most of the Ordovician period in this study. Therefore, organic matter may not be the primary host for Hg in this section.

R3-7: Thank you for your comments. We made revises on the host phases of Hg in study sections in main text (Line 230-237), and used more proxies in constrain volcanic Hg anomalies (Line 327-335), for example:

In Line 230-237: “In the study units, [Hg] exhibits pronounced positive covariation with TOC in the Chenjiahe ($r = +0.77$, $n = 50$, $p < 0.001$) and Wangjiawan sections ($r = +0.49$, $n = 52$, $p < 0.001$), but no significant covariation with [Al], [Mn], [Fe], or TS (Supplementary Fig. 11), implying that organic matter serves as the principal host of mercury. In comparison, [Hg] shows weak positive covariation with TOC, [Al] and [Fe] ($r = +0.35$, $n = 266$, $p < 0.001$; $r = +0.41$, $n = 82$, $p < 0.05$; $r = +0.35$, $n = 83$, $p < 0.05$, respectively), whereas no significant covariation with [Mn] and TS in the Huanghuachang section, suggesting the presence of Hg in multiple host phases (organic matter, clay minerals and Fe-oxides).”.

In Line 311-319: “While Hg/TOC normalization is suitable for tracing volcanic Hg anomalies in the Chenjiahe and Wangjiawan sections, it is inappropriate for the Huanghuachang section, in which organic matter, clay minerals and Fe-oxides all serve as Hg hosts, necessitating the use of Hg/Al and Hg/Fe as complementary proxies⁶⁰. In both the study sections and other successions globally, plateau values of [Hg_{cf}], Hg/TOC, and Hg_{EF} are above baseline levels (Fig. 2 and Supplementary Fig. 12), and the corresponding plateau/peak values in both Hg/Al and Hg/Fe ratios (Supplementary Fig. 9) indicate major volcanic Hg inputs during the late Floian, Sandbian, middle Katian, and late Katian to early Silurian (Supplementary Note S5).”.

C3-8: Furthermore, the gradual increase in Hg content upwards may be due to an increase in water depth and mudstone content, as evidenced by the lithological succession. Changes in Hg content may indicate that more non-carbonate material is absorbing Hg, rather than reflecting changes in regional or global Hg flux. I therefore suggest adding carbonate content data to the discussion of the mechanism of Hg changes.

R3-8: Good point. We newly added calcium carbonate content ([CaCO₃]) to discuss the mechanism of Hg changes, and applied additional proxy (carbonate-free Hg content (Hg_{cf})) to offset carbonate dilute effect of Hg in sediments⁷. Accordingly, we made revises and added new discussion in the manuscript, for example:

In Line 261-266: “Terrestrial fluxes serve as a major source of Hg to the ocean. The overall decrease in calcium carbonate content ([CaCO₃]) and increase in [Al] observed in the study sections suggest local increases in the content of non-carbonate components (e.g., terrestrial materials) during the Ordovician (Supplementary Fig. 9). However, [Hg] exhibits no correlation to [CaCO₃] or [Al] in the study sections (Supplementary Fig. 11), suggesting that the overall increasing trend of [Hg] through the Ordovician cannot be ascribed to Hg absorption by specific lithologic components.”

In Supplementary Note S4: “First, because carbonates exclude Hg during formation (relative to TOC in sediment), increasing CaCO₃ concentration can dilute the concentration of Hg in sediments, therefore, a carbonate-free Hg content ($Hg_{cf} = Hg / (1 - CaCO_3/100)$) is expected to better reflect actual Hg enrichment during deposition⁷. In the study sections, CaCO₃ content varies considerably between 0.1% and 99.4%, and there is a roughly increasing trend in Hg content and decreasing trend in CaCO₃ content from Lower to Upper

Ordovician (even there is no significant correlation between CaCO_3 and Hg), suggesting a potential carbonate dilution of Hg in study samples. Therefore, calculation of Hg_{cf} is applied to jointly evaluate Hg enrichment in study successions (Fig. 2E). ”

C3-9: 2) Carbon isotopes are an important tool in chemostratigraphy. However, this study did not report carbon isotope data for the GSSP sections, instead using the GTS 2020 curve. To enable more precise comparisons with the GTS 2020 curve, as well as to refine the stratigraphy of the sections in this study, I would recommend that the authors include carbonate and oxygen isotope data for these sections.

R3-9: Good point. To enhance the reliability of stratigraphic division and correlation, we conducted new inorganic carbon isotope analysis ($\delta^{13}\text{C}_{\text{carb}}$, as well as $\delta^{13}\text{O}_{\text{carb}}$), and compared the $\delta^{13}\text{C}_{\text{carb}}$ curve from our sections with global Geological Time Scale data (Cramer and Jarvis⁵) (e.g., Line 112-117, Supplementary Note S1, Supplementary Figure 5). The comparison identified several globally correlatable positive excursions of $\delta^{13}\text{C}_{\text{carb}}$, for example, during the Floian, Mid-Darriwilian (known as the Mid-Darriwilian inorganic carbon excursion - MDICE), and late Sandbian (known as the Guttenberg inorganic carbon excursion - GICE).

Then, we integrated conodont and graptolite biostratigraphic data and $\delta^{13}\text{C}_{\text{carb}}$ from the three study sections. This integrated dataset was compared with the Ordovician temporal stratigraphic framework of South China (Zhang et al.⁶), and the global Ordovician carbon isotope chemostratigraphic framework (Supplementary Figure 5) (Cramer and Jarvis⁵). Based on this analysis, we established the Ordovician stratigraphic and time frameworks in study sections, and discussed the basis for defining the boundaries of various Ordovician Stages and cited relevant literatures, as well as an evaluation of carbon isotope preservation during diagenesis (please see more details in Supplementary Note S1).

Collectively, the integration of published biostratigraphic data from our study sections with new inorganic carbon isotope datasets, supplemented by regional and global comparative analyses, has enabled us to establish a detailed stratigraphic and time frameworks for these sections.

The new $\delta^{13}\text{C}_{\text{carb}}$ further enable us to discussion marine productivity changes during Ordovician climatic cooling and volcanism, for example:

In Line 492-498: “As a pivotal adjunct, the volcanic cooling effect was further amplified by enhanced marine primary productivity and organic carbon burial (e.g., peak TOC values in the LDCE interval, Supplementary Fig. 9), driven by elevated nutrient fluxes (e.g., phosphorus) from the weathering of terrestrial volcanic rocks^{29,61}. This is supported by the long-term Ordovician climatic cooling and concurrent of positive excursions of $\delta^{13}\text{C}_{\text{carb}}$ (Fig. 2). These considerations establish connections between the GOBE, the Ordovician climatic cooling, and contemporaneous volcanism (e.g., Albanesi and Barnes⁶²; Miller et al.⁶³).”

C3-10: 3) Hg data from a single section can be interpreted in various ways and has limited global significance. If possible, I would advise the authors to conduct Hg isotope analyses from other continents (e.g., Laurentia) to demonstrate that Hg isotope changes in South China are a global phenomenon. This would greatly enhance the reliability and quality of the manuscript.

R3-10: We totally agree that if more samples for the other sections can be measured, it will enhance our arguments, and we certainly wish to do so. However, the samples from other continents (e.g., Laurentia) are

not available at present. As we responded in R3-2, we newly conducted a global review study on published Hg content, Hg/TOC ratios, $\Delta^{199}\text{Hg}$ and $\delta^{199}\text{Hg}$ values of the Ordovician, and made a comparison between our new data and published data, and finally confirmed global significance of our Hg content and Hg isotope data (Line 192-217; Supplementary Note S5). The global review study therefore enhanced the reliability of our conclusions.

Reviewer #4 (Remarks to the Author):

C4-1: I co-reviewed this manuscript with one of the reviewers who provided the listed reports. This is part of the Nature Communications initiative to facilitate training in peer review and to provide appropriate recognition for Early Career Researchers who co-review manuscripts.

R4-1: We sincerely appreciate your participation in the review process of this manuscript. We have carefully addressed concerns from all reviewers. Please see our detailed responses to each comment above. Additionally, we have made numerous revisions throughout the manuscript to improve various aspects of the manuscript. Please refer to the revised manuscript for details. Below we numbered the reviewer comments as “C1-1, C1-2 ...” and our responses as “R1-1, R1-2 ...”.

Reference:

- 1 Wang, X., Xiang, L. & Ni, S. Biostratigraphy of the Yangtze Gorge area (2): early Palaeozoic era. (1987).
- 2 Wang, X. *et al.* The Global Stratotype Section and Point for the base of the Middle Ordovician Series and the third stage (Dapingian): Episodes, v. 32. *Biostratigraphy and Chronostratigraphy of the Great American Carbonate Bank* **135**, 987-1003 (2009).
- 3 Chen, X. *et al.* The Global Boundary Stratotype Section and Point (GSSP) for the base of the Hirnantian Stage (the uppermost of the Ordovician System). *Episodes Journal of International Geoscience* **29**, 183-196 (2006).
- 4 Wang, Z., Zhen, Y., Ma, X. & Zhang, Y. Ordovician conodonts from the Kuniantan Topa Formation at Chenjihe and Zhenjin, Yichang, Hubei Province, China and their stratigraphic significance. *Acta Micropalaeontologica Sinica* **35**, 17 (2018).
- 5 Cramer, B. & Jarvis, I. in *Geologic Time Scale 2020* 309-343 (Elsevier, 2020).
- 6 Zhang, Y. *et al.* Ordovician integrative stratigraphy and timescale of China. *Science China Earth Sciences* **62**, 61-88 (2019).
- 7 Fendley, I. M. *et al.* Early Jurassic large igneous province carbon emissions constrained by sedimentary mercury. *Nature Geoscience* **17**, 241-248 (2024). <https://doi.org:10.1038/s41561-024-01378-5>
- 8 Goldman, D., Leslie, S. A., Liang, Y. & Bergström, S. M. Ordovician biostratigraphy: index fossils, biozones and correlation. *Geological Society of London, Special Publication* **532**, 31-62 (2022). <https://doi.org:10.1144/sp532-2022-49>
- 9 Su, W. *et al.* K-bentonite, black-shale and flysch successions at the Ordovician–Silurian transition, South China: Possible sedimentary responses to the accretion of Cathaysia to the Yangtze Block and its implications for the evolution of Gondwana. *Gondwana Research* **15**, 111-130 (2009).
- 10 Su, W., He, L., Wang, Yongbiao, Gong, S. & Zou, H. K-bentonite beds and high-resolution integrated stratigraphy of the uppermost Ordovician Wufeng and the lowest Silurian Longmaxi formations in South China. *Science in China (Earth Sciences)* (2003).

- 11 Su, W. *et al.* K-bentonite beds near the Ordovician–Silurian boundary on the Yangtze Platform, South China: preliminary study of the stratigraphic and tectonomagmatic significance. *Serie correlación geológica* **17**, 34 (2003).
- 12 Wang, L., Lin, S. & Xiao, W. Yangtze and Cathaysia blocks of South China: Their separate positions in Gondwana until early Paleozoic juxtaposition. *Geology* (2023). <https://doi.org:10.1130/g51362.1>
- 13 Gernon, T. M. *et al.* Global chemical weathering dominated by continental arcs since the mid-Palaeozoic. *Nature Geoscience* **14**, 690–696 (2021). <https://doi.org:10.1038/s41561-021-00806-0>
- 14 Mason, E., Edmonds, M. & Turchyn, A. V. Remobilization of crustal carbon may dominate volcanic arc emissions. *Science* **357**, 290–294 (2017).
- 15 Zhang, M., Xu, S. & Sano, Y. Deep carbon recycling viewed from global plate tectonics. *National Science Review* **11** (2024). <https://doi.org:10.1093/nsr/nwae089>
- 16 Lee, C. T. A. *et al.* Continental arc-island arc fluctuations, growth of crustal carbonates, and long-term climate change. *Geosphere* **9**, 21–36 (2012). <https://doi.org:10.1130/ges00822.1>
- 17 McKenzie, N. R. *et al.* Continental arc volcanism as the principal driver of icehouse–greenhouse variability. *Science* **352**, 444–447 (2016).
- 18 Sternai, P. *et al.* Magmatic forcing of Cenozoic climate? *Journal of Geophysical Research: Solid Earth* **125**, e2018JB016460 (2020).
- 19 Huff, W. D. Ordovician K-bentonites: Issues in interpreting and correlating ancient tephtras. *Quaternary International* **178**, 276–287 (2008).
- 20 Huff, W. D., Bergström, S. M., Kolata, D. R., Cingolani, C. A. & Astini, R. A. Ordovician K-bentonites in the Argentine Precordillera: relations to Gondwana margin evolution. *Geological Society of London, Special Publication* **142**, 107–126 (1998). <https://doi.org:10.1144/gsl.Sp.1998.142.01.06>
- 21 Wang, Y. *et al.* The influence of Late Ordovician volcanism on the marine environment based on high-resolution mercury data from South China. *GSA Bulletin* **135**, 787–798 (2022). <https://doi.org:10.1130/b36257.1>
- 22 Ni, X. *et al.* Mercury isotopes of the Late Ordovician to Middle Triassic tuff layers in South China link the fate of ancient volcanism and the mass extinction. *Journal of Asian Earth Sciences* **271** (2024). <https://doi.org:10.1016/j.jseaes.2024.106234>
- 23 Derakhshi, M., Ernst, R. E. & Kamo, S. L. Ordovician–Silurian volcanism in northern Iran: Implications for a new Large Igneous Province (LIP) and a robust candidate for the Late Ordovician mass extinction. *Gondwana Research* **107**, 256–280 (2022).
- 24 Barley, M. E., Bekker, A. & Krapež, B. Late Archean to Early Paleoproterozoic global tectonics, environmental change and the rise of atmospheric oxygen. *Earth and Planetary Science Letters* **238**, 156–171 (2005).
- 25 Ramos, V. A., Escayola, M. n., Mutti, D. I. & Vujovich, G. I. in *Ophiolites and Oceanic Crust: New Insights from Field Studies and the Ocean Drilling Program* (2000).
- 26 Parnell, J., Hole, M. & Boyce, A. J. Evidence for microbial activity in British and Irish Ordovician pillow lavas. *Geological Journal* **50**, 497–508 (2015).
- 27 Dilek, Y. & Newcomb, S. Ophiolite concept and its evolution. *Geological Society of America, Special Publication* **373**, 1–16 (2003).
- 28 Swanson-Hysell, N. L. & Macdonald, F. A. Tropical weathering of the Taconic orogeny as a driver for Ordovician cooling. *Geology* **45**, 719–722 (2017).
- 29 Avila, T. D. *et al.* Role of seafloor production versus continental basalt weathering in Middle to Late Ordovician seawater ⁸⁷Sr/⁸⁶Sr and climate. *Earth and Planetary Science Letters* **593**, 117641 (2022).
- 30 Grasby, S. E., Them, T. R., II, Chen, Z., Yin, R. & Ardakani, O. H. Mercury as a proxy for volcanic emissions in the geologic record. *Earth-Science Reviews* **196**, 102880 (2019).
- 31 Liu, M. *et al.* Mercury isotope evidence for Middle Ordovician photic-zone euxinia: Implications for termination of the Great Ordovician biodiversification event. *Gondwana Research* **137**, 131–144 (2025). <https://doi.org:10.1016/j.gr.2024.09.008>

- 32 Yuan, W. *et al.* Mercury isotopes show vascular plants had colonized land extensively by the early Silurian. *Science Advances* **9**, eade9510 (2023).
- 33 Sun, R. *et al.* Mercury isotope evidence for marine photic zone euxinia across the end-Permian mass extinction. *Communications Earth & Environment* **4** (2023).
- 34 Algeo, T. J. & Tribovillard, N. Environmental analysis of paleoceanographic systems based on molybdenum–uranium covariation. *Chemical Geology* **268**, 211–225 (2009).
- 35 Algeo, T. J. & Li, C. Redox classification and calibration of redox thresholds in sedimentary systems. *Geochimica et Cosmochimica Acta* **287**, 8–26 (2020).
- 36 Algeo, T. J. & Ingall, E. Sedimentary C_{org}: P ratios, paleocean ventilation, and Phanerozoic atmospheric pO₂. *Palaeogeography Palaeoclimatology Palaeoecology* **256**, 130–155 (2007).
- 37 Zheng, W. *et al.* Recurrent photic zone euxinia limited ocean oxygenation and animal evolution during the Ediacaran. *Nature Communications* **14** (2023).
- 38 Saltzman, M. R. *et al.* Calibration of a conodont apatite-based Ordovician ⁸⁷Sr/⁸⁶Sr curve to biostratigraphy and geochronology: Implications for stratigraphic resolution. *GSA Bulletin* **126**, 1551–1568 (2014).
- 39 Racki, G., Rakociński, M., Marynowski, L. & Wignall, P. B. Mercury enrichments and the Frasnian–Famennian biotic crisis: A volcanic trigger proved? *Geology* **46**, 543–546 (2018).
[https://doi.org:10.1130/g40233.1](https://doi.org/10.1130/g40233.1)
- 40 Gao, X. *et al.* Tracing the source and transport of Hg during pedogenesis in strongly weathered tropical soil using Hg isotopes. *Geochimica et Cosmochimica Acta* **361**, 101–112 (2023).
[https://doi.org:10.1016/j.gca.2023.10.009](https://doi.org/10.1016/j.gca.2023.10.009)
- 41 Charbonnier, G., Adatte, T., Föllmi, K. B. & Suan, G. Effect of Intense Weathering and Postdepositional Degradation of Organic Matter on Hg/TOC Proxy in Organic-rich Sediments and its Implications for Deep - Time Investigations. *Geochemistry Geophysics Geosystems* **21** (2020).
[https://doi.org:10.1029/2019gc008707](https://doi.org/10.1029/2019gc008707)
- 42 Zhang, L. *et al.* Raman spectral, elemental, crystallinity, and oxygen-isotope variations in conodont apatite during diagenesis. *Geochimica et Cosmochimica Acta* **210**, 184–207 (2017).
- 43 McArthur, J., Howarth, R., Shields, G. & Zhou, Y. in *Geologic Time Scale 2020* 211–238 (Elsevier, 2020).
- 44 Conwell, C. T., Saltzman, M. R., Edwards, C. T., Griffith, E. M. & Adiatma, Y. D. Nd isotopic evidence for enhanced mafic weathering leading to Ordovician cooling. *Geology* **50**, 886–890 (2022).
[https://doi.org:10.1130/g49860.1](https://doi.org/10.1130/g49860.1)
- 45 Trotter, J. A., Williams, I. S., Barnes, C. R., Lécuyer, C. & Nicoll, R. S. Did cooling oceans trigger Ordovician biodiversification? Evidence from conodont thermometry. *Science* **321**, 550–554 (2008).
- 46 Edwards, C. T., Jones, C. M., Quinton, P. C. & Fike, D. A. Oxygen isotope (δ¹⁸O) trends measured from Ordovician conodont apatite using secondary ion mass spectrometry (SIMS): Implications for paleo-thermometry studies. *GSA Bulletin* **134**, 261–274 (2022).
- 47 Albanesi, G. L., Barnes, C. R., Trotter, J. A., Williams, I. S. & Bergström, S. M. Comparative Lower-Middle Ordovician conodont oxygen isotope palaeothermometry of the Argentine Precordillera and Laurentian margins. *Palaeogeography Palaeoclimatology Palaeoecology* **549**, 109115 (2020).
- 48 Männik, P., Lehnert, O., Nolvak, J. & Joachimski, M. M. Climate changes in the pre-Hirnantian Late Ordovician based on δ¹⁸O_{phos} studies from Estonia. *Palaeogeography Palaeoclimatology Palaeoecology* **569**, 110347 (2021).
- 49 Liu, K., Jiang, M., Zhang, L. & Chen, D. A new high-resolution palaeotemperature record during the Middle–Late Ordovician transition derived from conodont δ¹⁸O palaeothermometry. *Journal of the Geological Society* **179**, jgs2021–2148 (2022).
- 50 Trotter, J. A. & Eggins, S. M. Chemical systematics of conodont apatite determined by laser ablation ICPMS. *Chemical Geology* **233**, 196–216 (2006).
- 51 Jin, J., Zhan, R. & Wu, R. Equatorial cold-water tongue in the Late Ordovician. *Geology* **46**, 759–762

- (2018). <https://doi.org:10.1130/g45302.1>
- 52 Trotter, J. A., Williams, I. S., Nicora, A., Mazza, M. & Rigo, M. Long-term cycles of Triassic climate change: a new $\delta^{18}\text{O}$ record from conodont apatite. *Earth and Planetary Science Letters* **415**, 165-174 (2015). <https://doi.org:10.1016/j.epsl.2015.01.038>
- 53 Bond, D. P. & Grasby, S. E. Late Ordovician mass extinction caused by volcanism, warming, and anoxia, not cooling and glaciation. *Geology* **48**, 777-781 (2020).
- 54 Finnegan, S. *et al.* The magnitude and duration of Late Ordovician–Early Silurian glaciation. *Science* **331**, 903-906 (2011).
- 55 Pucéat, E. *et al.* Revised phosphate–water fractionation equation reassessing paleotemperatures derived from biogenic apatite. *Earth and Planetary Science Letters* **298**, 135-142 (2010).
- 56 Thiagarajan, N. *et al.* Reconstruction of Phanerozoic climate using carbonate clumped isotopes and implications for the oxygen isotopic composition of seawater. *Proceedings of the National Academy of Sciences (U.S.A.)* **121**, e2400434121 (2024).
- 57 Scotese, C. R., Song, H., Mills, B. J. W. & van der Meer, D. G. Phanerozoic paleotemperatures: The earth's changing climate during the last 540 million years. *Earth-Science Reviews* **215** (2021). <https://doi.org:10.1016/j.earscirev.2021.103503>
- 58 Goldberg, S. L., Present, T. M., Finnegan, S. & Bergmann, K. D. A high-resolution record of early Paleozoic climate. *Proceedings of the National Academy of Sciences (U.S.A.)* **118**, e2013083118 (2021).
- 59 Liu, K., Jiang, M., Zhang, L. & Chen, D. A new high-resolution palaeotemperature record during the Middle–Late Ordovician transition derived from conodont $\delta^{18}\text{O}$ palaeothermometry. *Journal of the Geological Society* **179**, jgs2021-2148 (2022).
- 60 Shen, J., Yin, R., Algeo, T. J., Svensen, H. H. & Schoepfer, S. D. Mercury evidence for combustion of organic-rich sediments during the end-Triassic crisis. *Nature Communications* **13(1)**, 1307 (2022).
- 61 Longman, J., Mills, B. J., Manners, H. R., Gernon, T. M. & Palmer, M. R. Late Ordovician climate change and extinctions driven by elevated volcanic nutrient supply. *Nature Geoscience* **14**, 924-929 (2021).
- 62 Albanesi, G. L. & Barnes, C. R. Subspeciation within a Punctuated Equilibrium Evolutionary Event: Phylogenetic History of the Lower-Middle Ordovician *Paroistodus* *Originalis*–*P. Horridus* complex (Conodonts). *Journal of Paleontology* **74**, 492-502 (2000). [https://doi.org:10.1666/0022-3360\(2000\)074<0492:Swapee>2.0.Co;2](https://doi.org:10.1666/0022-3360(2000)074<0492:Swapee>2.0.Co;2)
- 63 Miller, A. I. & Mao, S. Association of orogenic activity with the Ordovician radiation of marine life. *Geology* **23**, 305-308 (1995).

Responses to Reviewers' Comments

Reviewer #1 (Remarks to the Author):

He Zhao et al., authors of the manuscript entitled “Basalt volcanism and erosion caused Ordovician climate cooling,” have thoroughly revised the original contribution in accordance with the reviewers' corrections and evaluations. The revision addresses in detail each of the issues raised by the referees. In particular, my specific comments have been fully taken into account. It should be noted that the incorporation of the information recently processed by the authors further enriches the original version of the manuscript and clearly makes it a more robust and almost new contribution, suitable for publication in Nature Communications. The conclusions are significant and supported by the data, inferences, and discussions on each topic addressed in relation to the progressive cooling of the Ordovician oceans, namely volcanism, weathering, and episodic anoxia. The authors lean toward massive volcanism as the main factor driving the cooling trend throughout the Ordovician and its relationship to the variable development of faunas over time.

Based on my understanding, I suggest publishing this article in Nature Communications, mediating the interpretation and final editorial decision.

Response: We sincerely thank the Reviewer for the time and positive feedback. We are pleased that our revisions have strengthened the manuscript and appreciate the endorsement of our conclusions on volcanism, weathering, and anoxia in Ordovician cooling. The Reviewer's insightful comments were invaluable for making these improvements. We are grateful for the recommendation to publish in *Nature Communications*.

Reviewer #2 (Remarks to the Author):

Review of the revised manuscript "Volcanism and basalt weathering drove Ordovician climatic cooling" submitted to NC by Zhao et al. (MS: NCOMMS-25-31073A)

General comments

Thank you for providing me with the opportunity to read through this revised version of this manuscript. The authors have made significant improvements to the original version and have obviously given good consideration to comments of the reviewers. In

this study, the authors present a detailed Ordovician Hg-O-Sr isotopic chemostratigraphy and convincingly demonstrate the important role of volcanic activity in driving climate change during the Ordovician. Unlike previous studies that primarily focused on volcanic activity near the Ordovician–Silurian (O/S) boundary, the novelty of this work lies in the first-time identification of two possible large-scale volcanic events during the Early to Middle Ordovician, along with an exploration of their coupling with global cooling episodes. This finding offers a new perspective on the mechanisms driving Ordovician climate change and provides important context for understanding biological evolution during this period. I believe the study not only enhances our understanding of Ordovician climate dynamics but also contributes valuable insights into the interactions between early Earth's environment and the evolution of life. As such, I recommend acceptance after minor revisions.

Response: We sincerely thank the Reviewer for the positive feedback and for recommending our manuscript for acceptance. We are delighted that the Reviewer finds our revised manuscript significantly improved and recognizes the novelty of identifying Early-Middle Ordovician volcanic events and their climate linkages. All further revisions suggested have been carefully addressed in the point-by-point responses below and incorporated into the manuscript.

Other comments

Fig.2: The bottom age of the Wufeng Formation is approximately ~499 Ma. Please mark the major biological evolution events (GOBE, LOME) on the biodiversity curve (I).

Response: We thank the reviewer for this suggestion. Accordingly, we have marked the positions of both the GOBE and LOME events next to the biodiversity curve in Figure 2.

The suggested age of “~499 Ma” appears to have been a typographical error, as this value falls outside the Ordovician Period (~486.85–443.07 Ma) according to the Geologic Time Scale 2020 (Goldman et al., 2020). The reviewer’s query prompted us to re-examine the age constraint for the base of the Wufeng Formation. Based on previous studies (Zhang et al., 2019), the base of the Wufeng Formation in South China correlates with the *Dicellograptus complexus* graptolite Zone, which has a basal age of ~449 Ma (Goldman et al., 2020, Fig. 20.2) and is globally correlatable. We recognize that the earlier label was indeed a typographical error (449 vs. 499). Accordingly, in the revised Figure 2, we have placed the base of the Wufeng Formation at ~449 Ma, which is consistent with the age model in our data table and maintains alignment with the

discussions and conclusions of the manuscript.

Zhang, Y. *et al.* Ordovician integrative stratigraphy and timescale of China. *Science China Earth Sciences* **62**, 61-88 (2019).

Goldman, D. *et al.* in *Geologic Time Scale 2020* 631-694 (Elsevier, 2020).

L276-278: Delete “However, the lack of high-resolution age models for the study sections precludes detailed examination of the relationship between sedimentation rate and Hg enrichment.”

Response: We have removed this sentence from the main text.

Reviewer #3 (Remarks to the Author):

This revised manuscript presents new evidence (e.g., carbonate-free Hg content, Hg_{cf}) and expands the discussion on the potential sources and causes of Hg. This supports the idea that the Hg in this study originates primarily from volcanism. Furthermore, the authors compiled geological records of Hg and bentonite from global sections, enabling a more accurate comparison between geochemical data and geological records. This new data is crucial for establishing Hg as a reliable indicator of global volcanism during the Ordovician period. The authors also compiled published Sr-C-Nd isotope and temperature data to consolidate the stratigraphic framework, verify changes in global temperature, and enhance the explanation of Hg changes. They also fully considered changes in the paleolatitude of South China during the Ordovician period. I believe that the new dataset and its interpretation in the current version are much more robust and reliable, and that most of my concerns have been addressed. Therefore, I believe it could be accepted for publication after very minor revisions.

Response: We sincerely thank the Reviewer for their positive evaluation of our revised manuscript and for acknowledging the robustness of the new datasets and interpretations. We are pleased that the additional evidence, including carbonate-free Hg content (Hg_{cf}) and global compilation of Hg and bentonite records, along with Sr–C–Nd isotopes and paleolatitude considerations, have strengthened the case for volcanic-derived Hg and its use as a global volcanism proxy during the Ordovician. We are grateful for the Reviewer's support and have carefully addressed all further comments as detailed in the point-by-point responses below.

Figure 4 in the main text lacks discussion. The data in the different panels is not explained well either. I suggest removing it, as it is not key evidence for your discussion.

Response: We thank the reviewer for this comment. In response to the observation that Figure 4 (showing geological responses to the Ordovician LIPs) received limited discussion in the main text, we have moved it to the Supplementary Information as the new Supplementary Figure 13. We believe this placement allows the figure to still serve its purpose of helping readers quickly grasp the evidence for LIP activity, while keeping the main text focused. We hope this adjustment is acceptable.

I am slightly concerned about the lithology of the bentonite that you identified in Figure S2E of the Dawan Formation. To substantiate the presence of bentonite in your study sections of the Dawan Formation, you should provide more solid evidence, such as photographs of the thin section. I recently found published data on bentonite from the Darriwilian Kuniutan Formation in Wangjiawan (YW2 core), close to the sections studied in this research (see link below). This data provides detailed isotope ages and lithological features that could strengthen argument for the presence of bentonite during the Middle Ordovician period.

Reference: Globally synchronous meteorite rain during the Middle Ordovician.

Response: We thank the reviewer for this suggestion. First, we have provided an enhanced perspective in Supplementary Figure 2E to better display the characteristics of the volcanic ash layer. Furthermore, following the reviewer's recommendation, we have added photomicrographs (both plane- and cross-polarized light image) of thin sections from the volcanic ash layers in both the Dawan and Wufeng Formations as new Supplementary Figures 2I-2L. These new images show typical minerals, such as quartz, feldspar and sericite, and clay mineral, rock fragment as well as vesicle, thus providing further petrographic support for the identification of these layers as volcanic ash deposits.

We also thank the reviewer for drawing our attention to this reference, which provides supporting evidence for the presence of Middle Ordovician volcanic ash deposits in the study area. The suggested publication has now been cited in Supplementary Note S5.

Reviewer #4 (Remarks to the Author):

Response: We thank the reviewer for their time and contribution to the peer-review process of our manuscript. We appreciate their participation in *Nature Communications'* initiative to support early-career researchers in peer review, and we value their input on our work.

Córdoba, Argentina, May 15, 2025

I have reviewed the paper by He Zhao et al. entitled “Volcanism and basalt weathering drove Ordovician climatic cooling”. This study is a classic chemostratigraphic work on isotopy (O, Hg, Sr) under biostratigraphic control of conodonts that integrates paleontological data on eutracheophyte precursors, and Total Organic Carbon analysis in the Ordovician of South China. Its purpose is understanding the cause of the cooling of the Ordovician Period and the subsequent diversification of life. The manuscript is well done and suitable for this journal.

The article is well written, almost free of typographical errors (indicated in post scriptum). The manuscript consists of an abstract, an introductory part, results on O, Sr and Hg isotopy, discussion on volcanic activity and long-term global cooling, including early evolution of land plants and TOC implications. In addition, a section on methods is developed. References, 3 composite figures and supplementary material are necessary complements to the text, in good shape and quality for publication.

The study is correct to my knowledge, recording a series of isotopic curves in 3 sections of S China under previously established conodont biostratigraphic control, whose references should be added. The authors argue that the Ordovician global cooling could have been caused by a complex interplay of CO₂ uptakes by early plant propagation and weathering of basalts after intense volcanism. In the specialized part of the article that concerns my knowledge, I urge the author to correct the names of several genera of conodonts, as indicated in the post scriptum and in the attached reviewed paper. Also, it is suggested to add a number of significant references throughout the manuscript.

I consider that the manuscript would be ready for publication after a moderate revision, on typographical errors and figure formats, and especially revising the nomenclature of conodont genera. In addition, take care of the references to be added as suggested, following the indications of the post scriptum and the attached reviewed document. This contribution to Nature Communications is an original and valuable work that could be considered for publication after present review and decision of editorial evaluation.

Sincerely

Guillermo L. Albanesi

PS:

Suggestions and corrections:

- Line 81: See also: Barnes, C.R., 2004, Ordovician oceans and climate, in Webby, B.D., Paris, F., Droser, M.L., and Percival, I.G., eds., The Great Ordovician Biodiversification Event: New York, Columbia University Press, p. 72–76.

- Lines 102-103: add the references of the papers related to the conodont biostratigraphy of the 3 stratigraphic sections analyzed.
 - Line 183: Floian.
 - Line 191: Fm,
 - Line 275: Also well reported in the Argentine Precordillera: Huff, W.D., Bergström, S.M., Kolata, D.R., Cingolani, C.A. and Astini, R.A. 1998. Orrlovician K-bentonites in the Argentine Precorrlillera: relations to Gondwana margin evolution. In: Pankhurst, R.I. and Rapela, C.W. (eds.) The Proto-Andean Margin of Gondwana. Geological Society, London, Special Publications, 142, 107-]26.
 - Line 278: See also: Barnes, C.R., 2004, Was there an Ordovician superplume event?, in Webby, B.D., Paris, F., Droser, M.L., and Percival, I.G., eds., The Great Ordovician Biodiversification Event: New York, Columbia University Press, p. 77-80.
 - Line 311: Add a reference for the Ordovician System divisions and chrono-biostratigraphy, e.g. Goldman, D., Leslie, S. A., Liang, Y. and Bergström, S.M., 2022. Ordovician biostratigraphy: index fossils, biozones and correlation. In: Harper, D. A. T., Lefebvre, B., Percival and I. G., Servais, T. (eds.), A Global Synthesis of the Ordovician System: Part 1. Geological Society, London, Special Publications, 532, 31–62.
- Also, clarify that, aside on the left margin of the panel A, the formations are named.
- Line 429: See also: Ramos, V.A., Escayola, M., Mutti, D. and Vujovich, G.I., 2000. Proterozoic-early Paleozoic ophiolites of the Andean basement of southern South America. In: Dilek, Y., Moores, E.M., Elthon, D. and Nicolas, A. eds. Ophiolites and Oceanic Crust: New insights from field studies and the Ocean Drilling Program: Boulder, Colorado, Geological Society of America, Special Paper 349: 331-349.
 - Line 448: See also: Albanesi, G.L. and Barnes, C.R., 2000, Subspeciation within a punctuated equilibrium evolutionary event: Phylogenetic history of the Lower-Middle Ordovician *Paroistodus originalis*–*P. horridus* complex (Conodonta). *Journal of Paleontology*, v. 74, p. 492–502, doi: 10.1666/0022-3360(2000)074<0492:SWAPEE>2.0.CO;2.
 - Line 491: delete “is”.
 - Line 530: The amount of conodont elements (absolute frequency) per sample should be indicated. Also indicate how many pelagic conodont elements per sample were used for geochemical analyses.
 - Lines 531-533: Caution: several genera of conodonts are misspelled and/or their identification are doubtfuls: *Belodella* (probably *Ansella*), *Pasoistodus* (probably *Paroistodus*), *Pesiodus* (? *Periodon*), *Sessagtognathus* (probably *Serratognathus*), *Trianglodus* (probably *Triangulodus*). In supplementary material you should correct: *Conurodus* (= *Cornuodus*), *Protopandesodus* (= *Protopanderodus*), *Nususguathus* (= *Nasusgnathus*), *Belodena* (= *Belodella*), *Juanogn* (= *Juanognathus*).

-

Dr. Guillermo L. Albanesi